# Soft-Di[M]O: Improving One-Step Discrete Image Generation with Soft Embeddings

**Yuanzhi Zhu**[1]   **Xi Wang**[1]   **Stéphane Lathuilière**[2]   **Vicky Kalogeiton**[1]

[1]LIX, École Polytechnique, CNRS, IPP    [2]Inria, Univ. Grenoble Alpes, CNRS, LJK

## Abstract

One-step generators distilled from Masked Diffusion Models (MDMs) compress multiple sampling steps into a single forward pass, enabling efficient text and image synthesis. However, they suffer two key limitations: they inherit modeling bias from the teacher, and their discrete token outputs block gradient flow, preventing post-distillation refinements such as adversarial training, reward-based fine-tuning, and Test-Time Embedding Optimization (TTEO). In this work, we introduce soft embeddings, a simple relaxation that replaces discrete tokens with the expected embeddings under the generator's output distribution. Soft embeddings preserve representation fidelity for one-step discrete generator while providing a fully differentiable continuous surrogate that is compatible with teacher backbones and tokenizer decoders while cause minimum bias. Integrating soft embeddings into the Di[M]O (Zhu et al., 2025b) distillation framework (denoted Soft-Di[M]O) makes one-step generators end-to-end trainable and enables straightforward application of GAN-based refinement, differentiable reward fine-tuning, and TTEO. Empirically, across multiple MDM teachers (e.g., MaskBit (Weber et al., 2024), MaskGen (Kim et al., 2025)), Soft-Di[M]O achieves state-of-the-art one-step results: improved class-to-image performance, a one-step FID of 1.56 on ImageNet-256 with GAN-based refinement, along with higher than teacher GenEval (Ghosh et al., 2023) and HPS (Wu et al., 2023) scores on text-to-image with reward fine-tuning, and further gains from TTEO.

## 1 Introduction

Masked Diffusion Models (MDMs) generate discrete data by iteratively replacing masked tokens with predicted unmasked ones. They have recently achieved strong results across diverse domains, including text generation (Austin et al., 2021; Lou et al., 2023; Nie et al., 2024; 2025; Lou et al., 2024; Gong et al., 2024; Deschenaux & Gulcehre, 2024; Sahoo et al., 2025; Shi et al., 2024; Arriola et al., 2025), image synthesis (Chang et al., 2022; Gu et al., 2022; Chang et al., 2023; Hu & Ommer, 2024; Bai et al., 2024; Patil et al., 2024; Issenhuth et al., 2021; Kim et al., 2025; Weber et al., 2024), and multi-modal understanding and generation (Hu et al., 2022; Li et al., 2024d; Xie et al., 2024a; Wang et al., 2024a; Yang et al., 2025; You et al., 2025; Yu et al., 2025; Li et al., 2025; Wang et al., 2025a; Shi et al., 2025b). Building on this success, recent work has demonstrated that these multi-step procedures can be distilled into one-step processes, substantially accelerating inference and reducing computational cost while maintaining competitive quality (Zhu et al., 2025b; Sahoo et al., 2025; Yoo et al., 2025).

Despite their remarkable performance, current one-step discrete generators face two fundamental limitations: First, they inherit both *modeling* and *alignment* errors from the teacher: imperfections in the teacher's generative modeling and the under-alignment of the pre-trained teacher prior to reward fine-tuning, both error types of teacher are transmitted to the student during data-free distillation (Zhou et al., 2024b; Luo et al., 2023b; Yin et al., 2024b). Second, their discrete token outputs block gradient flow, preventing the application of various key refinement techniques such as GAN training (Yin et al., 2024a; Zhou et al., 2024b), differentiable reward fune-tuning (Luo, 2024; Luo et al., 2024), and Test-Time Embedding Optimization (TTEO) (Eyring et al., 2024). As a result, cur-

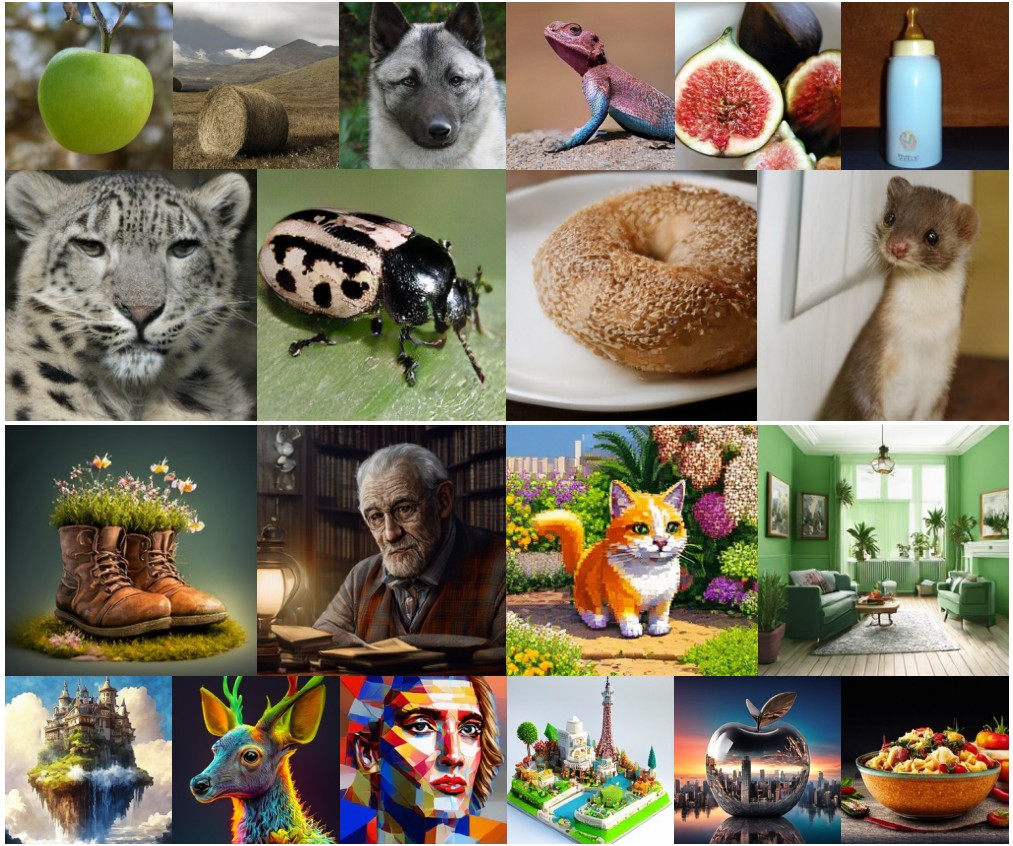

Figure 1: Qualitative results produced by our one-step generators distilled from MaskBit (Weber et al., 2024) (top) and MaskGen-L (Kim et al., 2025) (bottom).

rent distilled MDM generators have limited flexibility to leverage external knowledge, or incorporate user preferences to better align with downstream objectives.

The gradient-blocking problem is common across discrete modeling tasks. Prior remedies include policy-gradient estimators such as REINFORCE (Williams, 1992; Sutton et al., 1999; Ranganath et al., 2014) and differentiable relaxations (e.g., Gumbel–Softmax, soft-argmax) (Gumbel, 1954; Jang et al., 2016; Zhang et al., 2017; Gu et al., 2018). However, these methods often introduce biased or high-variance gradients and numerical instabilities (Jang et al., 2016; Liu et al., 2023a), and they have not been systematically applied to one-step discrete generators.

To address these limitations, we introduce a minimal-change, decoder-compatible relaxation of discrete outputs, called *soft embeddings*, tailored to one-step discrete generators. We define soft embedding as the expected token embedding under the generator's output probability distribution, serving directly as a continuous surrogate for discrete outputs. This simple construction: (i) maintains representation fidelity, as when the model is confident and its output distribution is sharply concentrated, the soft-embedding collapses to the corresponding discrete embedding, thereby mitigating the high-variance issue and stablizing training; (ii) is fully differentiable, enabling end-to-end gradient propagation through the generator logits; (iii) offers compatibility with the existing teacher backbone and tokenizer decoder; and (iv) provide small bias and very low variance as true loss surrogate.

Soft embeddings enable continuous training of one-step discrete generators, thus unlock refinements techniques that were previously inaccessible: (i) adversarial refinement (GAN training): discriminators operate in the encoded space while gradients backpropagate through the teacher backbone into the generator; (ii) differentiable reward fine-tuning: the one-step generator can be fine-tuned with the gradient from the differentiable reward functions; (iii) TTEO: once trained, the generator's input embeddings corresponding to $x_{\text{init}}$ are optimized at inference time with respect to reward signals to maximize reward-based metrics.

In this work, we combine the proposed soft embeddings with the prior state-of-the-art one-step generator Di[M]O, yielding Soft-Di[M]O. The continuous nature of soft embeddings enables Soft-Di[M]O to seamlessly incorporate the three post-distillation refinement techniques discussed above. In summary, our contributions are: (i) we introduce soft embeddings as a new representation for the outputs of one-step generators; (ii) we demonstrate that this representation enables *end-to-end differentiability* for methods previously inaccessible to discrete outputs, including: GAN training, differentiable reward fine-tuning, and test-time embedding optimization; (iii) we extend Di[M]O to additional MDM teachers and visual tokenizers, and empirically achieve state-of-the-art one-step performance on both class-to-image and text-to-image tasks.

## 2 PRELIMINARY: MASKED DIFFUSION MODELS AND DI[M]O

**Masked Diffusion Models (MDMs).** MDMs (Chang et al., 2022; Shi et al., 2025a; Hu & Ommer, 2024) generate discrete data by learning to reverse a discrete, absorbing-state diffusion process. For image modeling, a pre-trained tokenizer (e.g., VQ-VAE (Van Den Oord et al., 2017; Esser et al., 2021)) is used: the encoder maps an image into a sequence of discrete tokens $x_0 = \{x_0^i\}_{i=1}^L$, where $L$ is the sequence length, and the decoder reconstructs the image from this token sequence.

In the **forward process**, the original token sequence $x_0$ is gradually corrupted into a fully masked sequence $x_1$. At a timestep $t \in [0, 1]$, each token is independently masked with probability $r_t$, yielding an intermediate sequence $x_t$ containing both image and mask tokens.

The **reverse process** iteratively refines the sequence by filling the mask tokens. A model $\phi$ predicts a categorical distribution $p_\phi(x_0^i|x_t) := \text{softmax}(z_\phi^i(x_t))$ for each masked token at position $i$, where $z_\phi^i(x_t)$ is the predicted logits. Logits $z_\phi(x_t)$ has shape of $[L, V]$, where $L$ is the sequence length and $V = |\mathcal{V}|$ is the size of the vocabulary. Starting from a fully masked sequence $x_1$, masked tokens are replaced by tokens sampled from $p_\phi(x_0|x_t)$, while unmasked tokens remain unchanged, progressively refining the sequence back to $x_0$ over multiple steps.

MDM is trained to minimize the cross-entropy loss for reconstructing the original tokens from their masked versions:
$$\mathcal{L}_{\text{MDM}} = \mathbb{E}_{x_0,t} \left[ \mathbb{E}_{q_{t|0}}[-\log p_{0|t}(x_0|x_t, \phi)] \right], \tag{1}$$
where $t$ is uniformly sampled and the inner expectation is over the forward process. This objective enables efficient, parallel token prediction.

**Di[M]O Distillation.** To address the sampling inefficiency of multi-step MDMs, Zhu et al. (2025b) proposed Di[M]O, which distills a multi-step MDM into a one-step generator. Built on on-policy distillation (Agarwal et al., 2024; Ross et al., 2011; Ho et al., 2016; Balakrishna et al., 2020; Arora et al., 2022; Schmidt, 2019), the framework operates by first sampling a one-step prediction from the student. This output is then corrupted by the forward masking process to produce an intermediate state $\tilde{x}_t$. The objective is to match the teacher's and student's output distributions conditioned on $\tilde{x}_t$.

Two key designs enable Di[M]O to work effectively in practice: (i) an auxiliary model is introduced to approximate the training objective and its gradient, following strategies in prior work (Poole et al., 2022; Huang et al., 2024; Zhou et al., 2024c); and (ii) an entropy-preserving token initialization strategy is adopted to prevent student from mode collapsing when starting from the low-entropy, fully masked state. With these components, we obtain the following approximation to the gradient of the Di[M]O loss $\mathcal{L}_{\text{Di[M]O}}(\theta)$, which is used to update the student parameters $\theta$:

$$\nabla_\theta \mathcal{L}_{\text{Di[M]O}}(\theta) \approx \mathbb{E}_{x_{\text{init}},t} \left[ w(t) \left( \mathbb{E}_{q_{t|0}} \left[ \nabla_{z_\psi} D_{\text{div}}(p_\phi||p_\psi)(\tilde{x}_t) \frac{dz_\theta(x_{\text{init}})}{d\theta} \right] \right) \right], \tag{2}$$

where $x_{\text{init}} \sim p_{\text{init}}$ is sampled from a designed initialization distribution $p_{\text{init}}$, $z_\theta$ represents the student's output logits, $\tilde{x}_t$ denotes the intermediate state obtained by applying the forward mask kernel $q_{t|0}$ to the sampled tokens $x_\theta$, and $D_{\text{div}}(\cdot|\cdot)$ denotes the chosen divergence measure (summed over all masked token positions) between the distributions from the teacher ($\phi$) and the auxiliary model ($\psi$). Class condition and text condition $c$ is omitted for simplicity. The scalar function $w(t)$ balances the contribution of different mask ratios $t$ to the overall objective. The auxiliary model is trained with Eq. (1) with $x_\theta$ as clean tokens. Notably, this loss gradient is applied directly to the generated logits, avoiding the need to backpropagate through the non-differentiable sampling operation.

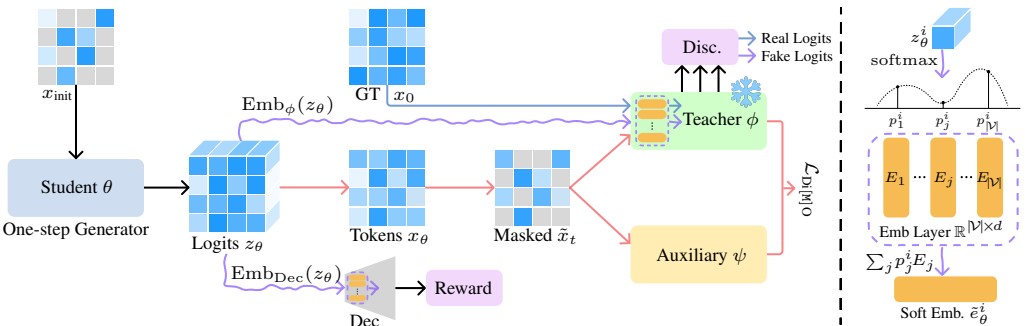

Figure 2: **Soft-Di[M]O Pipeline.** Given a sampled initialization $x_{\text{init}}$, the one-step generator (student $\theta$) outputs logits $z_\theta$; discrete image tokens $x_\theta = \{x_\theta^i\}_{i=1}^L$ are sampled from these logits and used to form the Di[M]O distillation loss (red path). In parallel, the logits are converted into *soft embeddings* (purple path); these differentiable soft embeddings permit various supervision for post-training, such as GAN loss and reward loss, allowing the student to compete with or even surpass the teacher's performance. The right panel illustrates the soft-embedding construction for each position $i$: logits $z_\theta^i \xrightarrow{\text{softmax}} p^i$ and $\sum_j p_j^i E_j$ yield $\tilde{e}_\theta^i$. We use $p_j^i = p_\theta(x_0^i = j|x_{\text{init}})$ for simplicity. More details on discriminator is visualized in Fig. 5.

## 3 METHODS

While one-step generators distilled from MDMs can be fast and effective, existing approaches (e.g., Di[M]O (Zhu et al., 2025b)) implicitly assume that the teacher fully captures the target distribution. Consequently, the student model inherits any modeling errors present in the teacher. Additionally, since the student produces discrete tokens, gradients cannot propagate through its outputs, preventing the direct application of standard post-training techniques such as GAN objectives and differentiable reward fine-tuning. To overcome these limitations, we build on the idea of continuous relaxations (e.g., Gumbel-Softmax (Gumbel, 1954; Jang et al., 2016), see Appendix E) and introduce a *soft embedding*, defined as the expected token embedding under the generator's output probability (Sec. 3.1). The soft embedding is compatible with the teacher backbone and tokenizer decoder while preserving end-to-end differentiability. Leveraging this representation, the distilled one-step student can incorporate various post-training and refinement techniques (Sec. 3.2), including GAN training (Sec. 3.2.1), differentiable reward fine-tuning (Sec. 3.2.2), and Test-Time Embedding Optimization (TTEO, Sec. 3.2.3). The overall training pipeline is illustrated in Fig. 2.

### 3.1 SOFT EMBEDDINGS

#### 3.1.1 INTRODUCTION OF SOFT EMBEDDING

For each masked position, the soft embedding is computed from the model's predicted logits at that position. Taking masked position $i$ as an example (right panel of Fig. 2), we begin with the model's predicted logits $z_\theta^i$. These logits are first converted into a probability distribution over the entire vocabulary $\mathcal{V}$ via the softmax function: $p_\theta(x_0^i|x_{\text{init}}) = \text{softmax}(z_\theta^i)$. This distribution $p_\theta$ represents the model's confidence for each possible token $j$ filling the masked position. Instead of conventional discrete token sampling, the soft embedding $\tilde{e}_\theta^i$ is computed as the expectation of token embeddings under this probability distribution $p_\theta(x_0^i|x_{\text{init}})$. Concretely, let $E \in \mathbb{R}^{|\mathcal{V}|\times d}$ denote the embedding matrix (from either the teacher backbone or the tokenizer decoder), and let $E_j$ be the $d$-dimensional embedding vector for token $j$ in the vocabulary $\mathcal{V}$. The soft embedding is then computed as:

$$\tilde{e}_\theta^i \; := \; \text{Emb}(z_\theta^i) = \sum_{j=1}^{|\mathcal{V}|} p_\theta(x_0^i = j|x_{\text{init}})\,E_j \; = \; E^\top p_\theta(x_0^i|x_{\text{init}}). \tag{3}$$

In other words, the operator $\text{Emb}(\cdot)$ here maps logits (via softmax) to a convex combination of all the token embeddings. We use $\text{Emb}_\phi$ and $\text{Emb}_{\text{Dec}}$ to denote this operation using the embedding layer from teacher $\phi$ and the tokenizer decoder, respectively.

#### 3.1.2 BENEFITS OF SOFT EMBEDDING

This design has three practical benefits. First, **representation fidelity**: high representation fidelity is essential for soft embeddings to serve as an effective approximation of tokens sampled from log-

its. Empirically, one-step generators distilled from MDMs (e.g., Di[M]O (Zhu et al., 2025b) and ReDi (Yoo et al., 2025)) often concentrate probability mass on only a few candidate tokens per position because they transfer randomness into the initial sequence (see Tab. 14 for more validation). Consequently, the soft embedding representation typically closely matches the embedding of the sampled token, with only a small approximation error between the continuous and discrete representations. We provide empirical verification for this representation fidelity in the Appendix I.2. Second, **differentiability**: the mapping from logits to soft embeddings is fully differentiable, allowing gradients from any downstream objective (e.g., a discriminator or a differentiable reward) to propagate back to the generator. This enables end-to-end optimisation for adversarial refinement and reward-based fine-tuning without resorting to non-differentiable sampling or gradient estimators. Third, **compatibility**: the soft embedding $\tilde{e}_\theta$ lies in the same embedding space used by the teacher or tokenizer decoder. As a result, soft embeddings can be fed directly into these components in the generation pipeline without additional projection layers.

### 3.1.3 THEORETICAL JUSTIFICATION OF SOFT EMBEDDINGS.

The soft-embedding operator admits a principled theoretical justification as a surrogate for discrete token sampling. Let the ideal objective be the expectation over possible discrete tokens:

$$\mathcal{L}_{\text{orig}}(\theta) = \mathbb{E}_{j \sim p_\theta}\big[f(e_j)\big], \tag{4}$$

where $f$ denotes the differentiable downstream objective (e.g., teacher backbone + discriminator head, or tokenizer decoder + reward, introduced later in Sec. 3.2). Since sampling is non-differentiable, we optimize the surrogate $\mathcal{L}_{\text{soft}}(\theta) = f(\tilde{e})$, where $\tilde{e} = E^\top p_\theta$ is the soft embedding introduced in Sec. 3.1.1. The validity of this relaxation follows from two theoretical observations.

**Second-order bias under concentrated logits (Lemma G.3).** Let $\Sigma := \mathbb{E}_{j \sim p_\theta}\big[(e_j - \tilde{e})(e_j - \tilde{e})^\top\big]$ be the embedding covariance under $p_\theta$. Lemma G.3 shows that under standard smoothness assumptions,

$$\big|f(\tilde{e}) - \mathbb{E}_j[f(e_j)]\big| \leq \tfrac{1}{2} L \|\Sigma\|. \tag{5}$$

Thus, the soft-embedding surrogate incurs only a *second-order* bias in the embedding variance. For one-step generators distilled from MDMs, the output logits are highly concentrated (empirically validated in previous work Di[M]O (Zhu et al., 2025b) and ReDi (Yoo et al., 2025) and our Fig. 7 and Tab. 14). If most probability is concentrated on token $j^*$ with top -1 probability $p_\theta(j^*) = 1 - \varepsilon$, then $\|\Sigma\| = \mathcal{O}(\varepsilon)$, making the bias term $\mathcal{O}(\varepsilon)$ and therefore negligible. This explains why the continuous embedding $\tilde{e}$ closely matches the embedding of the sampled token.

**Gumbel-Softmax Straight-Through (ST) exhibits first-order bias (Lemma G.4).** Lemma G.4 shows that the gradient estimator of Gumbel-Softmax straight-through satisfies

$$\big\|\mathbb{E}\big[\nabla_\theta \mathcal{L}_{\text{ST}}\big] - \nabla_\theta \mathcal{L}_{\text{orig}}\big\| \leq C \, \mathbb{E}[\|\hat{e}_{\text{hard}} - \hat{e}_{\text{soft}}\|] + \mathcal{O}(\tau) + \mathcal{O}(\|\Sigma\|). \tag{6}$$

The dominant term arises from the mismatch between the *hard* forward token and the *soft* backward relaxation, producing a *first-order* bias that does not vanish even in the limit $\tau \to 0$ unless the logits are exactly one-hot. Moreover, the injected Gumbel noise yields high gradient variance, which further destabilizes adversarial or reward-based optimization.

Soft embedding therefore provides an exceptionally stable surrogate with: (i) second-order bias that shrinks as logits concentrate (Lemma G.3), (ii) deterministic, variance-free pathwise gradients, and (iii) no forward/backward mismatch, unlike Gumbel-ST (Lemma G.4). These theoretical properties explain why soft embeddings is a good candidate to enable the refinement techniques described next.

### 3.2 REFINEMENT TECHNIQUES UNLOCKED WITH SOFT EMBEDDINGS

#### 3.2.1 ADVERSARIAL FINE-TUNING

The proposed soft embeddings allow us to feed generator outputs directly into a frozen, pre-trained teacher backbone and to backpropagate discriminator gradients to the generator logits. Building on this, we design a discriminator and a corresponding training protocol tailored to soft embeddings.

**Discriminator architecture.** We reuse the pre-trained teacher as a frozen multi-scale feature extractor and add lightweight discriminator heads on top of the extracted features (Sauer et al., 2024a; Chen et al., 2025b). Since the soft embedding is computed using the teacher's embedding layer, the remaining backbone layers can directly process it as input. For each selected transformer layer, we add a convolutional head to aggregate spatial information; the resulting outputs are concatenated along the channel dimension and passed through a final MLP to produce the discriminator logit. **Masked-embedding input augmentation.** As the teacher model is pre-trained on masked sequences, to keep the discriminator inputs close to the teacher's distribution and to increase robustness as in diffusion GAN (Wang et al., 2022; Lin et al., 2024; Yin et al., 2024a), we apply random masking to the sequence of soft embeddings fed into the backbone when computing the adversarial features. Specifically, similar to the diffusion GAN where the discriminator received Gaussian-noised input, we replace a random number of the computed soft embeddings or the ground truth embeddings with mask embeddings to predict fake and real logits, respectively. In this work, we use $\mathrm{Emb}_\phi(\cdot)$ to denote the soft embedding operation with the embedding layer of teacher model $\phi$. This operator also maps one-hot tokens from ground truth image token sequence to the corresponding token embeddings. We then use $\mathrm{Emb}_\phi(x_{\mathrm{gt}})_r$ and $\mathrm{Emb}_\phi(z_\theta)_r$ to denote the masked embedding sequences with mask ratio $r$ as input to the teacher backbone.

**GAN objective.** Let $p_{r_{\mathrm{GAN}}}$ be the distribution over mask ratios used for discriminator augmentation and $p_{\mathrm{init}}$ the generator's initialization distribution. The GAN training objective in our method is:

$$\min_{G_\theta} \max_{D_\eta} \mathbb{E}_{x_{\mathrm{gt}}\sim p_{\mathrm{data}},r\sim p_{r_{\mathrm{GAN}}}}[\log D_\eta(\mathrm{Emb}_\phi(x_{\mathrm{gt}})_r, r)] + \mathbb{E}_{x_{\mathrm{init}}\sim p_{\mathrm{init}},r\sim p_{r_{\mathrm{GAN}}}}[\log(1-D_\eta(\mathrm{Emb}_\phi(z_\theta)_r, r))],$$

$$(7)$$

where $D_\eta$ is the full discriminator (frozen backbone + trainable heads). Class condition and text condition is omitted for simplicity.

For stable generator updates we use the non-saturating GAN loss (Goodfellow et al., 2014; Poole et al., 2016; Shannon et al., 2020) for the generator:

$$\mathcal{L}_{\mathrm{GAN}}(\theta) = \mathbb{E}_{x_{\mathrm{init}}\sim p_{\mathrm{init}},r\sim p_{r_{\mathrm{GAN}}}}[-\log(D_\eta(\mathrm{Emb}_\phi(z_\theta)_r, r))].$$

$$(8)$$

Thanks to the differentiability of operator $\mathrm{Emb}_\phi(\cdot)$, gradients from $D_\eta$ with respect to its input can propagate through the masking operator and the embedding matrix back to the generator logits $z_\theta$, enabling effective end-to-end adversarial training.

### 3.2.2 DIFFERENTIABLE REWARD FINE-TUNING

*Prompt-following* and *aesthetic quality* are the two primary criteria in text-to-image generation (Luo, 2024; Luo et al., 2024; Wang et al., 2025b). To directly optimize these criteria, the research community has increasingly adopted learned reward models (Wu et al., 2023; Radford et al., 2021; Black et al., 2023) and human-preference signals (Wallace et al., 2024; Liu et al., 2025a; Kirstain et al., 2023; Yang et al., 2024), which supply scalable, task-aligned supervision that complements likelihood-based training. However, since reward functions typically operate on decoded images, directly optimizing such rewards is impractical for discrete generators because gradients cannot pass through the tokenizer decoders. By using soft embeddings to make model outputs differentiable with respect to logits, we can now fine-tune the one-step generator end-to-end on differentiable reward functions. Concretely, we fine-tune with a set of differentiable rewards $\mathcal{R}$ (e.g., a CLIP score for prompt–image alignment (Radford et al., 2021) and ImageReward for aesthetics (Xu et al., 2023)):

$$\mathcal{L}_{\mathrm{reward}}(\theta) = -\sum_i \lambda_i \mathcal{R}_i(\mathrm{Dec}(\mathrm{Emb}_{\mathrm{Dec}}(z_\theta), c),$$

$$(9)$$

where $\lambda_i$ denotes the weighting for reward model $\mathcal{R}_i$, $\mathrm{Dec}(\cdot)$ is the tokenizer decoder, $\mathrm{Emb}_{\mathrm{Dec}}(\cdot)$ is the corresponding embedding operator, $z_\theta$ is the model output logits and $c$ is the prompt condition.

By mapping generator logits to soft embeddings using the embedding layer from the tokenizer, we can feed the resulting embedding sequence into the tokenizer's decoder and obtain a differentiable image. A reward score is then computed to construct loss and backward to the generator.

The final generator loss is:

$$\mathcal{L}_{\mathrm{gen}}(\theta) = \mathcal{L}_{\mathrm{Di[M]O}}(\theta) + w_{\mathrm{GAN}}\mathcal{L}_{\mathrm{GAN}}(\theta) + w_{\mathrm{reward}}\mathcal{L}_{\mathrm{reward}}(\theta),$$

$$(10)$$

where $w_{\mathrm{GAN}}$ and $w_{\mathrm{reward}}$ balance the contribution of the GAN and reward terms against the base Di[M]O distillation loss.

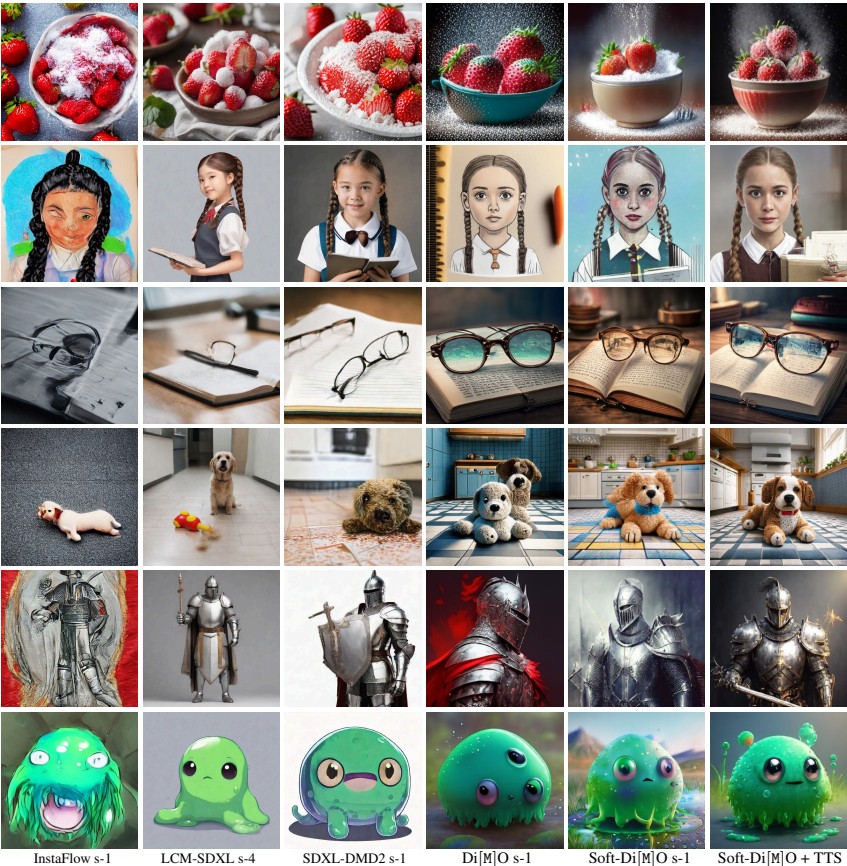

InstaFlow s-1     LCM-SDXL s-4     SDXL-DMD2 s-1     Di[M]O s-1     Soft-Di[M]O s-1     Soft-Di[M]O + TTS

Figure 3: Qualitative comparison of one- and few-step distillation methods by using the same text prompts. Di[M]O and Soft-Di[M]O are both from MaskGen-L teacher. The number of inference steps is indicated by 's'.

### 3.2.3 TEST-TIME EMBEDDING OPTIMIZATION

The soft embedding also enables Test-Time Scaling (TTS) (Eyring et al., 2024; Guo et al., 2024; Ma et al., 2025) by optimizing the embedding input of the one-step student. TTS for one-step generator involves spending extra inference compute to search over different initial inputs $x_{\text{init}}$, including through gradient-based optimization using the tokenizer decoder and through sampling-based methods like Best-of-N (BoN). Optimizing the discrete initial code sequence can be difficult and inefficient for MDMs; instead, since the embedding layer is fixed, we initialize from the embeddings corresponding to $x_{\text{init}}$ and directly optimize it. This optimization (TTEO) can be expressed as:

$$e^{\star} = \arg\max_{e_{in}} \mathcal{R}(\text{Dec}(\text{Emb}_{\text{Dec}}(z_{\theta}(e_{in}))), c). \tag{11}$$

where $e_{in}$ denotes the input embedding to be optimized, $c$ is the prompt text condition, and $z_{\theta}(e_{in})$ is the logits generated by the one-step generator with input $e_{in}$. We apply the same technique for the tokenizer decoder so that the entire test-time optimization remains fully differentiable, enabling a ReNO-style (Eyring et al., 2024) test-time scaling implementation.

## 4 EXPERIMENTS

In Sec. 4.1, we summarize the experimental setup. We evaluate our method on two representative tasks: class-to-image generation and text-to-image generation. Following prior work (Yin et al., 2024a; Zhou et al., 2024b; Luo, 2024; Li et al., 2024b), we pair the base distillation algorithm (Di[M]O) with different post-training techniques depending on the task. For class-to-image, we add adversarial fine-tuning (Di[M]O + GAN, see Sec. 4.2) because FID — our primary evaluation metric — measures distribution-level fidelity, and common reward-based objectives risk driving samples

away from the class-conditional target. For text-to-image, we instead use differentiable reward-based fine-tuning (Di[M]O + Reward, see Sec. 4.3) to directly optimize prompt adherence and aesthetics using CLIP-based (CLIP score) and learned aesthetic reward (ImageReward). In addition, we show that applying TTS to the one-step text-to-image generator can further improve prompt adherence and aesthetic quality. In Sec. 4.4 we ablate key design choices and hyperparameters.

## 4.1 EXPERIMENTAL SETUP

**Teacher Models and Datasets.** Together with MaskGit (Chang et al., 2022) and Meissonic (Bai et al., 2024) reported in Di[M]O, we also include MaskBit (Weber et al., 2024) and MaskGen (Kim et al., 2025) as stronger teachers for class-to-image and text-to-image generation, respectively. For both MaskGit and MaskBit, we use the ImageNet-256 (Deng et al., 2009) dataset. For text-to-image generation, since both Meissonic and MaskGen are SFTed on the aesthetic dataset, we conduct the corresponding experiment with synthetic images from the DALLE3-1M dataset (Egan et al., 2024) when using GAN training. And we use the prompts from LAION-Aesthetics-6+ dataset Cherti et al. (2023) for Di[M]O loss following the original implementation. Since our one-step student inherits many components and features from the teacher model, we list the key configurations including vocabulary sizes and output logits dimensions in Tab. 5.

**Evaluation.** The metrics we use for comparing ImageNet results are Fréchet Inception Distance (FID) (Heusel et al., 2017) and Inception Score (IS) (Salimans et al., 2016). We also use precision (Prec.) and recall (Rec.) (Kynkäänniemi et al., 2019), density (Den.) and coverage (Cov.) (Naeem et al., 2020) to further evaluate the fidelity and diversity of generated images. For text-to-image generation, we follow Zhu et al. (2025b) to measure the HPSv2 (Wu et al., 2023) for aesthetic quality and GenEval (Ghosh et al., 2023) and CLIP score for prompt following ability. We also measure the FID on COCO-30K (Lin et al., 2014) and MJHQ-30K (Li et al., 2024a), denoted as $FID_{COCO}$ and $FID_{MJHQ}$, respectively. In our experiments on ImageNet-256, we calculate the FID using 5k generated images for ablations and 50k generated for benchmarking (50 images per class), comparing the statistics of the generated images with the reference statistics from ADM (Dhariwal & Nichol, 2021) using Clean-FID (Parmar et al., 2022b). On top of the quantitative metrics, we also provide visual generations of different methods as qualitative comparison.

## 4.2 CLASS-TO-IMAGE GENERATION

**MaskGit teacher.** For MaskGit teacher, we initialize the generator from the provided one-step generator in Di[M]O, and use the same teacher model as Di[M]O. In Tab. 2, we present a quantitative performance comparison of our Soft-Di[M]O with various acceleration methods, including the high-order sampler $\theta$-trapezoidal (Ren et al., 2025), other distillation techniques (Hayakawa et al., 2024; Yoo et al., 2025), and the teacher model MaskGit (Chang et al., 2022). This result, a FID of 6.4 using only one forward pass, demonstrates that our method effectively improves Di[M]O, yielding superior metric scores that surpass all prior approaches using the same teacher model. These comparisons highlights the effectiveness of our method in matching the performance of multi-step teacher models while dramatically reducing the computational cost for inference.

**Benchmarking with MaskBit teacher.** MaskBit is a powerful MDM built on Lookup-Free Quantization (LFQ) (Yu et al., 2023) tokenizer, with relatively fewer parameters. With this strong teacher, we compare our method to leading deep generative models, including GANs, diffusion models, their distilled counterparts, and various variations, focusing on generation quality. We first reimplement Di[M]O on the MaskBit teacher, and we find that this baseline show limited generation diversity, as reflected by the recall (see Tab. 1). After adding GAN loss to the training, we achieve a FID of 1.98, lower than 2, which can be considerred as sufficient good for ImageNet-256 generation. With a further longer training with slightly change of hyperparameters, we got a state-of-the-art one-step FID of 1.56, comparable to the teacher model and beat the other GAN, diffusion models by a large margin, demonstrate the goodness of our method.

## 4.3 TEXT-TO-IMAGE GENERATION

**Result with Meissonic teacher.** Since the Meissonic model is fine-tuned with human preference data, we found that the teacher model distribution is already aligned with the reward encouraged dis-

Table 1: Class-conditional ImageNet-256 results with MaskBit teacher.

| | Method | Steps (↓) | #Params | FID(↓) | IS(↑) | Pr(↑) | Rec(↑) |
|---|---|---|---|---|---|---|---|
| Diffusion & Flow | ADM (Dhariwal & Nichol, 2021) | 250 | 554M | 4.59 | 186.7 | 0.82 | 0.52 |
| | U-ViT-H (Bao et al., 2023) | 50 | 501M | 2.29 | 263.9 | 0.82 | 0.57 |
| | DiT-XL/2 (Peebles & Xie, 2023) | 250 | 675M | 2.27 | 278.2 | 0.83 | 0.57 |
| | SiT-XL/2 (Ma et al., 2024) | 250 | 675M | 2.06 | 277.5 | 0.83 | 0.59 |
| Masked & AR | LlamaGen-3B (Sun et al., 2024) | 576 | 3.1B | 2.18 | 263.3 | 0.81 | 0.58 |
| | MAR (Li et al., 2024c) | 100 | 400M | 1.98 | - | - | - |
| | MAGVIT-v2 (Yu et al., 2023) | 64 | 307M | 1.78 | 319.4 | - | - |
| | MaskBit (Weber et al., 2024) | 64 | 305M | 1.66 | 320.0 | 0.81 | 0.60 |
| Few-Step from Scratch | iCT (Song et al., 2023) | 1 | 675M | 34.24 | - | - | - |
| | | 2 | 675M | 20.3 | - | - | - |
| | MeanFlow (Geng et al., 2025) | 1 | 676M | 3.43 | - | - | - |
| | | 2 | 676M | 2.93 | - | - | - |
| | + train longer | 2 | 676M | 2.20 | - | - | - |
| Discrete Distillation | LlamaGen-L-DD (Liu et al., 2024) | 2 | 326M | 7.58 | 237.5 | 0.84 | 0.37 |
| | Di[M]O-MaskGit (Zhu et al., 2025b) | 1 | 174M | 6.91 | 214.1 | 0.83 | 0.38 |
| | Di[M]O-MaskBit (Zhu et al., 2025b) | 1 | 305M | 2.89 | 310.1 | 0.87 | 0.49 |
| | Soft-Di[M]O-MaskBit | 1 | 305M | 1.96 | 281.4 | 0.84 | 0.55 |
| | + train longer | 1 | 305M | **1.56** | 273.2 | 0.81 | 0.60 |

Table 2: Quantitative results on class-conditional ImageNet-256 with MaskGit teacher. † We reproduced Halton Sampler on ImageNet-256 with MaskGit.

| | Method | Steps (↓) | FID (↓) | IS (↑) | Prec. (↑) | Rec. (↑) | Den. (↑) | Cov. (↑) |
|---|---|---|---|---|---|---|---|---|
| Teacher | MaskGit (Besnier & Chen, 2023) | 16 | 6.60 | 224.1 | 0.83 | 0.40 | 1.25 | 0.98 |
| | MaskGit (Besnier & Chen, 2023) | 8 | 6.66 | 221.6 | 0.83 | 0.40 | 1.23 | 0.97 |
| | MaskGit (Besnier & Chen, 2023) | 4 | 10.73 | 192.3 | 0.75 | 0.31 | 1.01 | 0.92 |
| | MaskGit (Besnier & Chen, 2023) | 2 | 91.35 | 13.4 | 0.18 | 0.16 | 0.09 | 0.12 |
| Sampler | $\theta$-trapezoidal (Ren et al., 2025) | 64 | 6.70 | - | - | - | - | - |
| | $\theta$-trapezoidal (Ren et al., 2025) | 32 | 7.10 | - | - | - | - | - |
| | Halton† (Besnier et al., 2025) | 32 | 9.62 | 296.2 | 0.90 | 0.28 | 1.47 | 0.98 |
| | Halton† (Besnier et al., 2025) | 16 | 8.73 | 283.2 | 0.89 | 0.28 | 1.46 | 0.98 |
| | Halton† (Besnier et al., 2025) | 8 | 7.73 | 248.8 | 0.86 | 0.31 | 1.37 | 0.97 |
| Discrete Distillation | SDTT (Deschenaux & Gulcehre, 2024) | 4 | 8.97 | 205.0 | 0.88 | 0.41 | 1.43 | 0.97 |
| | SDTT (Deschenaux & Gulcehre, 2024) | 1 | 90.40 | 14.0 | 0.31 | 0.13 | 0.21 | 0.34 |
| | di4c (Hayakawa et al., 2024) | 4 | 6.79 | 209.2 | - | - | - | - |
| | di4c-d (Hayakawa et al., 2024) | 4 | 6.57 | 213.6 | - | - | - | - |
| | ReDi[1] (Yoo et al., 2025) | 4 | 7.58 | 228.0 | 0.87 | 0.46 | 1.33 | 0.98 |
| | ReDi[2] (Yoo et al., 2025) | 4 | 7.86 | 240.0 | 0.87 | 0.44 | 1.31 | 0.97 |
| | ReDi[3]-distill (Yoo et al., 2025) | 1 | 11.68 | 182.0 | 0.83 | 0.44 | 1.25 | 0.96 |
| | Di[M]O (Zhu et al., 2025b) | 1 | 6.91 | 214.0 | 0.83 | 0.38 | 1.26 | 0.97 |
| | Soft-Di[M]O | 1 | **6.40** | 214.8 | 0.83 | 0.39 | 1.27 | 0.97 |

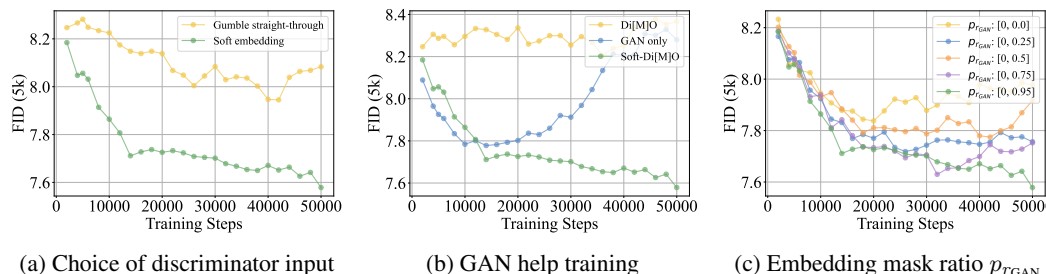

(a) Choice of discriminator input     (b) GAN help training     (c) Embedding mask ratio $p_{r_{GAN}}$

Figure 4: Ablation studies on ImageNet-256 using FID as the evaluation metric with MaskBit teacher.

tribution. We simply use reward loss to fine-tuning the one-step Meissonic generator from Di[M]O. We observe in Tab. 3 that this straightforward reward-only fine-tuning produces consistent improvements in both HPS and GenEval scores.

**Result with MaskGen teacher.** MaskGen is a strong MDM that uses a semantic 1D tokenizer (Yu et al., 2024b) and is trained at $256 \times 256$ resolution. Although the provided MaskGen-L and MaskGen-XL differ in size, their performance is similar in our experiments, so we use MaskGen-L as the teacher to train our one-step generator. We reimplemented Di[M]O on the MaskGen-L teacher to provide an initialization for reward fine-tuning. We observe that reward-only fine-tuning can induce reward hacking, producing over-saturated images (see Appendix I.10); this motivates keeping the Di[M]O loss as a regularization to limit large divergence from the teacher distribution. By combining the reward loss with the Di[M]O loss we obtain higher GenEval and HPS scores than the teacher when sampling with 32 steps (see Tab. 3 and more comparison with multi-step teacher in Appendix I.12). Visual comparisons appear in Fig. 3.

**TTS with MaskGen teacher.** After the rewards-fine-tuned one-step generator is obtained, we apply TTEO with it. We use the SGD (Ruder, 2016) optimizer with a learning rate of 0.2 and no gradient clipping; other settings follow Eyring et al. (2024), e.g., we use the same combination of 4 rewards (CLIP score, ImageReward, HPS, and PickScore) as selection metric to pick the final samples . Our Soft-Di[M]O-MaskGen-L model further improves both GenEval and HPS V2.1 scores and substantially outperforms the multi-step teacher (see Tab. 3). Example outputs and qualitative comparisons can be found in Fig. 3. We provide corresponding ablation and cost-vs-quality analysis in Appendix I.8.

Table 3: Comparison of text-to-image generation methods across multiple metrics. Methods marked with † are evaluated by running their publicly available checkpoints; all other numbers are taken from the corresponding papers as reported. ↑ denotes the higher is better and ↓ denotes the lower is better.

| Methods | Steps (↓) | #Params | FID↓ | CLIP↑ | GenEval ↑ | | | | | | | HPS V2.1 ↑ | | | | |
|---|---|---|---|---|---|---|---|---|---|---|---|---|---|---|---|---|
| | | | | | Single | Two | Counting | Colors | Position | Color Attr. | Overall | Anim. | Concept | Painting | Photo | Averaged |
| **Pre-train Models** | | | | | | | | | | | | | | | | |
| LDM (Rombach et al., 2022) | 50 | 1.4B | 12.64 | - | 0.92 | 0.29 | 0.23 | 0.70 | 0.02 | 0.05 | 0.37 | 20.63 | 19.65 | 19.79 | 21.26 | 20.34 |
| DALLE 2 (Ramesh et al., 2022) | - | 4.2B | 10.39 | - | 0.94 | 0.66 | 0.49 | 0.77 | 0.10 | 0.19 | 0.52 | 26.38 | 24.51 | 24.93 | 25.55 | 25.34 |
| SDXL (Podell et al., 2023) | 50 | 2.6B | 6.63 | 0.290 | 0.98 | 0.74 | 0.39 | 0.85 | 0.15 | 0.23 | 0.55 | 32.84 | 31.36 | 30.86 | 27.48 | 30.63 |
| Meissonic (Kim et al., 2025) | 32 | 1.0B | 50.13 | 0.318 | 0.92 | 0.53 | 0.33 | 0.80 | 0.08 | 0.13 | 0.46 | 31.67 | 27.27 | 29.67 | 29.93 | 29.63 |
| MaskGen-L (Kim et al., 2025) | 16 | 0.6B | 22.64 | 0.312 | 0.97 | 0.55 | 0.38 | 0.80 | 0.08 | 0.14 | 0.48 | 29.09 | 27.54 | 27.32 | 25.87 | 27.60 |
| **Continuous Distillation** | | | | | | | | | | | | | | | | |
| InstaFlow (Liu et al., 2023b) | 1 | 0.9B | 13.10 | - | 0.88 | 0.21 | 0.20 | 0.66 | 0.03 | 0.03 | 0.33 | 21.25 | 21.12 | 21.41 | 20.92 | 21.18 |
| SiD-LSG† (Zhou et al., 2024a) | 1 | 0.9B | 8.15 | 0.304 | 0.93 | 0.37 | 0.21 | 0.57 | 0.03 | 0.03 | 0.36 | 23.08 | 21.37 | 22.55 | 21.53 | 22.13 |
| RG-LCM (HPS)† (Li et al., 2024b) | 2 | 0.9B | 24.04 | - | 0.97 | 0.54 | 0.35 | 0.82 | 0.07 | 0.14 | 0.48 | 30.85 | 33.66 | 33.35 | 33.66 | 32.88 |
| TDM† (Luo et al., 2025c) | 4 | 0.9B | 20.44 | - | 0.99 | 0.57 | 0.49 | 0.78 | 0.09 | 0.09 | 0.50 | 32.91 | 31.73 | 32.18 | 29.95 | 31.37 |
| SDXL-LCM† (Luo et al., 2023a) | 1 | 2.6B | 72.50 | 0.286 | 0.75 | 0.11 | 0.14 | 0.59 | 0.01 | 0.03 | 0.27 | 18.48 | 19.57 | 17.88 | 18.95 | 18.72 |
| SDXL-LCM† (Luo et al., 2023a) | 4 | 2.6B | 17.83 | 0.327 | 0.99 | 0.57 | 0.39 | 0.86 | 0.09 | 0.18 | 0.51 | 27.17 | 29.02 | 25.52 | 27.26 | 27.24 |
| SDXL-Turbo† (Sauer et al., 2024b) | 1 | 2.6B | 19.40 | 0.342 | 0.99 | 0.65 | 0.52 | 0.87 | 0.12 | 0.19 | 0.55 | 28.67 | 30.91 | 28.45 | 26.62 | 28.66 |
| SDXL-DMD2† (Yin et al., 2024a) | 1 | 2.6B | 14.49 | 0.343 | 0.99 | 0.68 | 0.64 | 0.90 | 0.08 | 0.19 | 0.55 | 28.29 | 30.36 | 27.19 | 27.92 | 28.44 |
| **Discrete Distillation** | | | | | | | | | | | | | | | | |
| Di[M]O-Meissonic (Zhu et al., 2025b) | 1 | 1.0B | 38.45 | 0.322 | 0.91 | 0.53 | 0.22 | 0.75 | 0.07 | 0.11 | 0.43 | 27.29 | 28.34 | 28.25 | 30.47 | 28.59 |
| Di[M]O-MaskGen-L (Zhu et al., 2025b) | 1 | 0.6B | 24.15 | 0.299 | 0.93 | 0.39 | 0.35 | 0.74 | 0.07 | 0.08 | 0.42 | 27.30 | 28.83 | 27.06 | 25.38 | 27.14 |
| **Soft-Di[M]O-Meissonic** | 1 | 1.0B | 28.33 | 0.319 | 0.98 | 0.75 | 0.39 | 0.83 | 0.10 | 0.14 | 0.53 | 30.45 | 32.80 | 32.54 | 33.63 | 32.35 |
| **Soft-Di[M]O-Maskgen-L** | 1 | 0.6B | 23.43 | 0.321 | 0.98 | 0.59 | 0.41 | 0.81 | 0.09 | 0.18 | 0.51 | 29.51 | 30.62 | 29.34 | 28.06 | 29.38 |
| + TTS | 1 | 0.6B | - | - | 0.99 | 0.78 | 0.68 | 0.83 | 0.14 | 0.33 | 0.63 | 30.01 | 32.33 | 32.20 | 33.25 | 31.95 |

## 4.4 ABLATIONS STUDY

In this section, we present ablation studies to validate several key design choice in our method. More ablations can be found in Appendices I.3, I.7, I.8, I.10 and I.11.

**Choice of embedding representation.** Because the distilled one-step generator yields highly concentrated logits, using a straight-through Gumbel-softmax estimator is not required and can introduce bias or high-variance errors in gradient estimation during training. As shown in Fig. 4a, a comparison between the Gumbel hard straight-through estimator and our soft-embedding method indicates that the soft embedding consistently improves FID throughout training.

**Limitations of GAN-only distillation.** Prior work suggests that GAN training alone can convert a multi-step continuous diffusion model into a one-step generator (Xu et al., 2024; Lin et al., 2025; Sauer et al., 2024b;a; Chen et al., 2025a), but those approaches rely on sophisticated engineering tricks to work in practice. In our experiments, we first show that continued training with the Di[M]O loss does not improve results (see Fig. 4b); conversely, using only the GAN loss (i.e., omitting the Di[M]O objective) fails to produce a competitive generator and can even perform worse than the Di[M]O baseline (see Fig. 4b and Fig. 10). Moreover, when distilling from the MaskGen-L teacher with a large initial mask ratio (e.g., 0.95 as in our MaskGen setup), GAN-only distillation exhibits severe mode collapse: different prompts often yield identical outputs (see Fig. 13). Together, these findings indicate that Di[M]O loss as a regularizer is crucial for our refinement algorithm.

**Choice of GAN mask schedule.** Analogous to denoising GANs, our mask-embedding GAN objective not only leverages the teacher backbone's training input distribution but also acts as a form of data augmentation for the discriminator, helping to prevent discriminator overfitting. In Fig. 4c we evaluate several mask ranges, from no masking up to a range of [0,0.95]. The results show that using a GAN mask improves training, and that larger mask ranges yield the largest gains.

## 5 CONCLUSION

In this work, we advance visual MDM distillation by integrate additional post-training techniques to the Di[M]O framework. Our approach tackles key challenges of discontinuity by proposing *soft embedding*. Through theoretical justification and extensive experiments across multiple tasks, including class-to-image generation and text-to-image generation, we show that with diverse pretrained models Soft-Di[M]O consistently surpasses both the base teacher MDMs and other distilled variants. This work demonstrates the power of distribution matching methods on distillation for MDMs, contributes to the growing community of research exploring efficient generation of discrete data.

ACKNOWLEDGEMENTS

This work was supported by Hi! PARIS and ANR/France 2030 program (ANR-23-IACL-0005), ANR-22-CE23-0007, ANR-22-CE39-0016, Hi!Paris Grant and Chair, and was granted access to the IDRIS High-Performance Computing (HPC) resources under the allocation 2025-AD011014300R2 and 2025-AD011015894 made by GENCI. We sincerely thank Shilin Lu, Yuqing Wang and Yao Teng for their insightful discussions that contributed to this work. We are also grateful to Runlong Liao, Nizar Benbouchta, and Nicolas Dufour for their meticulous proofreading.

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

# Appendix for Soft-Di[M]O

This appendix is organized as follows:

## A  USE OF LARGE LANGUAGE MODELS

We used large language models solely for text polishing and grammar correction during manuscript preparation. No LLMs were involved in the conception or design of the method, experiments, or analysis. All technical content, results, and conclusions have been independently verified and validated by the authors.

## B  REPRODUCIBILITY STATEMENT

We provide detailed training and evaluation protocols, including a full hyperparameter table, model and tokenizer specifications in the appendix. To enable reproduction and follow-up work, we will publicly release the codes and model checkpoints (together with implementation notes and licensing information) upon publication.

## C  BROADER IMPACTS

Our work focuses on distilling the multi-step generation process of MDMs into one step, significantly reducing inference time and computational costs, therefore lowering the carbon footprint during the inference. This advancement has the potential to make high-quality generative models more accessible, facilitating applications in creative industries, content generation, and real-time systems. However, as with many generative modeling techniques, our method inherits biases from the teacher models. This could potentially lead to ethical concerns, including the generation of misleading or harmful content. Additionally, by enabling faster and more efficient content generation, our approach could lower the barrier to misuse, such as the creation of deepfakes or other deceptive media.

## D  LIMITATION AND FUTURE WORKS

**Method Limitations.** While our distillation method is not limited by the teacher model, the teacher's performance still plays a big role in the distillation, as shown in our text-to-image experiments. A stronger text-to-image teacher model can push the limit of efficient discrete visual generation. In addition, because our GAN operates in the latent space, the student's performance is bounded by the tokenizer's reconstruction FID (rFID). Notably, discrete tokenizers typically yield much higher rFID than continuous ones (Li et al., 2024c). In the future, we expect to apply our

method to MDM trained with the adapted continuous tokenizer (Wu et al., 2025; Liu et al., 2025b; Wang et al., 2025c), such as TokenBridge (Wang et al., 2025c) to reach better generation performance. Besides, our model inherits the weakness of the teacher models, For example, like most MDM for image generation, our model can not generate images at varying resolutions.

**Potential for Other Distillation Methods.** Using soft embeddings not only enables gradient flow into the student's parameters for GAN training, it also makes a SiD-style distillation for MDMs feasible: both the teacher and auxiliary model can accept embedding inputs, allowing gradients to be backpropagated to update the student. Since Di[M]O is currently the only publicly available one-step discrete visual generative model (the ReDi model (Yoo et al., 2025) is not available), we leave fine-tuning one-step generators distilled with alternative distillation methods to future work.

**Distillation for Multi-modal Discrete Model.** While this work focuses on class-to-image and text-to-image generation, we believe that the proposed algorithm can also be applied to the visual generation of multi-modal discrete diffusion models (Hu et al., 2022; Li et al., 2024d; Xie et al., 2024a; Wang et al., 2024a; Yang et al., 2025; You et al., 2025; Yu et al., 2025; Li et al., 2025; Wang et al., 2025a; Shi et al., 2025b), such as show-o (Xie et al., 2024a) and Fudoki (Wang et al., 2025a). Due to limited computational resources, we leave this direction to future work.

**Discrete Diffusion beyond MDMs.** For discrete diffusion schemes that employ a uniform (or uniform-like) kernel (Austin et al., 2021; Gat et al., 2024; Shaul et al., 2024), our method naturally extends and avoids the out-of-distribution issues that can arise from the one-step generator's initialization. This stems from the fact that our distributional-matching objective (including the original Di[M]O loss) targets the generator's *outputs* regardless of its initialization distribution. We also acknowledge the related approach of Yoo et al. (2025), which first fine-tunes a masked diffusion model into a uniform-kernel discrete diffusion model and then applies ReFlow (Liu, 2022a).

## E RELATED WORK

**Continuous Diffusion Distillation.** Continuous diffusion models learn a Probability Flow ODE (PF-ODE) that connects the initial distribution to the target distribution. Sampling requires numerically solving this PF-ODE, often with many iterative steps. To accelerate generation, various distillation techniques have been proposed, which can be broadly grouped into *ODE-based* and *distribution-based* methods. *ODE-based methods* (Luhman & Luhman, 2021; Song et al., 2023; Liu, 2022a; Salimans & Ho, 2022; Gu et al., 2023; Meng et al., 2023; Yan et al., 2024; Zhu et al., 2025a) train a student model through regression objectives derived from the teacher's PF-ODE trajectory. However, these approaches are difficult to apply to masked diffusion models (MDMs), which lack an explicit PF-ODE formulation. *Distribution-based methods* (Luo et al., 2023b; Yin et al., 2024b;a; Xu et al., 2024; Kim et al., 2023; Zhou et al., 2024c;a;b; Nguyen & Tran, 2024; Dao et al., 2025; Luo et al., 2025a; Zhou et al., 2025b) instead directly align the student's output distribution with the teacher's multi-step sampling distribution, typically via divergence minimization or adversarial training (Sauer et al., 2024b;a). Variational Score Distillation (VSD) (Wang et al., 2024b; Yin et al., 2024b;a; Nguyen & Tran, 2024; Dao et al., 2025; Luo et al., 2023b; Xie et al., 2024b; Salimans et al., 2025) follows this paradigm by introducing an auxiliary model $\psi$ that approximates intermediate distributions of a one-step generator, serving as a proxy to match the student and teacher.

**Acceleration of Masked Diffusion Models.** MDMs achieve strong generation performance, but their iterative nature makes inference both time- and compute-intensive. Consequently, improving MDM inference efficiency has become an active research topic. Prior work has approached this from several angles. Some methods focus on optimizing the sampling process: Park et al. (2024) propose optimizing sampling schedules to reduce compounding decoding errors, while Ren et al. (2025) employ higher-order solvers to allow larger sampling steps. Halton sampling (Besnier et al., 2025) and entropy-bounded sampling (Ben-Hamu et al., 2025) have been proposed to accelerate generation by sampling multiple relatively independent tokens, leveraging spatial sparsity or entropy constraints. Other approaches explore distillation: Hayakawa et al. (2024) adopt a strategy similar to Progressive Distillation (Salimans & Ho, 2022), training a few-step student with a consistency loss that leverages intermediate states from real data, and Deschenaux & Gulcehre (2024) distill a student to approximate the teacher's multi-step output distribution by minimizing divergence, which requires multiple teacher inferences and often multi-round distillation. Recently, (Zhu et al., 2025b) intro-

duced Di⟦M⟧O, the first framework enabling effective one-step generation for MDMs, significantly simplifying and accelerating inference. Finally, Yoo et al. (2025) propose fine-tuning an MDM with stochastic initial states, followed by a reflow stage with the simulated couplings (Liu, 2022a;b; Liu et al., 2023b). This idea of stochastic initialization is conceptually similar to that used in Di⟦M⟧O.

**Adversarial Training for Discrete Modelling.** Adversarial training for discrete data is challenging because the discriminator must compare generated sequences with ground truth, while the sampling process introduces non-differentiability. Several strategies have been proposed to address this. A common approach is the REINFORCE estimator (Williams, 1992; Sutton et al., 1999; Ranganath et al., 2014), which treats generation as a policy and the discriminator score as a reward. It has been widely applied in discrete sequence modeling (Fedus et al., 2018; Rennie et al., 2017; Yu et al., 2017; Che et al., 2017; Zhu et al., 2025c), but suffers from high-variance gradients that often destabilize training. Another line of work adopts differentiable relaxations of discrete tokens. The Gumbel-Softmax trick (Gumbel, 1954; Jang et al., 2016; Gu et al., 2018) generates a soft one-hot approximation: $p_{\text{soft}}^i = \text{softmax}\left((z_\theta^i + g^i)/(\tau)\right), \quad g_i \sim \text{Gumbel}(0, 1)$, where $\tau > 0$ controls the smoothness. This method has been successfully applied in adversarial training for discrete data (Kusner & Hernández-Lobato, 2016; Guo et al., 2021; Wang, 2025; Tang et al., 2025), but suffers from biased gradients and potential numerical instabilities (Jang et al., 2016; Liu et al., 2023a). Similarly, the soft-argmax operator (Zhang et al., 2016; 2017) has been used to compute the soft embedding for LSTM-based generators (Hochreiter & Schmidhuber, 1997), and recently in discrete diffusion models (Stark et al., 2024; Liu, 2025; Sahoo et al., 2025). However, no prior work has explored using soft embeddings for fast visual discrete generation.

# F  DISCUSSION ON DIFFUSION LARGE LANGUAGE MODELS (DLLM)

Most Diffusion Large Language Models (DLLMs) are instances of MDMs. Here, we discuss the challenges of directly applying Di⟦M⟧O to language modelling for one-step generation with MDMs as an example. At first glance, one might expect that the only difference between image-based and text-based MDMs lies in the underlying data modality. However, this distinction turns out to be fundamental for distillation. **Visual Signals: Pseudo-Discrete with Redundancy.** Visual signals are inherently continuous and highly redundant—human perception remains unchanged even if many pixels are perturbed. This redundancy persists after discretization with tokenizers such as VQGAN. Even if some visual tokens are distorted, the decoded image is often perceptually similar to the original. Thus, visual discrete codes can be regarded as pseudo-discrete: although tokenized, they retain an underlying continuity. **Language Tokens: Intrinsically Discrete.** Language tokens, in contrast, are intrinsically discrete. Every token choice has precise semantic and syntactic consequences, and small perturbations typically make a sentence ungrammatical or nonsensical.

This difference creates a fundamental challenge for MDM one-step generation. For existing discrete diffusion modelling, a one-step generator produces all logits for each token, then samples each of the tokens in parallel *__independently__*. To sample a sentence correctly, the model's joint output distribution must collapse into an almost exact delta distribution over one sequence, which means the entropy for each prediction position must be as low as possible, otherwise the probability of generating a valid sentence is essentially zero (Feng et al., 2025). This requirement for near-perfect precision in the output distribution makes DLLM distillation significantly more challenging than visual diffusion model distillation.

# G  THEORETICAL JUSTIFICATION OF SOFT EMBEDDING

This section provides a principled justification for using Soft Embedding as the surrogate for reward fine-tuning or adversarial training in one-step discrete generators. We establish (i) the ideal REINFORCE gradient as a reference, (ii) the bias and variance analysis of soft embedding and Gumbel-Softmax; and (iii) why soft embedding is particularly suitable for one-step generators whose logits are typically highly concentrated.

### G.1 PROBLEM SETUP

Consider the following ideal discrete objective[1]:

$$\mathcal{L}_{\text{orig}}(\theta) = \mathbb{E}_{j \sim p_\theta}[f(e_j)], \tag{12}$$

where $p_\theta$ is the predicted token distribution, $e_j$ is the embedding of token $j$, $f$ denotes a deterministic differentiable mapping (e.g., teacher backbone + discriminator heads in adversarial training, or tokenizer decoder + reward function in reward fine-tuning).

The difficulty arises because sampling $j \sim p_\theta$ is non-differentiable. Common surrogates include:

1. REINFORCE, unbiased loss estimator but with high gradient variance.
2. Gumbel-Softmax (ST), biased and noisy.
3. Soft Embedding, a smooth relaxation used in our method.

**REINFORCE (ideal reference).** Using the likelihood-ratio identity:

$$\nabla_\theta \mathcal{L}_{\text{orig}}(\theta) = \mathbb{E}_{j \sim p_\theta}[f(e_j)\nabla_\theta \log p_\theta]. \tag{13}$$

This estimator is unbiased, but for high-dimensional outputs (hundreds or thousands of tokens), the loss gradient variance is prohibitive.

**Gumbel-Softmax (Straight-Through).** Gumbel-Softmax defines a soft relaxation:

$$\hat{p}_\theta = \text{softmax}\left((z_\theta + g)/(\tau)\right). \tag{14}$$

And the loss $\mathcal{L}_{\text{hard}} = \mathbb{E}[f(\hat{e}_{\text{hard}})]$ uses a *hard sample* $\hat{j} = \arg\max(\hat{p}_\theta)$ in the forward pass and a *soft sample* in the backward pass to calculate the approximation $\frac{d\hat{e}_{\text{soft}}}{d\theta} \approx \frac{d\hat{e}_{\text{hard}}}{d\theta}$:

$$\hat{e}_{\text{hard}} = \hat{e}_{\arg\max(\hat{p}_\theta)} \quad \text{(forward)}, \quad \hat{e}_{\text{soft}} = E^T \hat{p}_\theta \quad \text{(backward)}. \tag{15}$$

**Soft Embedding (ours).** Soft embedding replaces the discrete token by its embedding expectation:

$$\tilde{e} = \mathbb{E}_{j \sim p_\theta}[e_j] = E^T p_\theta. \tag{16}$$

The surrogate objective becomes $\mathcal{L}_{\text{soft}}(\theta) = f(\tilde{e})$. This yields a fully pathwise gradient:

$$\nabla_\theta \mathcal{L}_{\text{soft}}(\theta) = \nabla_{\tilde{e}} f(\tilde{e}) E^T (\text{diag}(p_\theta) - p_\theta^T p_\theta) \frac{dz_\theta}{d\theta}. \tag{17}$$

### G.2 BIAS ANALYSIS OF SOFT EMBEDDING AND GUMBEL-SOFTMAX

In this subsection, we provide a formal analysis of the bias introduced by Soft Embedding and Gumbel-Softmax when used as relaxations of the original objective in Eq. (12). We work under standard smoothness assumptions and show that (i) the Soft-Embedding surrogate introduces only a *second-order* bias in the embedding variance, while (ii) Gumbel-Softmax straight-through produces a *first-order* biased estimator whose bias does not vanish even under concentrated logits.

#### G.2.1 ASSUMPTIONS

**Assumption G.1 (Smoothness of $f$)** *The function $f : \mathbb{R}^d \to \mathbb{R}$ is twice continuously differentiable with Hessian bounded as*

$$\|\nabla^2 f(e)\| \le L \quad \text{for all } e; . \tag{18}$$

**Assumption G.2 (Embedding variance)** *Let $\mu = \tilde{e}$ the expected embedding, define the embedding variance as:*

$$\Sigma := \mathbb{E}_{j \sim p_\theta}\left[(e_j - \mu)(e_j - \mu)^\top\right]. \tag{19}$$

*We denote its spectral norm by $\|\Sigma\|$.*

---

[1]For simplicity, we consider only loss on one token since they are sampled independently.

### G.2.2 Soft Embedding Bias

We first show that the bias of Soft Embedding is *second-order* in the embedding variance.

**Lemma G.3 (Second-order bias of Soft Embedding)** *Under Assumption G.1, the soft-embedding surrogate satisfies:*

$$|f(\mu) - \mathbb{E}_{j \sim p_\theta}[f(e_j)]| \leq \frac{L}{2} \|\Sigma\|. \tag{20}$$

*Thus the soft-embedding objective $f(\mu)$ matches the true discrete objective up to a second-order term in $\|\Sigma\|$. Furthermore, if the predicted distribution is concentrated on a mode $j^*$ with*

$$p_\theta(j^*) = 1 - \varepsilon, \qquad \sum_{j \neq j^*} p_\theta(j) = \varepsilon, \tag{21}$$

*and if all embeddings satisfy $\|e_j\|_2 \leq B$, then $\|\Sigma\| = \mathcal{O}(\varepsilon)$, and thus the soft-embedding bias in Eq. (20) is also $\mathcal{O}(\varepsilon)$.*

**Proof:** **(Second-order bias).** Using the second-order Taylor expansion of $f(e_j)$ around $\mu$:

$$f(e_j) = f(\mu) + \nabla f(\mu)^\top (e_j - \mu) + \frac{1}{2}(e_j - \mu)^\top \nabla^2 f(\xi_j)(e_j - \mu),$$

where $\xi_j$ lies on the line segment between $e_j$ and $\mu$. Taking expectation under $j \sim p_\theta$, the linear term vanishes since $\mathbb{E}[e_j - \mu] = 0$. Therefore,

$$\mathbb{E}[f(e_j)] = f(\mu) + \frac{1}{2}\mathbb{E}\left[(e_j - \mu)^\top \nabla^2 f(\xi_j)(e_j - \mu)\right].$$

By Assumption G.1,

$$\left|(e_j - \mu)^\top \nabla^2 f(\xi_j)(e_j - \mu)\right| \leq L\|e_j - \mu\|^2.$$

Taking expectation:

$$|\mathbb{E}[f(e_j)] - f(\mu)| \leq \frac{L}{2}\mathbb{E}\|e_j - \mu\|^2 = \frac{L}{2}\|\Sigma\|.$$

**(Covariance under concentrated logits).** Assume $p_\theta$ satisfies Eq. (21) and $\|e_j\|_2 \leq B$. Define the weighted average of non-max tokens

$$m := \frac{1}{\varepsilon} \sum_{j \neq j^*} p_\theta(j) e_j.$$

Then: (i) $\|\mu\|_2 \leq B$ and $\|m\|_2 \leq B$ (ii) $\|e_j - \mu\|_2 \leq 2B$ for all $j$ (iii) since $\mu = (1 - \varepsilon)e_{j^*} + \varepsilon m$,

$$\|e_{j^*} - \mu\|_2 = \varepsilon\|e_{j^*} - m\|_2 \leq 2\varepsilon B.$$

Now expand the covariance:

$$\Sigma = (1 - \varepsilon)(e_{j^*} - \mu)(e_{j^*} - \mu)^\top + \sum_{j \neq j^*} p_\theta(j)(e_j - \mu)(e_j - \mu)^\top.$$

Using the bounds above,

$$(1 - \varepsilon)\|e_{j^*} - \mu\|_2^2 \leq (1 - \varepsilon)(2B)^2 \varepsilon^2,$$

and

$$\sum_{j \neq j^*} p_\theta(j)\|e_j - \mu\|_2^2 \leq \varepsilon(2B)^2.$$

Hence,

$$\|\Sigma\| \leq (1 - \varepsilon)(2B)^2 \varepsilon^2 + \varepsilon(2B)^2 = \mathcal{O}(\varepsilon).$$

Substituting this into Eq. (20) proves that the soft-embedding bias is also $\mathcal{O}(\varepsilon)$. $\qquad\square$

**Remarks.** Lemma G.3 shows that the soft-embedding surrogate is intrinsically accurate when the model places most of its probability mass on a small number of candidate tokens. In the concentrated-logit regime (common for one-step generators distilled from diffusion models), the embedding covariance scales as $\|\Sigma\| = \mathcal{O}(\varepsilon)$, and therefore the bias between $f(\mu)$ and the true discrete objective becomes negligible. This explains why soft embeddings provide a faithful continuous approximation of discrete sampling in practice.

### G.2.3 BIAS OF GUMBEL-SOFTMAX STRAIGHT-THROUGH

We now analyze the bias of the Gumbel-Softmax Straight-Through (ST) estimator. Unlike soft embedding, Gumbel-Softmax ST exhibits *first-order* bias that persists even under concentrated logits.

**Lemma G.4 (First-order bias of Gumbel-Softmax ST)** *Suppose Assumption G.1 holds, and in addition $\nabla f$ is $L_\nabla$-Lipschitz on the convex hull of the embeddings. Let $\hat{e}_{\mathrm{hard}}(g)$ and $\hat{e}_{\mathrm{soft}}(g)$ denote the hard and soft embeddings produced by Gumbel-Softmax ST for a given Gumbel noise $g$. Then the bias of the ST gradient estimator satisfies*

$$\left\| \mathbb{E}_g[\nabla_\theta \mathcal{L}_{\mathrm{ST}}] - \nabla_\theta \mathcal{L}_{\mathrm{orig}} \right\| \leq C \cdot \mathbb{E}_g\left[\|\hat{e}_{\mathrm{hard}}(g) - \hat{e}_{\mathrm{soft}}(g)\|\right] + \mathcal{O}(\tau) + \mathcal{O}(\|\Sigma_\theta\|), \qquad (22)$$

*where $C > 0$ is a constant depending on $L_\nabla$ and $\sup_\theta \|\partial \hat{e}_{\mathrm{soft}}/\partial\theta\|$, $\tau$ is the Gumbel-Softmax temperature, and $\Sigma_\theta$ denotes the embedding covariance under $p_\theta$. Moreover, whenever $p_\theta(j^*) < 1$ for the mode $j^*$ and $\|\nabla f(e_{j^*})\| \geq c_0 > 0$, the bias term in Eq. (22) does not vanish as $\tau \to 0$.*

**Proof:** [Proof sketch] For a fixed Gumbel noise $g$, the ST gradient can be written as

$$g_{\mathrm{ST}}(\theta; g) = J_f(\hat{e}_{\mathrm{hard}}(g)) \cdot \frac{\partial \hat{e}_{\mathrm{soft}}(g)}{\partial \theta},$$

where $J_f$ is the Jacobian of $f$ with respect to its input. Consider the "fully soft" reparameterized gradient

$$g_{\mathrm{soft}}(\theta; g) = J_f(\hat{e}_{\mathrm{soft}}(g)) \cdot \frac{\partial \hat{e}_{\mathrm{soft}}(g)}{\partial \theta}.$$

Using the Lipschitz continuity of $\nabla f$ and the chain rule, we obtain

$$\left\| \mathbb{E}_g[g_{\mathrm{ST}}(\theta; g)] - \mathbb{E}_g[g_{\mathrm{soft}}(\theta; g)] \right\| \leq C \cdot \mathbb{E}_g\left[\|\hat{e}_{\mathrm{hard}}(g) - \hat{e}_{\mathrm{soft}}(g)\|\right].$$

On the other hand, $\mathbb{E}_g[g_{\mathrm{soft}}(\theta; g)]$ is the reparameterized gradient of a continuous relaxation objective, whose bias with respect to the discrete objective $\mathcal{L}_{\mathrm{orig}}$ is $\mathcal{O}(\tau) + \mathcal{O}(\|\Sigma_\theta\|)$ under Assumption G.1. Combining the two bounds yields Eq. (22).

Finally, in the concentrated regime where $p_\theta(j^*) \approx 1$ but $p_\theta(j^*) < 1$, the hard sample equals $e_{j^*}$ with probability $p_\theta(j^*)$, whereas the soft embedding stays strictly inside the convex hull. Consequently,

$$\mathbb{E}_g\left[\|\hat{e}_{\mathrm{hard}}(g) - \hat{e}_{\mathrm{soft}}(g)\|\right] \gtrsim (1 - p_\theta(j^*)) \cdot \mathbb{E}_{j \neq j^*}[\|e_{j^*} - e_j\|] > 0,$$

and if $\|\nabla f(e_{j^*})\| \geq c_0 > 0$ this implies a non-vanishing bias as $\tau \to 0$. $\qquad \square$

**Remarks.** The key distinction is that Gumbel-Softmax ST incurs a *first-order* bias term proportional to the **mismatch between forward and backward embeddings**, $\|\hat{e}_{\mathrm{hard}} - \hat{e}_{\mathrm{soft}}\|$. This bias: (i) does not vanish under temperature annealing when $p_\theta(j^*) < 1$; (ii) scales with $\|\nabla f\|$; and (iii) persists even in the concentrated logit regime. In addition, the soft sample $\hat{e}_{\mathrm{soft}} = E^\top \hat{p}_\theta$ is a random variable whose expectation $\mathbb{E}_g[\hat{p}_\theta] \neq p_\theta$ for finite $\tau$, creating additional bias due to **Gumbel noise**.

### G.3 VARIANCE COMPARISON

In addition to the bias gap, the two methods exhibit fundamentally different *variance* behavior. Soft embedding produces a fully deterministic forward pass and a pathwise gradient $\nabla_\theta f(E^\top p_\theta)$, whose variance is identically zero given $x_{\mathrm{init}}$. In contrast, Gumbel–Softmax introduces stochastic Gumbel noise in every forward pass and further mismatches the backward pass (soft) and forward pass (hard), yielding high-variance gradients of the form $\nabla_\theta f(e_{\hat{j}})$ where $\hat{j}$ depends on random perturbations.

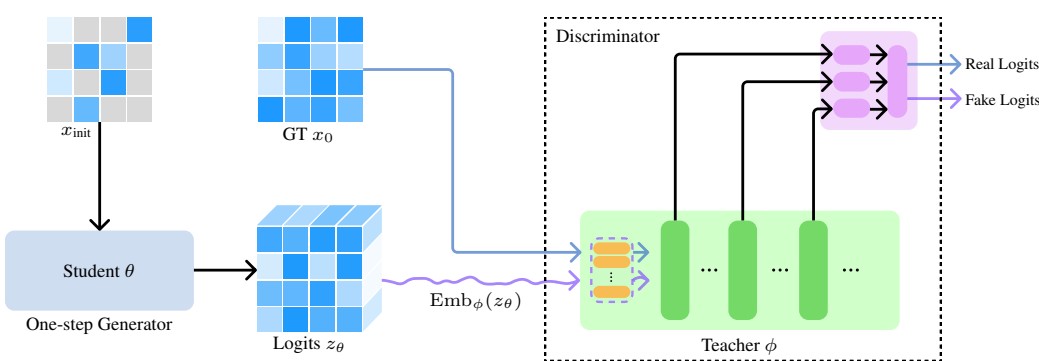

Figure 5: **Soft-Di[M]O Zoom in details**: the multi-scale discriminator in the pipeline Fig. 2.

## G.4 SUMMARY

Tab. 4 summarizes the comparison. Soft-embedding yields (i) a principled and low-variance surrogate objective, (ii) a bias that is provably second-order in embedding variance, and (iii) excellent stability in the highly concentrated logit regime characteristic of one-step generators. In contrast, Gumbel-Softmax straight-through combines significant bias with high variance, making it unstable for reward- or GAN-based fine-tuning.

Table 4: Comparison of gradient estimators for optimizing discrete generators.

| Method | Bias | Variance |
|---|---|---|
| REINFORCE | None (unbiased) | Very high |
| Gumbel–Softmax ST | High | High |
| Soft Embedding (ours) | Small (2nd-order) | Very low |

## H MORE EXPERIMENTAL DETAILS

In this section, we provide further implementation and training details to facilitate reproducibility. The hyperparameters customized in this work and the model configurations are listed in Tab. 5. The complete algorithm with the input output dimension is presented in Algorithm 1.

### H.1 IMPLEMENTATION

Soft-Di[M]O is implemented on top of the Di[M]O codebase. For MaskGit teacher models, we use the PyTorch implementation from (Besnier & Chen, 2023), which relies on the original VQGAN (Esser et al., 2021); see discussion in (Sun et al., 2024). All teacher sampling procedures follow the corresponding official implementations.

Unless otherwise specified, we use the following defaults:

- Gradient clipping with norm 1.
- Constant learning rate scheduler with a linear warmup of 100 steps.
- All model temperatures are fixed to 1 during distillation (no top-p or top-k sampling).
- The generator embedding layer is *frozen* during distillation to improve stability.
- Loss weight $w(t) = 1$ is used in Eq. (2) for simplicity.

### H.2 DISTILLATION AND TEACHER SCHEDULES

For the Di[M]O loss, we use the same mask schedule both for (i) obtaining the intermediate state $\tilde{x}_t$ using generation with the one-step model and (ii) training the auxiliary model. Regarding the

Table 5: Hyperparameters and configurations used for distillation. Model architectures are the same as the teacher models. Largest Experiments in this work are conducted with 4 NVIDIA H100 GPUs. In the *Output Logits Dimension*, $B$ is the batch size, $V+1$ in MaskGit means mask token included, $[B, L, 2, 128]$ in MaskBit represents the split group strategy where vocabulary $V = 16384 = 2^{14}$ is split into 2 groups with $2^7$ each.

| Teacher Model | MaskGit | MaskBit (longer training) | Meissonic | MaskGen |
|---|---|---|---|---|
| Batch Size | 256 | 256 | 16 | 64 |
| Training Steps | 30k | 50k(+50k) | 10k | 50k |
| Teacher CFG | 2.0 | 1.5 | 4.0 | 4.0 |
| Initial Mask Ratio | 0.6 | 0.6 | 0.6 | 0.9 |
| Di[M]O Divergence | Jeffreys -0.2 | FKL | FKL | FKL |
| Fix Embedding Layer | True | True | True | True |
| Sampling Temperature | 12 | 0.1 | 0.5 | 1. |
| Generator Mask Schedule | arccos | arccos | cosine | arccos |
| Auxiliary Mask Schedule | arccos | arccos | cosine | arccos |
| GAN Loss Weight | 100 | 25 (200) | - | 100 |
| GAN Mask Schedule | $\mathcal{U}[0, 0.5]$ | $\mathcal{U}[0, 0.95]$ | - | $\mathcal{U}[0, 0.95]$ |
| # Discriminator Heads | 4 | 4 | - | 6 |
| Total #params in Discriminator | 72M | 85M | - | 65M |
| Reward Loss Weight | - | - | 2000 | 2000 |
| CLIP Reward Relative Weight | - | - | 0.2 | 0.2 |
| Mixed Precision | bf16 | bf16 | bf16 | bf16 |
| Optimizer | Adam | Adam (AdamW) | Adam | Adam |
| Generator $(\beta_1, \beta_2)$ | (0.9,0.999) | (0.9,0.999) | (0.9,0.999) | (0.9,0.999) |
| Auxiliary Model $(\beta_1, \beta_2)$ | (0.9,0.999) | (0.9,0.999) | (0.9,0.999) | (0.9,0.999) |
| Discriminator $(\beta_1, \beta_2)$ | (0.0,0.999) | (0.0,0.999) | (0.0,0.999) | (0.0,0.999) |
| Learning Rate | 1e-5 | 1e-6 (5e-7) | 1e-6 | 1e-6 |
| Weight Decay | 0.0 | 0.0 (0.01) | 0.0 | 0.0 |
| EMA Ratio | 0.9999 | 0.9999 | 0.9999 | 0.9999 |
| Generation Image Resolution | 256×256 | 256×256 | 1024×1024 | 256×256 |
| Sequence Length ($L$) | 256 | 256 | 4096 | 128 |
| Downsampling Factor (Patch Size) | 16×16 | 16×16 | 16×16 | 512 (1D) |
| Vocabulary Size ($V = |\mathcal{V}|$) | 1024 | 16384 | 8192 | 8192 |
| Output Logits Dimension | $[B, L, V+1]$ | $[B, L, 2, 128]$ | $[B, L, V]$ | $[B, L, V]$ |

divergence $D_{\text{div}}$ in Di[M]O loss, we follow Di[M]O for MaskGit and Meissonic, and use Forward KL (FKL) divergence for both MaskBit and MaskGen-L experiments. The same masking schedule is also used when training the teacher model. Both teacher models employ Classifier-Free Guidance (CFG) Ho & Salimans (2022). Since our generator distills the CFG output of the teacher models, its inference does not require CFG; this removes the extra sampling cost and improves sampling efficiency.

## H.3 DISCRIMINATOR AND GAN TRAINING

When training the discriminator, we apply the same mask to both ground-truth embeddings and generated (fake) embeddings, and we use the same text prompts for real and fake samples. To balance gradient magnitudes, the GAN loss weight is set to 100 in most experiments so that GAN gradients are comparable to the Di[M]O gradients. For the longer runs, in MaskBit experiments, we increase this weight to 200. For MaskGit experiment, we use a mask ratio $r_{GAN} \in [0, 0.5]$, along with a noise perturbation for the features from the teacher model backbone for stable training, which achieves slight better FID (FID 5k from 11.7 (only $r_{GAN} \in [0, 0.95]$ as in MaskBit experiment) to 11.6). For the longer MaskBit training reported in Tab. 1, we initialize the generator from the previous-stage best checkpoint. In addition, we halve the learning rate to 5e-7, increase the GAN loss weight to 200, and reduce the dropout rate to 0. And we use an additional weight decay of 0.01 with AdamW optimizer (Loshchilov & Hutter, 2017). However, we find that either increasing the GAN loss weight or removing dropout (setting the dropout rate to 0) fails to improve the training of the one-step generator from the previous stage.

---

**Algorithm 1** Soft-Di[M]O Distillation

---

**Require:** Pre-trained teacher model $\phi$, Pre-trained one-step generator $\theta_0$ (optional), condition dataset $\mathcal{D}_c$, ground truth dataset $\mathcal{D}_d$, loss weights $w_{\text{GAN}}$ and $w_{\text{reward}}$
1: $\theta \leftarrow \text{copyWeights}(\phi)$, $\theta \leftarrow \text{copyWeights}(\theta_0)$ (optional), $\psi \leftarrow \text{copyWeights}(\phi)$  // initialize
2: **repeat**
3:     *### Generate logits $z_\theta$ and tokens $x_\theta$*
4:     Sample $x_{\text{init}} \sim p_{\text{init}}$, $c \sim \mathcal{D}_c$  // with strategy in Di[M]O
5:     Get generator logits $z_\theta(x_{\text{init}}, c) \in \mathbb{R}^{B \times h \times w \times |\mathcal{V}|}$
6:     $x_\theta \in \mathbb{R}^{B \times h \times w} \xleftarrow{\text{sample}} p_\theta(x_0|x_{\text{init}}) = \text{softmax}(z_\theta(x_{\text{init}}, c))$
7:     *### Update generator $\theta$*
8:     Sample $t \sim \mathcal{U}[0,1]$, $\tilde{x}_t \sim q_{t|0}(\tilde{x}_t|x_\theta(x_{\text{init}}, c))$  // Forward
9:     Calculate $p_\phi(x_0|\tilde{x}_t, c)$ and $p_\psi(x_0|\tilde{x}_t, c)$
10:     *# calculate Di[M]O loss*
11:     $\mathcal{L}_{\text{Di[M]O}}(\theta) \longleftarrow \mathbb{E}_{x_{\text{init}}, t, x=G_\theta(x_{\text{init}})} \left[ w(t) \left( \mathbb{E}_{q_{t|0}} [\nabla_{z_\psi} D_{\text{div}}(p_\phi || p_\psi)(\tilde{x}_t)] \right) \right]$
12:     Calculate soft embedding $\tilde{e}_\theta = \text{Emb}(z_\theta) = E^\top p_\theta(x_0|x_{\text{init}})$
13:     *# calculate GAN loss*
14:     $\mathcal{L}_{\text{GAN}}(\theta) \longleftarrow \mathbb{E}_{x_{\text{init}} \sim p_{\text{init}}, r \sim p_{r_{\text{GAN}}}} [-\log(D_\eta(\text{Emb}(z_\theta)_r, r))].$
15:     *# calculate reward loss*
16:     $\mathcal{L}_{\text{reward}}(\theta) \longleftarrow -\sum_i \lambda_i \mathcal{R}_i(\text{Dec}(\text{Emb}_{\text{Dec}}(z_\theta), c)$
17:     *# calculate total loss and update*
18:     Update $\theta$ using gradient of $\mathcal{L}_{\text{gen}}(\theta) = \mathcal{L}_{\text{Di[M]O}}(\theta) + w_{\text{GAN}}\mathcal{L}_{\text{GAN}}(\theta) + w_{\text{reward}}\mathcal{L}_{\text{reward}}(\theta)$
19:     *### Update auxiliary model $\psi$*
20:     Sample $t' \sim \mathcal{U}[0,1]$, $\tilde{x}_{t'} \sim q_{t|0}(\tilde{x}_{t'}|x_\theta(x_{\text{init}}, c))$
21:     Update $\psi$ with cross entropy loss (Eq. (1))
22:     *### Update discriminator $\eta$*
23:     Sample $t'' \sim \mathcal{U}[0, 0.95]$, calculate $\tilde{e}_{t''} = \text{Emb}(z_\theta)_{t''}$
24:     Sample real data $(x_{\text{gt}}, c) \sim \mathcal{D}_d$, calculate $\text{Emb}(x_{\text{gt}})_{t''}$
25:     Update $\eta$ with GAN objective (Eq. (7))
26: **until** *convergence*
27: **Return** one-step generator $\theta$

---

### H.4 STRAIGHT-THROUGH GUMBEL IMPLEMENTATION

In frameworks such as PyTorch, setting `hard=True` in `F.gumbel_softmax` implements a *straight-through estimator* that discretizes the forward pass using $\arg\max$ while preserving continuous gradients using Gumbel softmax in the backward pass Wang (2025).

### H.5 ADDITIONAL EVALUATION DETAILS

For the evaluation of $\text{FID}_{\text{COCO}}$ and CLIP score, we follow the exact evaluation setup as GigaGAN (Kang et al., 2023) and DMD2 (Yin et al., 2024a). More specifically, we use 30K prompts from the COCO 2014 validation set to generate the corresponding images. The outputs are resized to 256×256 and compared with 40,504 real images from the same validation set using clean-FID (Parmar et al., 2022a). We compute the CLIP score using the *ViT-g-14-laion2b_s12b_b42k* encoder. All MaskBit ablations are trained with a batch size of 64 for 50k iterations. For ImageNet experiments, we evaluate final checkpoints across a sweep of generation temperatures and report the best metric for each checkpoint. For text-to-image experiments, the generation temperature is fixed at 1. The FID, precision, recall, density and coverage in the imagenet experiments are all calculated with features extracted from the InceptionV3 network. Note that the generation FID of our distilled MaskGit one-step generator outperforms that of the reconstruction FID of the tokenizer VQGAN. This is reasonable, as the reconstruction FID and generation FID for ImageNet are calculated against different reference data statistics, this is discussed in https://github.com/bytedance/1d-tokenizer/issues/46 (Yu et al., 2024a).

## I ADDITIONAL EXPERIMENTS AND FINDINGS

In this section, we present additional experimental analysis and findings. In Appendix I.1, we visualize and compare the tokenizers used in our work. In Appendices I.2, I.13 and I.14, we provide

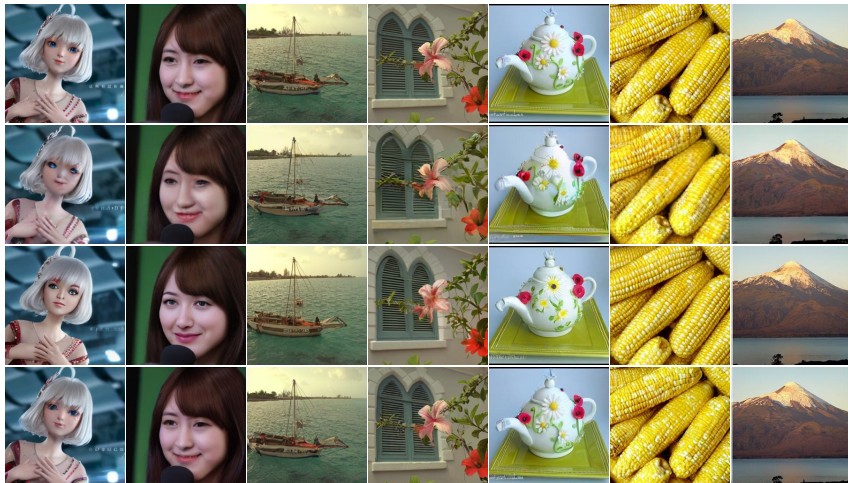

Figure 6: Comparison of reconstructions produced by different tokenizers. From top to bottom: (1) original image, (2) reconstruction using the MaskBit tokenizer, (3) reconstruction using the MaskGen TATiTok tokenizer, and (4) reconstruction from the Stable Diffusion v1.5 continuous VAE.

empirical evidence that the generator logits are highly concentrated. In Appendices I.3, I.4 and I.6, we show that our adversarial training algorithm is robust to a wide range of hyperparameters. In Appendices I.5 and I.7, we offer further comparisons between soft embedding and the Gumbel-Softmax straight-through estimator. In Appendix I.8, we ablate the number of TTEO steps and analyze the corresponding cost–quality tradeoff. In Appendix I.9, we study the effect of different teacher CFG value for distillation. In Appendices I.10 and I.11, we report results obtained with individual reward terms and investigate the effect of different reward weights. In Appendix I.12, we present additional comparisons to multi-step teacher models under different evaluation settings. Finally, in Appendix I.15, we provided the result of soft embedding in the Di[M]O distillation framework.

## I.1 RECONSTRUCTION ABILITY OF TOKENIZERS

We show reconstruction results produced by different tokenizers in Fig. 6. The figure compares the original image with reconstructions from the MaskBit tokenizer, the MaskGen TATiTok tokenizer, and the Stable Diffusion v1.5 continuous VAE. This comparison highlights the gap between the discrete image tokenizers and the continuous VAEs, illustrating the performance bottleneck of our method.

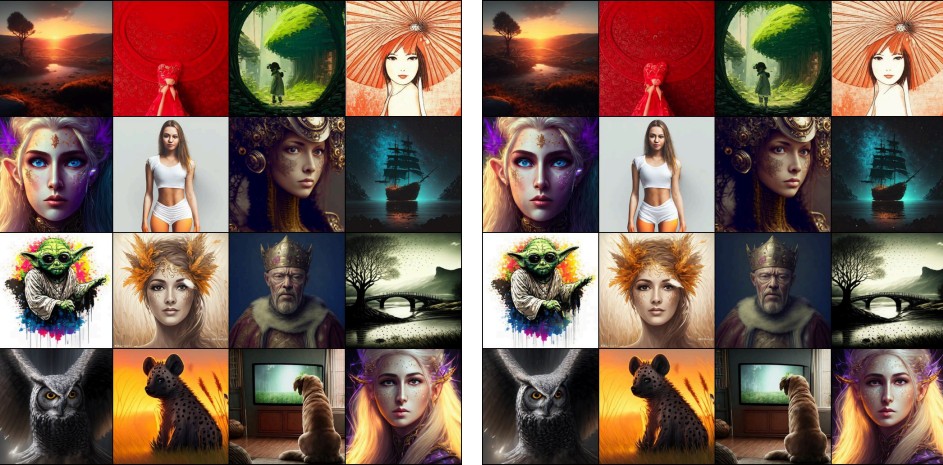

Figure 7: Comparison of decoded images produced from sampled discrete tokens (left) and from soft embeddings (right).

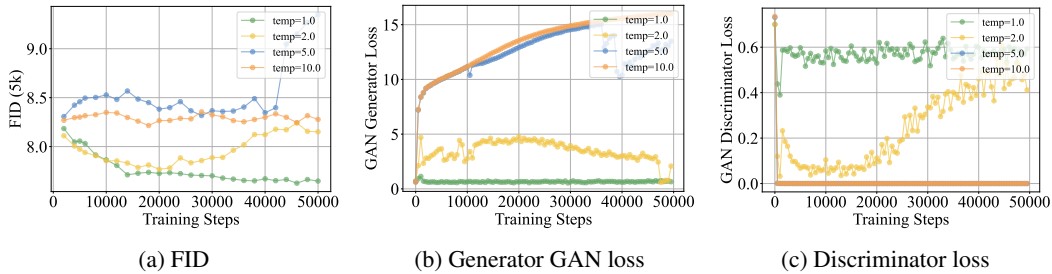

(a) FID       (b) Generator GAN loss       (c) Discriminator loss

Figure 8: Influence of different temperature in the calculation of softmax

## I.2 DECODED IMAGES FROM SAMPLED TOKENS VS. SOFT EMBEDDINGS

To validate the use of soft embeddings for our one-step generator, we compare images decoded from (i) sampled discrete tokens and (ii) the corresponding soft embeddings. Visually, the two sets of reconstructions are nearly identical (see Fig. 7). Quantitatively, over two batches of $64$ images, we measure a Mean Squared Error (MSE) of $6.02 \times 10^{-6}$ between the sampled-token reconstructions and the soft-embedding reconstructions. For reference, the tokenizer reconstruction MSE is $3.84 \times 10^{-6}$. These results support the practical usage of soft embeddings in our pipeline.

To further demonstrate the importance of representation fidelity, we vary the softmax temperature used to compute soft embeddings. In Fig. 8 we show that both model performance and training stability degrade as the temperature increases — higher temperatures produce more diffuse probability distributions and thus larger approximation errors in the soft embeddings.

## I.3 INFLUENCE OF GAN LOSS WEIGHT

In Fig. 9, we conduct experiments with different GAN loss weights when applying our method to the MaskBit teacher and showcase that we need larger GAN loss weights to achieve lower FID. In our setting, the Di[M]O loss weight for the generator is fixed to 1, and no reward function is applied during the distillation. The loss weight for the auxiliary model and discriminator are both set to 1, as they are decoupled. We only need to balance the GAN loss with the Di[M]O loss for the generator. We also validate by comparing the loss gradients w.r.t. logits $z_\theta$ gradients for different GAN loss weights and observe that we need larger GAN loss weights to have comparable GAN loss gradients against Di[M]O loss gradient.

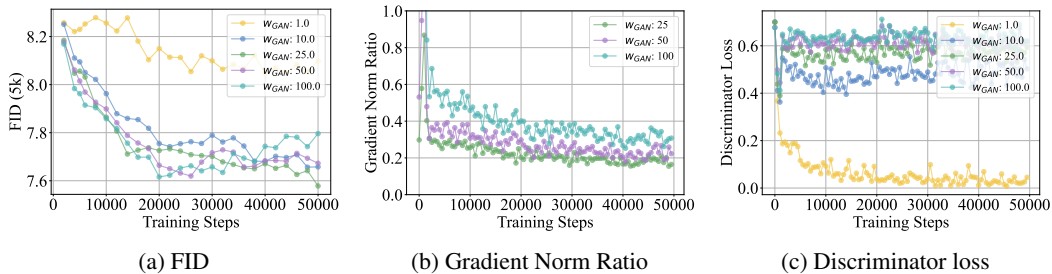

(a) FID       (b) Gradient Norm Ratio       (c) Discriminator loss

Figure 9: Influence with different GAN loss weights $w_{\text{GAN}}$. The GAN loss is not applied to the generator for the first 1000 steps.

## I.4 GAN LOSS REDUCES Di[M]O'S SENSITIVITY TO HYPERPARAMETERS

In addition to improving final performance, the GAN loss makes training more robust and easier to optimize. As shown in Fig. 10, when we initialize from a multistep MaskBit teacher and train with the Di[M]O loss (teacher CFG= 2) only, optimization rapidly converges to a local minimum. Introducing the GAN loss—either alone or in combination with the Di[M]O objective—helps the model escape that local minimum and attain a better FID. In practice, the best results are obtained by combining both losses: the Di[M]O term preserves faithful distillation from the teacher, while the GAN loss increases robustness and improves final sample quality.

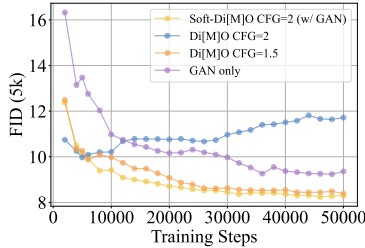 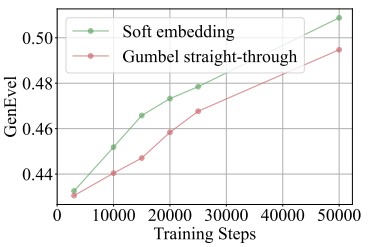

Figure 10: Ablation on the cfg value and GAN training. The default CFG value for MaskBit is 1.5 (see Tab. 5).

Figure 11: Ablation on the representation used for reward fine-tuning.

## I.5 ADDITIONAL COMPARISON WITH GUMBEL STRAIGHT-THROUGH

In Fig. 12 we provide comparison between our soft-embedding representation to the Gumbel straight-through in addition to Fig. 4a. Training with soft embeddings is noticeably more stable: the GAN generator loss is smoother and the real/fake discriminator logits show substantially less variance over the course of training.

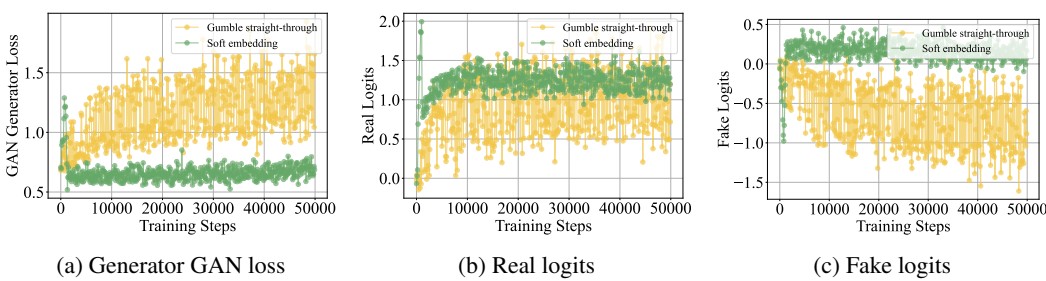

(a) Generator GAN loss      (b) Real logits      (c) Fake logits

Figure 12: Influence with different GAN loss weights $w_{\text{GAN}}$. The GAN loss is not applied to the generator for the first 1000 steps.

## I.6 GAN ALONG TRAINING LEADS TO MODE COLLAPSE

In our MaskGen experiments, when we have a large initial mask ratio $r_{\text{init}} = 0.95$, we observe mode collapse. As shown in Fig. 13, for different prompts the generator will produce the same output.

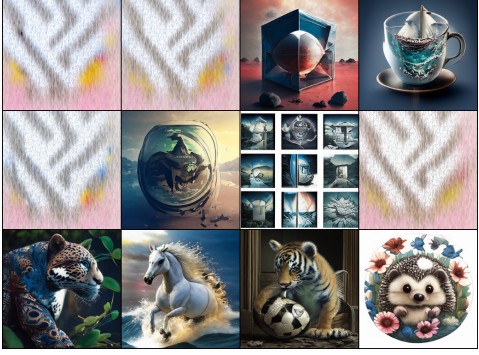

Figure 13: Mode collapse with GAN alone distillation.

## I.7 ABLATION OF SOFT EMBEDDING FOR MASKGEN-L TEACHER

We replicate the soft embedding ablation from Fig. 4a on the MaskGen-L setup and present the training curves with GenEval as metric in Fig. 11. This result shows that soft embeddings yield consistently higher GenEval scores during training, indicating improved prompt-following compared to using soft embeddings. In fact, the Gumbel–softmax behaves similarly to the direct softmax used in our soft-embedding approach when the injected Gumbel noise is small. However, as the noise increases, the approximation error in the resulting soft embedding also grows and will cause a drop in performance.

## I.8 Test-Time Embedding Optimization (TTEO) with Multi-Start Strategy

To reduce computational cost while maintaining performance, we employ TTEO combined with a multi-start Best-of-N (BoN) strategy, where multiple TTEO runs are executed in parallel for each prompt. In Fig. 14, we present the GenEval scores as a function of both the number of BoN seeds $N$ and the number of TTEO optimization iterations. Our results demonstrate that while the standard BoN baseline provides performance improvements, incorporating TTEO yields additional gains, thanks to soft embeddings. For all experiments reported in the main paper, we use $N = 4$ seeds and 4 TTEO iterations, which requires less inference time as pure BoN with $N = 64$. We provide detailed runtime comparison in Tab. 6, where the batch size is 4 for each prompt.

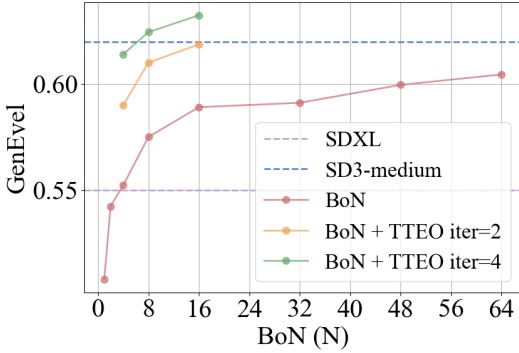

Figure 14: GenEval scores as a function of BoN sampling iterations $N$. Curves compare the BoN baseline and BoN combined with TTEO (iter=2,4); horizontal dashed lines show multi-step diffusion reference performance (SDXL (Podell et al., 2023), SD3-medium (Esser et al., 2024)).

Table 6: Runtime cost of different configuration in TTEO.

| Configuration | BoN=8 | BoN=16 | BoN=32 | BoN=64 | BoN=4 & TTEO iter=2 | BoN=8 & TTEO iter=2 | BON=4 & TTEO ITER=4 | BoN=8 & TTEO iter=4 |
|---|---|---|---|---|---|---|---|---|
| Runtime speed (s/prompt) | 1.46 | 2.64 | 4.70 | 8.57 | 2.52 | 4.32 | 5.66 | 9.88 |

## I.9 CFG of Teacher Model

The CFG mechanism for discrete models (e.g., AR and MDM) differs from the CFG commonly used in continuous diffusion/flow models (Dieleman, 2022). In this work, we use a constant CFG value across mask ratios as our default. We also experimented with several alternative CFG schedules (for example, linear schedules from 1.5 to 2 and cosine schedules from 1.5 to 2), but found that they do not improve Di[M]O's performance.

Tab. 7 reports metrics for Di[M]O applied to a MaskGen-L teacher using different teacher CFG values. Consistent with Zhou et al. (2024a), we observe a tradeoff between FID and CLIP score as the teacher CFG changes: a higher CFG produces better CLIP alignment but worse FID (for both COCO and MJHQ reference set). Importantly, further fine-tuning the distilled generator with a combination of the Di[M]O objective and a GAN loss reduces FID, with the largest improvements observed for $FID_{MJHQ}$. We also attempted experiments with MaskGen-XL; however, the provided checkpoint cause unstable training under large CFG values and the model tend to collapse during distillation with Di[M]O in those settings. We use the generator distilled with CFG=4 MaskGen-L teacher as the initialization of our Soft-Di[M]O training.

## I.10 Reward-only Fine-tuning

In Tab. 8, we present results from fine-tuning the generator using different reward functions alone. We make the following observations. CLIP score is more strongly correlated with GenEval, reflecting improved prompt-following ability; in contrast, HPS primarily reflect aesthetic quality. Fine-tuning with CLIP reward alone improves prompt following (GenEval and CLIP score) but does not influence HPS score. On the other hand, optimizing HPS, ImageReward, or PickScore improves HPS and also yields slightly gains in GenEval (also hurt the FID to different extends).

Table 7: Di[M]O performance on MaskGen-L with different CFG value. Models are trained with batch size of 64 and 100k steps

| CFG value | 2.0 | 2.0 +GAN | 3.0 | 4.0 | bs×4 | +GAN | 5.0 | 6.0 | 7.0 | 8.0 |
|---|---|---|---|---|---|---|---|---|---|---|
| $FID_{MJHQ}$ (↓) | 6.79 | 5.87 | 8.52 | 9.73 | 9.68 | 6.58 | 10.45 | 10.96 | 11.35 | 11.57 |
| $FID_{COCO}$ (↓) | 22.74 | 21.26 | 23.99 | 24.15 | 24.56 | 21.46 | 24.56 | 24.60 | 24.72 | 24.77 |
| CLIP-Score (↑) | 0.296 | 0.294 | 0.298 | 0.299 | 0.299 | 0.298 | 0.298 | 0.298 | 0.298 | 0.298 |
| GenEval (↑) | 0.420 | 0.406 | 0.432 | 0.424 | 0.430 | 0.429 | 0.426 | 0.427 | 0.439 | 0.425 |
| HPSv2.0 (↑) | 27.46 | 27.28 | 27.49 | 27.50 | 27.55 | 27.37 | 27.50 | 27.48 | 27.46 | 27.45 |
| HPSv2.1 (↑) | 27.16 | 26.63 | 27.19 | 27.14 | 27.35 | 26.85 | 27.13 | 27.08 | 27.01 | 26.99 |

We further provide qualitative visual comparisons of generators fine-tuned with different single rewards in Fig. 15. The over-saturated appearance in some outputs is consistent with observations reported in Li et al. (2024b).

Table 8: Di[M]O model fine-tuned with different reward model alone, trained with 50k steps and bs=32. The similar effect of single reward alone fine-tuning is also reported in Reward-Instruct (Luo et al., 2025b), but they did not report that the CLIP score can help with GenEval metric.

| Reward Function | HPS (Wu et al., 2023) | ImageReward (Xu et al., 2023) | CLIP (Radford et al., 2021) | PickScore (Kirstain et al., 2023) | Baseline (initial model) |
|---|---|---|---|---|---|
| $FID_{MJHQ}$ (↓) | 34.59 | 21.80 | 9.59 | 22.36 | 9.73 |
| $FID_{COCO}$ (↓) | 38.40 | 33.62 | 20.68 | 35.96 | 24.15 |
| CLIP-Score (↑) | 0.287 | 0.276 | 0.332 | 0.302 | 0.299 |
| GenEval (↑) | 0.433 | 0.450 | 0.503 | 0.432 | 0.424 |
| HPSv2.0 (↑) | 28.58 | 28.45 | 27.72 | 27.60 | 27.50 |
| HPSv2.1 (↑) | 33.89 | 31.18 | 27.44 | 28.83 | 27.14 |

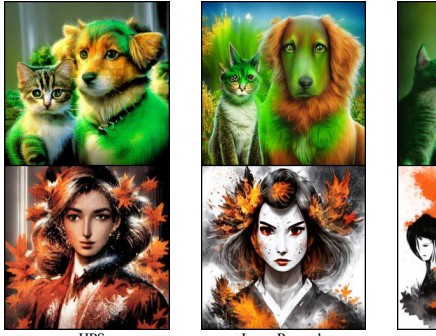 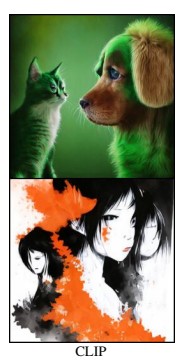 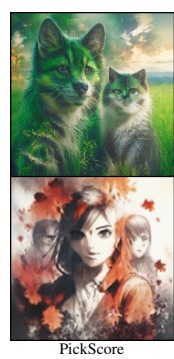 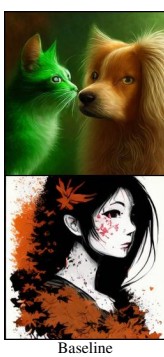

HPS    ImageReward    CLIP    PickScore    Baseline

Figure 15: Visual comparison of models fine-tuned with individual reward functions. Each column shows outputs from a model fine-tuned only on the indicated reward (HPS, ImageReward, CLIP, PickScore) and the baseline (initial checkpoint used for fine-tuning). Prompts used are: `A green dog and a red cat` and `A dynamic anime poster featuring a collection of Asian women in a style blending realistic watercolors and modern anime. The palette is gloomy and dramatic, dominated by dark orange and white. Weathercore aesthetic with swirling autumn leaves and mist, mysterious realism, evocative and emotional, cinematic composition --ar 69:128 --s 750 --niji 5.`

## I.11 BALANCING CLIP AND IMAGEREWARD

Based on the results and analysis in Appendix I.10, we use only the ImageReward and CLIP in our reward loss for the fine-tuning stage. In Tab. 9, we show results from fine-tuning the distilled MaskGen-L generator with ImageReward fixed at weight 1 and varying the relative CLIP weight (overall reward loss weight = 2000 in comparison to a Di[M]O loss weight of 1). Increasing the CLIP weight consistently improves CLIP-Score and GenEval (better prompt alignment) but degrades Aesthetic metrics (HPSv2.0 / HPSv2.1). In practice, a moderate CLIP weight (roughly 0.2–0.5) offers a good trade-off between prompt adherence and image quality, and we select 0.2 as our final choice. A similar trend is observed for the Meissonic teacher as in Tab. 10 (even only reward loss is used).

Table 9: fine-tune Di[M]O model of MaskGen teacher with Di[M]O loss and reward, trained with 50k steps and bs=32. We use both ImageReward and CLIP as reward and ablate the relative loss weight of CLIP compared to ImageReward. The Di[M]O loss weight is 1, and the overall reward loss weight is 2000. Inside the reward loss, the ImageReward loss weight is 1., and we change the CLIP loss weight here.

| CLIP relative weight | 0.0 | 0.01 | 0.05 | 0.1 | 0.2 | 0.5 | 1.0 |
|---|---|---|---|---|---|---|---|
| $FID_{MJHQ}$ ($\downarrow$) | 12.11 | 11.36 | 10.48 | 9.74 | 10.36 | 9.86 | 10.43 |
| $FID_{COCO}$ ($\downarrow$) | 28.32 | 26.89 | 25.71 | 24.24 | 23.30 | 22.73 | 22.29 |
| CLIP-Score ($\uparrow$) | 0.295 | 0.298 | 0.309 | 0.314 | 0.321 | 0.322 | 0.326 |
| GenEval ($\uparrow$) | 0.470 | 0.451 | 0.493 | 0.498 | 0.509 | 0.496 | 0.494 |
| HPSv2.0 ($\uparrow$) | 28.31 | 28.30 | 28.30 | 28.26 | 28.18 | 28.01 | 27.90 |
| HPSv2.1 ($\uparrow$) | 30.49 | 30.41 | 30.16 | 30.02 | 29.45 | 28.85 | 28.28 |

Table 10: fine-tune Di[M]O model of Meissonic teacher with Di[M]O loss and reward, trained with 10k steps and bs=16. We use both ImageReward and CLIP as reward and ablate the relative loss weight of CLIP compared to ImageReward. The Di[M]O loss weight is 0, and the overall reward loss weight is 1000. Inside the reward loss, the ImageReward loss weight is 1, and we change the CLIP loss weight here. This table is measured with generator temperature=0.5.

| CLIP relative weight | 0.0 | 0.1 | 0.2 | 0.5 | 1.0 |
|---|---|---|---|---|---|
| GenEval ($\uparrow$) | 0.496 | 0.518 | 0.523 | 0.523 | 0.529 |
| HPSv2.0 ($\uparrow$) | 29.33 | 29.32 | 29.22 | 28.93 | 28.89 |
| HPSv2.1 ($\uparrow$) | 33.15 | 32.98 | 32.54 | 31.51 | 31.34 |

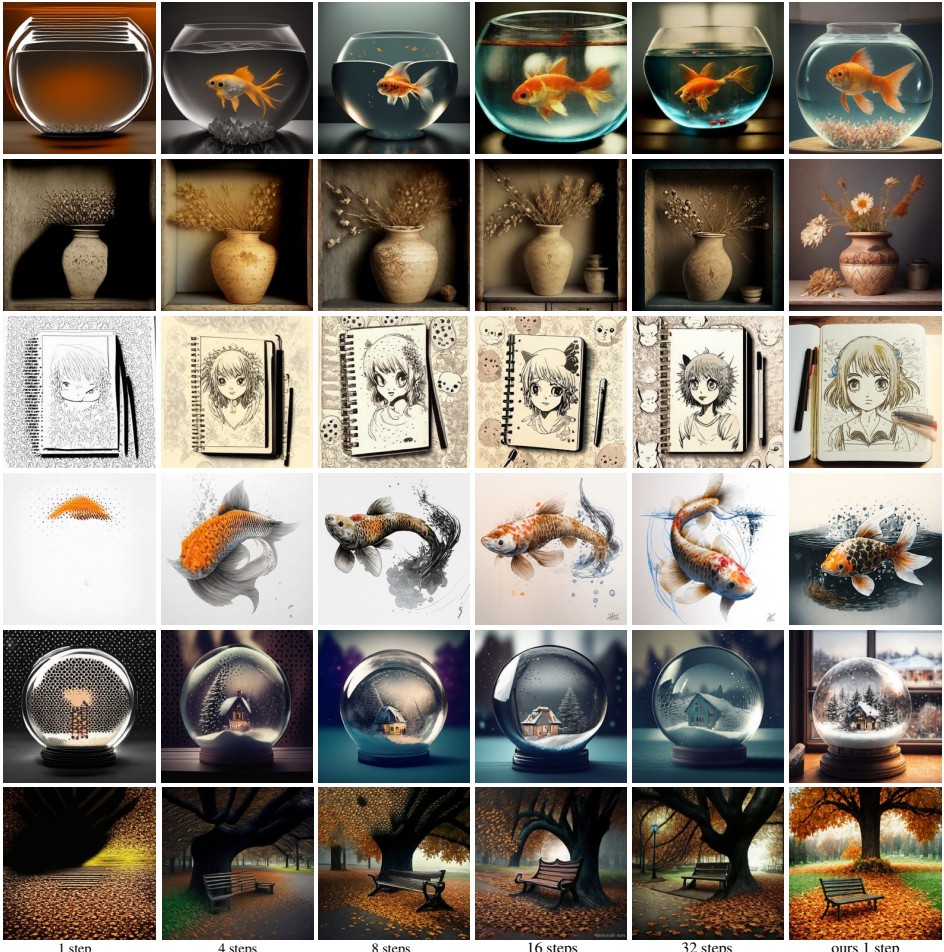

|  |  |  |  |  |  |
|---|---|---|---|---|---|
| 1 step | 4 steps | 8 steps | 16 steps | 32 steps | ours 1 step |

Figure 16: One-step generation from our one-step generator compared to multi-step MaskGen-L teacher.

## I.12 COMPARISON WITH MULTI-STEP TEACHER

In Tabs. 11 to 13, we compare our final one-step generator with the multi-step MaskGen-L teacher across all metrics. The results show that our one-step generator achieves both stronger prompt-following ability and higher aesthetic quality, without sacrificing FID. We provide a visual comparison in Fig. 16.

Table 11: Comparison of FID$_{COCO}$, FID$_{MJHQ}$ and CLIP-Score for MaskGen-L (Kim et al., 2025) across varying generation steps and our one-step generator. The results are evaluated on MSCOCO-val 30k dataset. The teacher sampling config is the same as the official ones. The asterisk item denotes the performance reported in the original paper.

| Steps ($\downarrow$) | 64 | 32 | 16* | 16 | 8 | 4 | 2 | 1 | 1 *(ours)* |
|---|---|---|---|---|---|---|---|---|---|
| FID$_{COCO}$ ($\downarrow$) | 22.60 | 22.70 | 22.78 | 22.64 | 22.89 | 24.84 | 44.21 | 88.70 | 23.44 |
| FID$_{MJHQ}$ ($\downarrow$) | 7.35 | 7.43 | 7.74 | 7.59 | 8.32 | 11.58 | 29.92 | 77.57 | 10.59 |
| CLIP-Score ($\uparrow$) | 0.312 | 0.312 | 0.307 | 0.312 | 0.311 | 0.308 | 0.287 | 0.245 | 0.322 |

Table 12: Performance of Geneval evaluation across varying generation steps for L and XL. Each cell shows L / XL.

| Steps ($\downarrow$) | 64 | 32 | 16 | 8 | 4 | 2 | 1 | 1 *(ours)* |
|---|---|---|---|---|---|---|---|---|
| Position ($\uparrow$) | 0.080 / 0.063 | 0.055 / 0.058 | 0.075 / 0.058 | 0.080 / 0.048 | 0.075 / 0.038 | 0.050 / 0.013 | 0.020 / 0.000 | 0.09 |
| Two Objects ($\uparrow$) | 0.540 / 0.543 | 0.545 / 0.546 | 0.540 / 0.515 | 0.477 / 0.498 | 0.414 / 0.371 | 0.249 / 0.205 | 0.020 / 0.040 | 0.59 |
| Colors ($\uparrow$) | 0.787 / 0.811 | 0.795 / 0.814 | 0.800 / 0.843 | 0.763 / 0.822 | 0.766 / 0.793 | 0.559 / 0.559 | 0.287 / 0.330 | 0.81 |
| Color Attributes ($\uparrow$) | 0.148 / 0.140 | 0.133 / 0.158 | 0.138 / 0.150 | 0.120 / 0.118 | 0.103 / 0.105 | 0.030 / 0.033 | 0.010 / 0.010 | 0.18 |
| Counting ($\uparrow$) | 0.384 / 0.400 | 0.372 / 0.403 | 0.375 / 0.381 | 0.353 / 0.381 | 0.378 / 0.378 | 0.244 / 0.203 | 0.113 / 0.050 | 0.41 |
| Single Object ($\uparrow$) | 0.972 / 0.944 | 0.972 / 0.947 | 0.972 / 0.947 | 0.978 / 0.947 | 0.966 / 0.950 | 0.813 / 0.844 | 0.525 / 0.563 | 0.98 |
| Overall Score ($\uparrow$) | 0.485 / 0.483 | 0.479 / 0.487 | 0.484 / 0.482 | 0.462 / 0.469 | 0.448 / 0.439 | 0.324 / 0.309 | 0.178 / 0.165 | 0.509 |

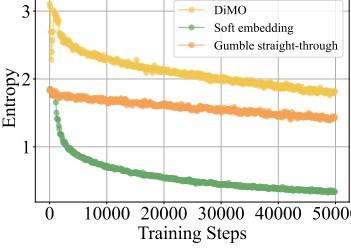

Figure 17: Entropy evolution during distillation.

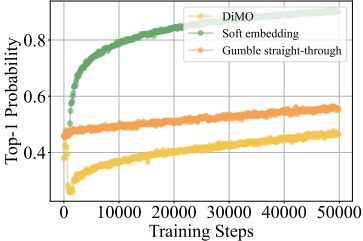

Figure 18: Top-1 probability evolution during distillation.

### I.13 SAMPLING FROM ONE-STEP GENERATOR WITH $\arg\max$

To validate the observation from Di[M]O (Zhu et al., 2025b) and ReDi (Yoo et al., 2025) that output logits are highly concentrated with sharp predicted distributions, we compare the FID of images sampled from the output distribution versus images obtained using Argmax in Tab. 14. As shown, both methods achieve nearly identical results, demonstrating that the output distribution is sharp enough for deterministic Argmax sampling (given the redundancy of the discrete latent space).

### I.14 ENTROPY AND TOP-1 PROBABILITY EVOLUTION

In this subsection, we analyze the evolution of entropy and top-1 probability under different configurations. Both metrics are computed as the mean across all tokens during distillation. We first present the evolution during Di[M]O distillation as a reference, where entropy decreases and top-1 probability increases over training. This Di[M]O checkpoint is then used to initialize our Soft-Di[M]O distillation. We compare the training dynamics of Soft-Di[M]O using our soft embedding approach versus Gumbel softmax straight-through estimation. As shown in Figs. 17 and 18, our proposed soft embedding exhibits stronger concentration of the predicted logits compared to Gumbel softmax, achieving lower entropy and higher top-1 probability, corresponding to Fig. 4a.

Table 13: Comparison of HPSV2.0 and HPSV2.1 scores across different generation steps for various visual styles. (L). Each cell shows v2.0 / v2.1.

| Style (↑) | 64 | 32 | 16 | 8 | 4 | 2 | 1 | 1 (*ours HPSV2.1*) |
|---|---|---|---|---|---|---|---|---|
| Anime | 28.20 / 29.37 | 28.19 / 29.34 | 28.13 / 29.09 | 28.04 / 28.72 | 27.79 / 27.68 | 26.65 / 23.78 | 24.73 / 17.93 | 29.51 |
| Concept-Art | 27.42 / 27.83 | 27.37 / 27.60 | 27.35 / 27.54 | 27.26 / 27.20 | 27.04 / 26.15 | 26.00 / 22.52 | 24.30 / 17.15 | 30.62 |
| Paintings | 27.48 / 27.73 | 27.44 / 27.59 | 27.39 / 27.32 | 27.37 / 27.15 | 27.02 / 25.89 | 25.93 / 21.98 | 24.16 / 16.43 | 29.34 |
| Photo | 27.45 / 25.89 | 27.47 / 25.87 | 27.45 / 25.78 | 27.38 / 25.51 | 27.14 / 24.70 | 26.09 / 21.25 | 24.13 / 15.08 | 28.06 |
| Average | 27.64 / 27.70 | 27.62 / 27.60 | 27.58 / 27.43 | 27.52 / 27.14 | 27.25 / 26.11 | 26.17 / 22.38 | 24.33 / 16.65 | 29.38 |

Table 14: Comparison of sampling methods showing nearly identical performance between sampling from logits and using $\arg\max$.

| Method | FID-50K | IS | Precision | Recall |
|---|---|---|---|---|
| Sample from logits: `Categorical(logits).sample` | 1.56 | 273.2 | 0.81 | 0.60 |
| Argmax: `torch.argmax(logits, dim=-1)` | 1.57 | 273.3 | 0.81 | 0.60 |

Table 15: Comparison of standard Di[M]O and Di[M]O using soft embedding.

| Method | FID-50K | IS | Precision | Recall |
|---|---|---|---|---|
| Standard Di[M]O | 2.89 | 310.1 | 0.87 | 0.49 |
| Soft embedding variant DiMO | 3.33 | 315.9 | 0.88 | 0.47 |

## I.15    USE SOFT EMBEDDING ALONE IN DI[M]O

While soft embedding is originally introduced to provide additional supervision beyond the teacher model, it can also be integrated directly into the Di[M]O loss computation **alone, without applying further downstream refinement losses**. As illustrated in Fig. 19, the standard Di[M]O pipeline (top) first samples discrete tokens $x_\theta$, applies the forward mask-diffusion process to obtain the intermediate state $\tilde{x}_t$, and then feeds $\tilde{x}_t$ into the teacher $\phi$ and auxiliary model $\psi$ to construct the Di[M]O loss. In the soft-embedding variant (bottom), we instead take the model logits $z_\theta$ and immediately compute the corresponding soft embeddings using the teacher's embedding layer (the embedding layer of the teacher and the auxiliary model is the same and fixed all the time). Following the same mask schedule as in the standard diffusion process, a subset of these embeddings is then replaced by the mask-token embedding. This partially masked soft-embedding representation is subsequently passed to the teacher $\phi$ and auxiliary $\psi$ to form the Di[M]O loss. Compared to the original Di[M]O loss, the only change is replacing sampled discrete tokens with their soft-embedding counterparts in the forward pass. Importantly, in this variant, the Di[M]O loss **does not backpropagate** through the soft-embedding computation, the relaxation is used only to stabilize the forward intermediate representation, not as a differentiable surrogate. This contrasts with the refinement techniques introduced in the main method (GAN loss, reward tuning, TTEO), where gradients do propagate through the soft embedding, which is essential for enabling these refinement techniques on one-step MDMs.

In Tab. 15, we show that soft embedding variant of DiMO has a slight degradation in FID and recall compared to standard DiMO, likely due to the smoothing effect of soft embedding. However, the IS score and precision improves, potentially indicating that soft embedding helps generate higher-quality samples at the cost of diversity. This result also suggests that soft embedding serves as a good representation for the sampled tokens. In Fig. 20, we further provide the FID curve during distillation for both approaches, it shows that while the soft embedding variant (yellow curve) gives fast convergence, but the final converged performance of both approaches are similar with small gap. We leave it as future work to investigate this early convergence.

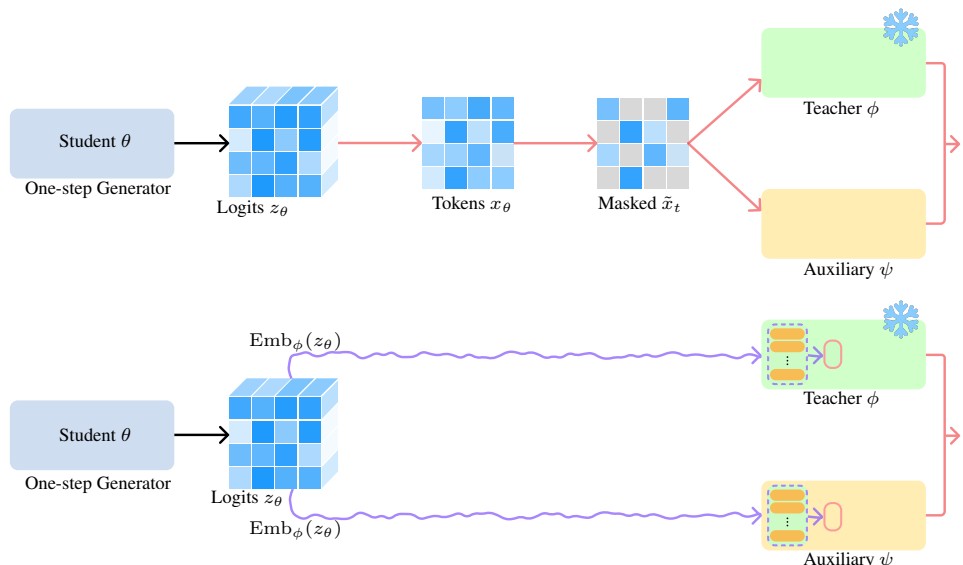

Figure 19: **Using soft embedding in Di[M]O.** Top: the original Di[M]O loss formulation. Bottom: the Di[M]O variant with soft embedding, where intermediate states are represented using "partially masked" embeddings. The pink operator after the embedding layer indicates that a random subset of soft embeddings is replaced with mask embeddings, following the same masking schedule as in the standard forward mask-diffusion process.

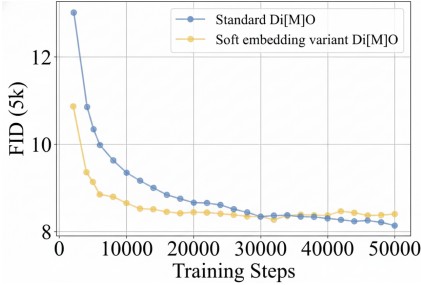

Figure 20: **FID comparison of standard Di[M]O and soft embedding variant Di[M]O.** The only difference of two experiments is the way to construct the input to teacher and auxiliary model, as illustrated in Fig. 19.

## J FAILURE CASES

Compared to the Di[M]O generator, we observe that our reward fine-tuned generator produces less diverse images for certain prompts (only a small portion in practice), as illustrated in Fig. 21.

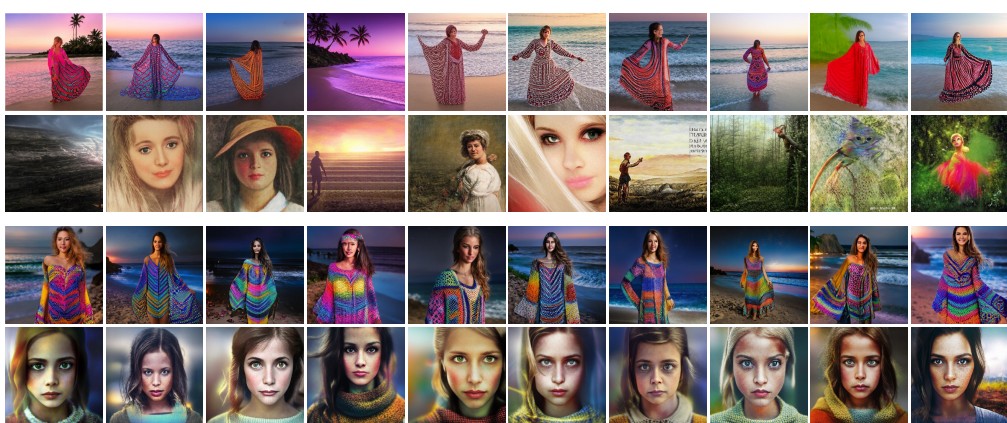

Figure 21: Visual examples of mode collapse of Soft-Di[M]O-MaskGen-L for certain prompts. Upper are generated from Di[M]O generator and lower is generated with Soft-Di[M]O with MaskGen-L teacher. Prompts used are: `Fantasie Antheia Crochet Beach Cover Up 6552 Small Twilight` and `could not found the image.`

Table 16: Class-conditional ImageNet-256 results with MaskBit teacher. This is a more complete version of Tab. 1.

| Family | Method | Steps ($\downarrow$) | #Params | FID($\downarrow$) | IS($\uparrow$) | Pr($\uparrow$) | Rec($\uparrow$) |
|---|---|---|---|---|---|---|---|
| GAN | BigGAN (Brock et al., 2018) | 1 | 112M | 6.95 | 224.46 | 0.87 | 0.28 |
| | GigaGAN (Kang et al., 2023) | 1 | 569M | 3.45 | 225.52 | 0.84 | 0.61 |
| | StyleGAN-XL (Sauer et al., 2022) | 1 | 166M | 2.30 | 265.12 | 0.78 | 0.53 |
| Diffusion & Flow | ADM (Dhariwal & Nichol, 2021) | 250 | 554M | 4.59 | 186.70 | 0.82 | 0.52 |
| | LDM-4-G (Rombach et al., 2022) | 250 | 400M | 3.60 | 247.67 | - | - |
| | U-ViT-H (Bao et al., 2023) | 50 | 501M | 2.29 | 263.88 | 0.82 | 0.57 |
| | DiT-XL/2 ($w = 1.5$) (Peebles & Xie, 2023) | 250 | 675M | 2.27 | 278.24 | 0.83 | 0.57 |
| | SiT-XL/2 ($w = 1.5$ SDE) (Ma et al., 2024) | 250 | 675M | 2.06 | 277.50 | 0.83 | 0.59 |
| Masked & AR | MaskGIT (Chang et al., 2022) | 8 | 227M | 6.18 | 182.1 | 0.80 | 0.51 |
| | LlamaGen-3B (Sun et al., 2024) | 576 | 3.1B | 2.18 | 263.3 | 0.81 | 0.58 |
| | GIVT (Tschannen et al., 2024) | 256 | 304M | 3.35 | - | 0.84 | 0.53 |
| | MAR (Li et al., 2024c) | 100 | 400M | 1.98 | - | - | - |
| | VAR-$d$20 (Tian et al., 2024) | 10 | 600M | 2.57 | 302.6 | 0.83 | 0.56 |
| | TiTok-S-128 (Yu et al., 2024b) | 64 | 287M | 1.97 | 281.8 | - | - |
| | MAGVIT-v2 (Yu et al., 2023) | 64 | 307M | 1.78 | 319.4 | - | - |
| | MaskBit (Weber et al., 2024) | 256 | 305M | 1.52 | 328.6 | - | - |
| | MaskBit (Weber et al., 2024) | 256 | 305M | 1.58 | 306.12 | 0.81 | 0.60 |
| | MaskBit (Weber et al., 2024) | 64 | 305M | 1.66 | 319.97 | 0.81 | 0.60 |
| Few-Step from Scratch | iCT (Song et al., 2023) | 1 | 675M | 34.24 | - | - | - |
| | | 2 | 675M | 20.3 | - | - | - |
| | Shortcut (Frans et al., 2024) | 1 | 675M | 10.60 | - | - | - |
| | | 4 | 675M | 7.80 | - | - | - |
| | IMM (Zhou et al., 2025a) (XL/2, $w = 1.5$) | 1 | 675M | 8.05 | - | - | - |
| | | 8 | 675M | 1.99 | - | - | - |
| | MeanFlow (Geng et al., 2025) (XL/2) | 1 | 676M | 3.43 | - | - | - |
| | | 2 | 676M | 2.93 (2.20 train longer) | - | - | - |
| Discrete Distillation | LlamaGen-L-DD (Liu et al., 2024) | 2 | 326M | 7.58 | 237.5 | 0.84 | 0.37 |
| | Di[M]O-MaskGit (Zhu et al., 2025b) | 1 | 174M | 6.91 | 214.05 | 0.828 | 0.377 |
| | Di[M]O-MaskBit (Zhu et al., 2025b) | 1 | 305M | 2.89 | 310.13 | 0.87 | 0.49 |
| | Soft-Di[M]O (MaskBit teacher) | 1 | 305M | 1.96 | 281.35 | 0.84 | 0.55 |
| | Soft-Di[M]O (MaskBit teacher) (train longer) | 1 | 305M | 1.56 | 273.2 | 0.81 | 0.60 |

## K    MORE QUALITATIVE RESULTS

In Figs. 22 and 23, we present randomly sampled ImageNet images generated in a single step by our distilled models with MaskBit teacher and MaskGit teacher, respectively. In Fig. 24, we present image samples generated unconditionally in a single step by our distilled models with MaskGen-L teacher. Finally, in Figs. 25 and 26, we provide additional text-to-image one-step generation results from our distilled model with the Meissonic teacher and MaskGen-L teacher.

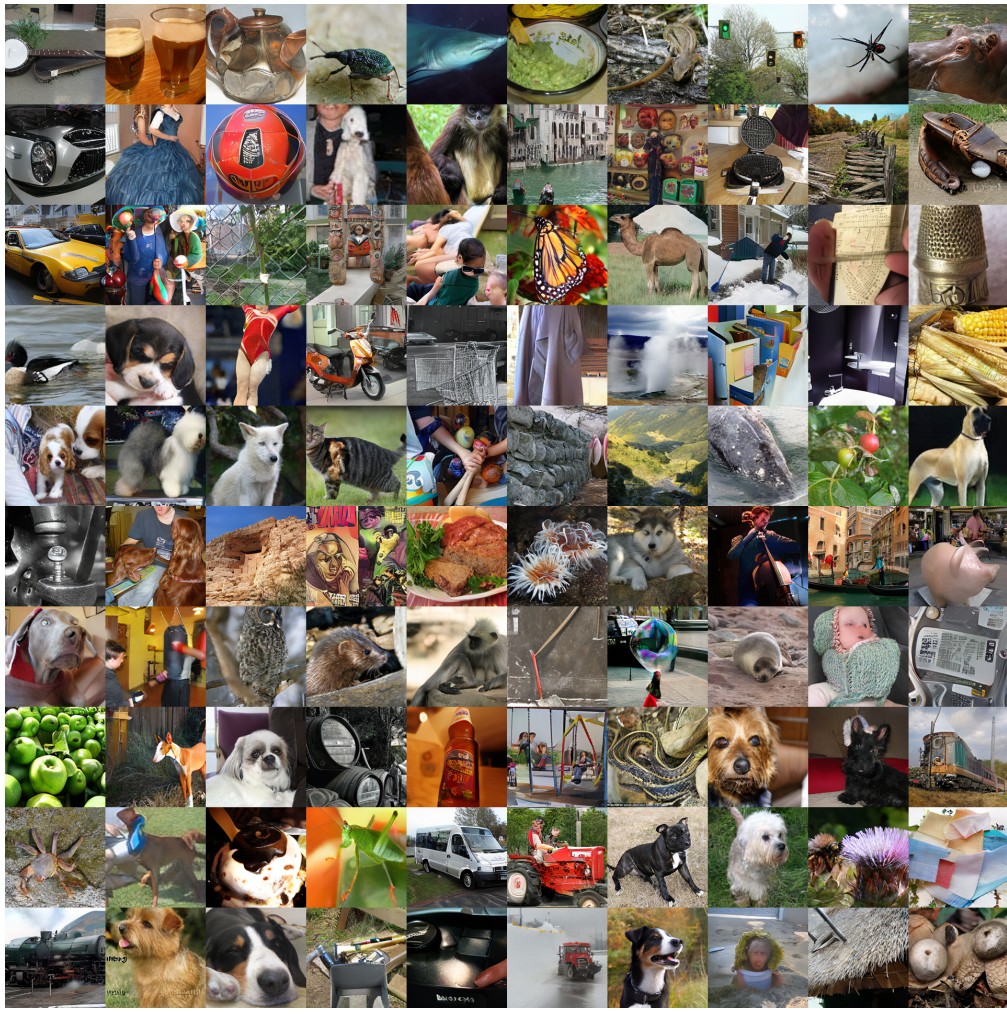

Figure 22: Uncurated images from one-step generation with MaskBit teacher.

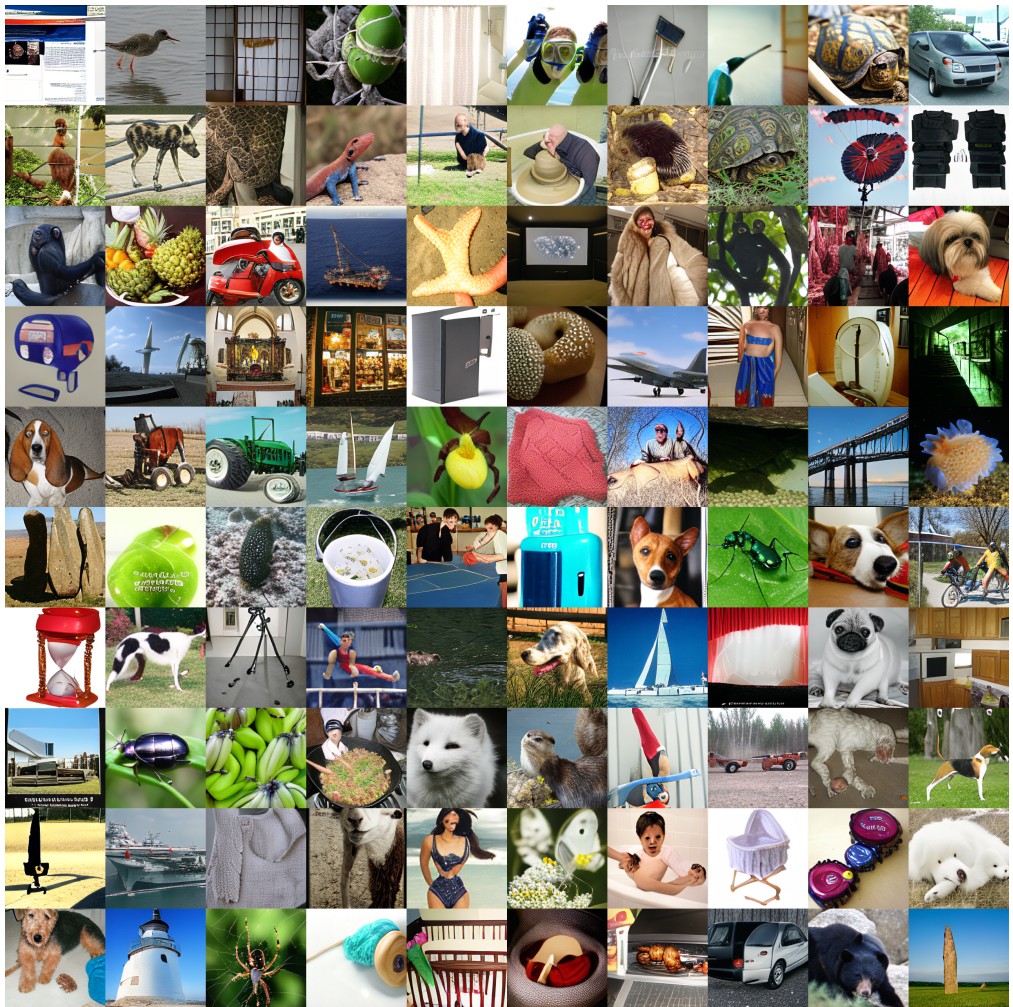

Figure 23: Uncurated images from one-step generation with MaskGit teacher.

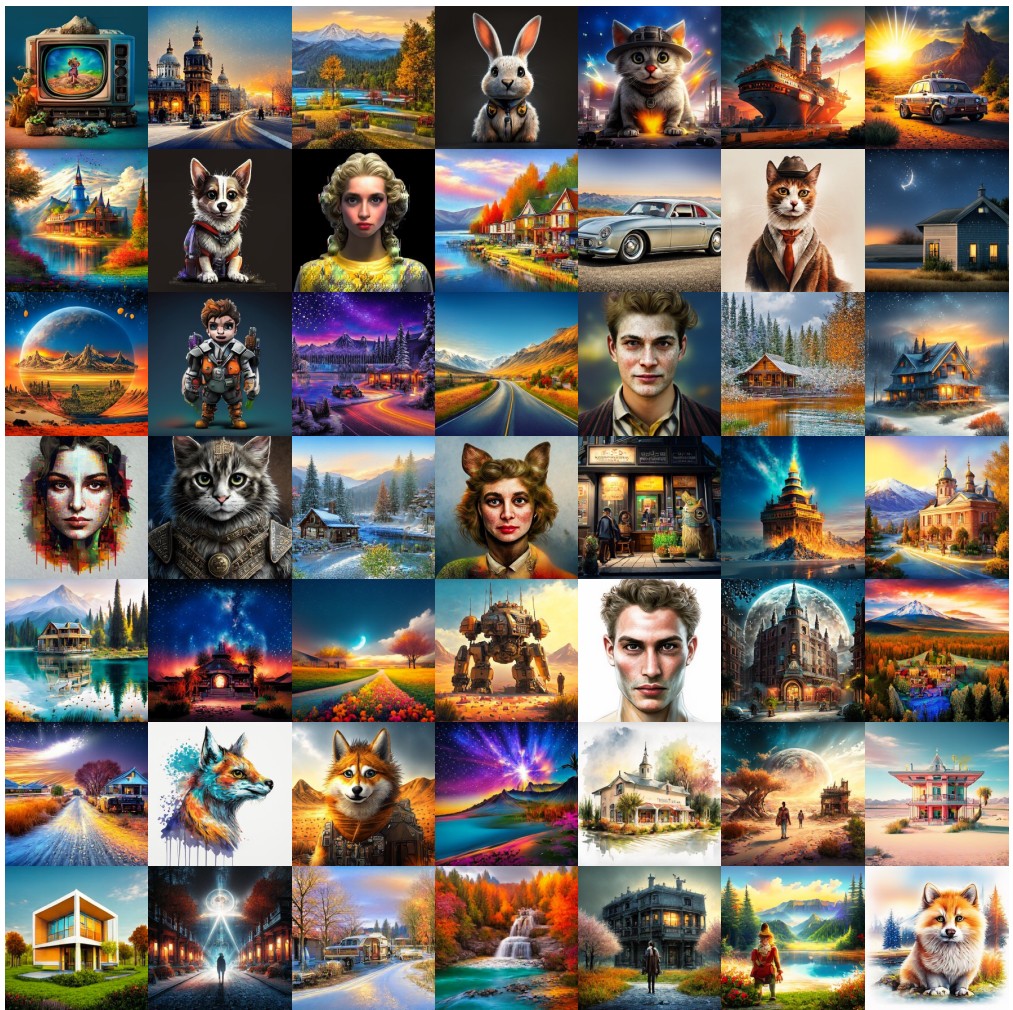

Figure 24: One-step *unconditional* generation from our reward fine-tuned one-step generator with MaskGen-L teacher.

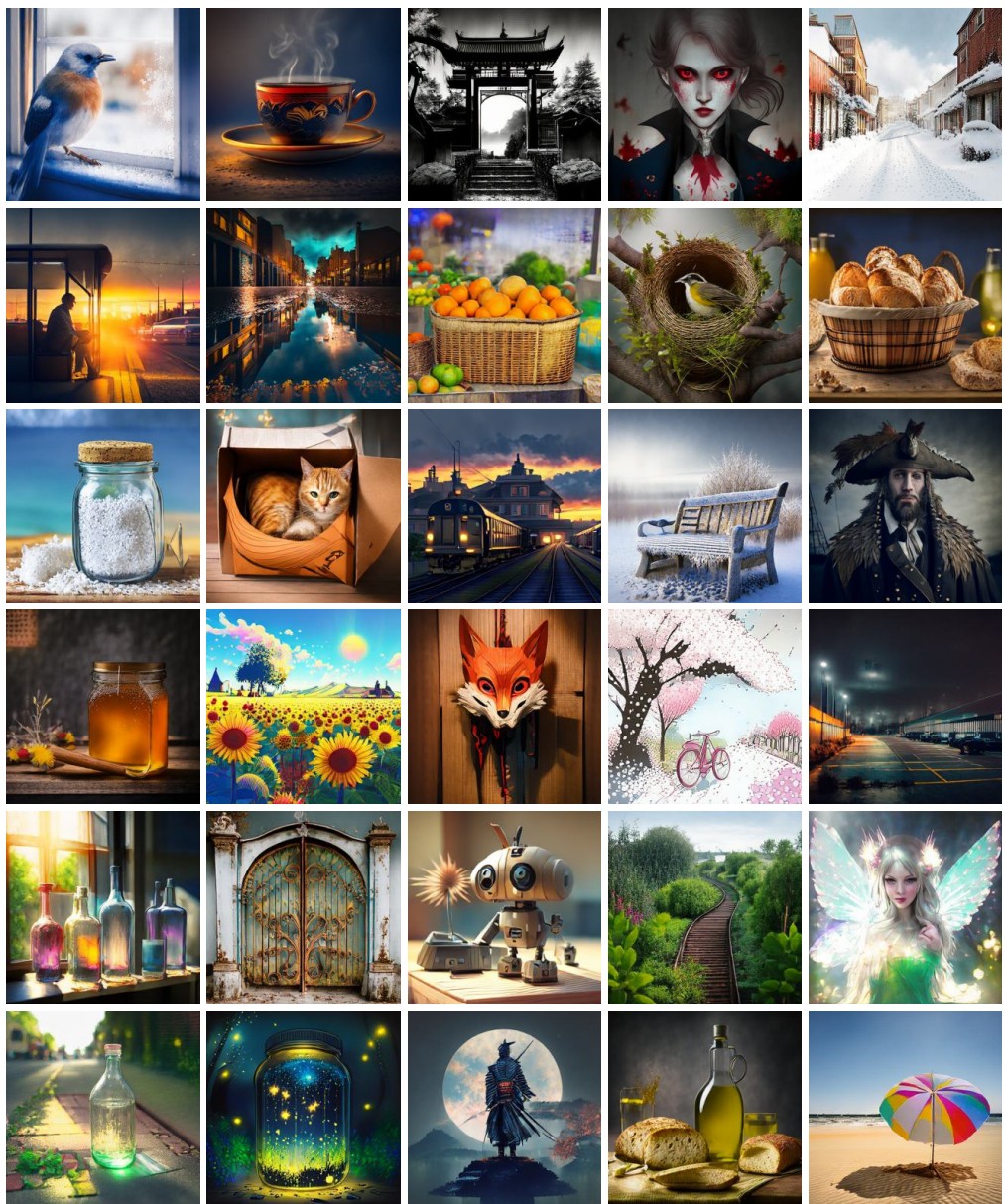

Figure 25: One-step generation from our reward fine-tuned one-step generator with MaskGen-L teacher.

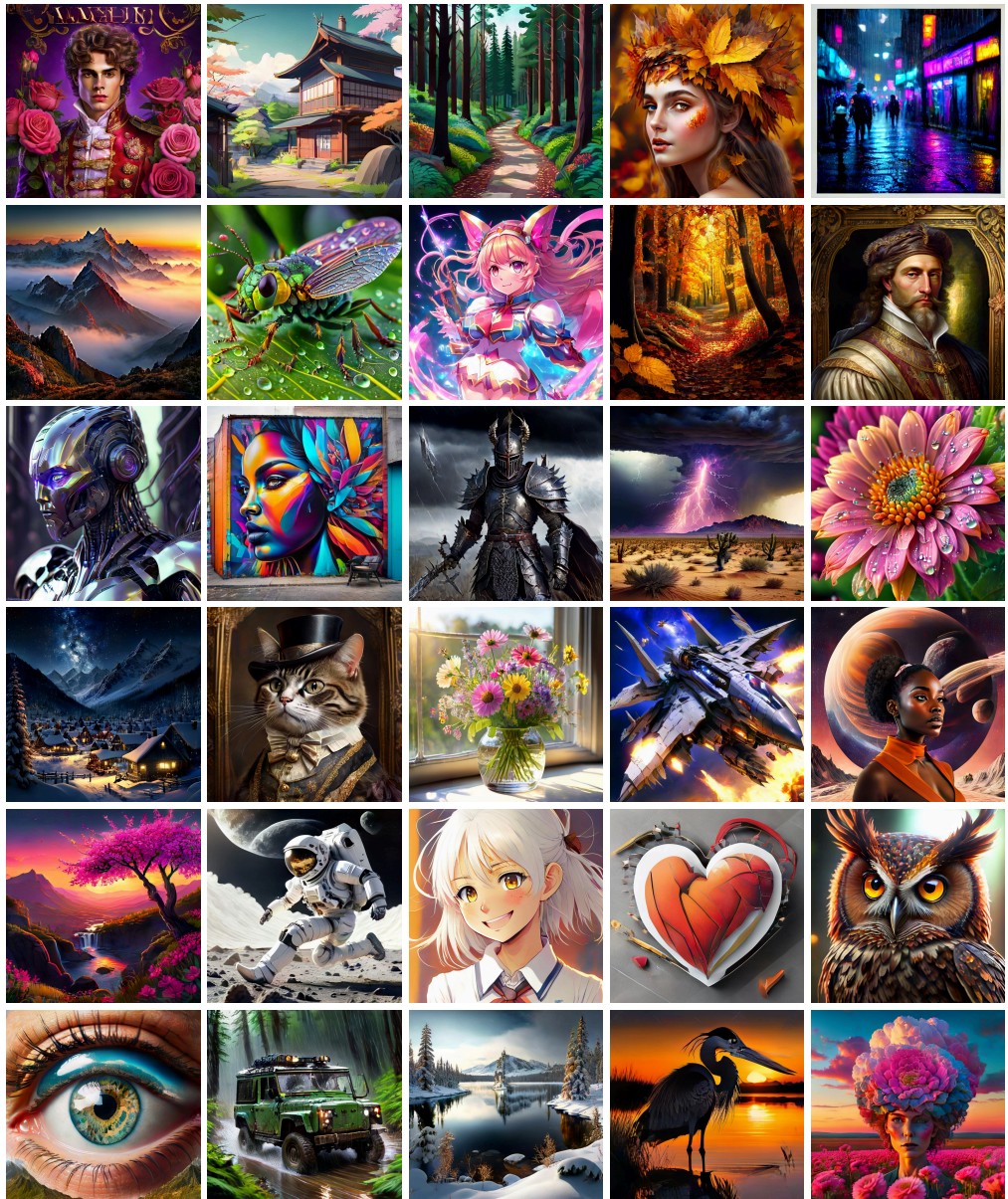

Figure 26: One-step generation from our reward fine-tuned one-step generator with Meissonic teacher.

## L  MISC.

**Prompts**  Below is a collection of creative prompts we used to generate images in Figs. 1, 3, 7, 16, 25 and 26 (in the order from left to right, from top to bottom):

**Fig. 1**:

- A pair of old leather boots sprouting tiny photorealistic flowers and moss, blending nature and object.
- A distinguished older gentleman in a vintage study, surrounded by books and dim lighting, his face marked by wisdom and time. 8K, hyper-realistic, cinematic, post-production.
- A cute pixel art of a cat in a sunny garden.
- Design tranquil interiors that celebrate the calming power of sage green.
- A floating castle in the sky, waterfalls cascading into the clouds, painted in a whimsical Studio Ghibli style.
- Ombre color scheme of neon yellow, neon blue, neon pink, neon green, neon orange, cute adorable alien insectoid Siberian musk deer, Chinese water deer creature with (Clawed Appendages) and a colorful (Turquoise)-Branched Horns, transparent nodules, highly detailed alien wildlife photography, intricate biological detail, creature, insect focus, no humans, spray painted fluorescence on particularly skin, fluorescent, fluorescent dust, fluorescent spray paint, glowing dust, subsurface scattering, Photorealistic, Hyper realistic, analog style, realistic, film photography, soft lighting, heavy shadow.
- portrait in the style of polygonal painting in the style of Alexander Deineka, the face is divided into colored polygons of different sizes, mosaic, a combination of complex cold and warm colors, a play with large color planes, modernism, cubism, sophistication, tenderness and expression. Light hair flutters in the wind, creating picturesque intersecting lines and planes.
- A detailed isometric LEGO diorama of a private plot landscape LEGO design on a slope, showcasing Tokyo Tower and nice parks with plants decorated ocean side. Design was made with LEGO blocks. The diorama includes visual elements like flowers, pathway, grill area, pond, plants and trees. The background is minimalist white, focusing on the structured and secure architecture of the landscape design.
- A metallic apple reflecting a distorted city skyline, hyper-detailed reflections.
- Pasta & Pepper Salad.

**Fig. 3**:

- a bowl of strawberries with powdered sugar sprinkled on top.
- a schoolgirl with twin braids holding a sketchbook.
- a pair of glasses resting on an open notebook.
- a dog toy lying on a tiled kitchen floor.
- a knight in silver armor holding a baner.
- an anime-style slime creature with cute round eyes.

**Fig. 7**:

- saharian landscape at sunset, 4k ultra realism, BY Anton Gorlin, trending on artstation, sharp focus, studio photo, intricate details, highly detailed, by greg rutkowski.
- chinese red blouse, in the style of dreamy and romantic compositions, floral explosions –ar 24:37 –stylize 750 –v 6.
- Digital 2D, Miyazaki's style, ultimate detailed, tiny finnest details, futuristic, sci-fi, magical dreamy landscape scenery, small cute girl living alone with plushified friendly big tanuki in the gigantism of wilderness, intricate round futuristic simple multilayered architecture, habitation cabin in the trees, dramatic soft lightning, rule of thirds, cinematic.

- poster art for the collection of the asian woman, in the style of gloomy, dark orange and white, dynamic anime, realistic watercolors, nintencore, weathercore, mysterious realism –ar 69:128 –s 750 –niji 5.
- A fantasy-themed portrait of a female elf with golden hair and violet eyes, her attire shimmering with iridescent colors, set in an enchanted forest. 8K, best quality, fine details.
- 'very beautiful girl in bright leggings, white short top, charismatic personality, professional photo, style of jessica drossin, super realistic photo, hyper detail, great attention to skin and eyes, professional photo.
- (steampunk atmosphere, a stunning girl with a mecha musume aesthetic, adorned in intricate cyber gogle,) digital art, fractal, 32k UHD high resolution, highres, professional photography, intricate details, masterpiece, perfect anatomy, cinematic angle , cinematic lighting, (dynamic warrior pose:1)
- (Pirate ship sailing into a bioluminescence sea with a galaxy in the sky), epic, 4k, ultra.
- tshirt design, colourful, no background, yoda with sun glasses, dancing at a festival, ink splash, 8k.
- Half-length head portrait of the goddess of autumn with wheat ears on her head, depicted as dreamy and beautiful, by wlop.
- Walter White dressed as a medieval-style king.
- A serene meadow with a tree, river, bridge, and mountains in the background under a slightly overcast sunrise sky.
- A closeup portrait of a gray owl with spreaded wings attacking in cinematic lighting, digital painting by Greg Rutkowski used as album cover art on Artstation.
- A hyena fursona sitting in the grass in a savannah at sunset.
- Large dog looking at television show in living room.
- A fantasy-themed portrait of a female elf with golden hair and violet eyes, her attire shimmering with iridescent colors, set in an enchanted forest. 8K, best quality, fine details.

**Fig. 16**:

- a goldfish swimming in a round glass bowl.
- a ceramic vase with dried flowers on a shelf.
- a notebook with doodles on the cover, anime style with cute screentones.
- a koi fish swimming in a pond with rippling ink-drawn water.
- a snow globe sitting on a windowsill.
- a bench under a tree with fallen leaves.

**Fig. 25**:

- a bird perched on a windowsill with frost outside.
- a steaming cup of tea on a small saucer.
- a shrine gate drawn with sharp black-and-white contrasts.
- an anime-style vampire with elegant clothing and sharp eyes.
- a street covered in fresh snow.
- a commuter waiting at a bus stop during sunset.
- a street with puddles reflecting city lights.
- a basket of oranges on a market stall.
- a bird's nest in the branches of a tree.
- a basket of fresh bread on a kitchen counter.
- a jar of seasalt with a cork lid beside it.

- a cat curled up inside a cardboard box.
- a train arriving at a suburban station in the evening.
- a wooden bench covered in frost.
- a pirate with a large hat ad feathered coat.
- a jar of honey with a wooden dipper beside it.
- a sunflower field under a bright sky, anime style with vibrant colors.
- a fox mask hanging from a wooden wall, anime style with dramatic lighting.
- a bicycle parked under cherry blossom trees, petals falling in manga style.
- an empty parking lot at night.
- a collection of glass bottles lined up on a windowsill catching afternoon sunlight.
- a metal gate with peeling paint.
- a wind-up toy robot on a wooden table, anime style with cel shading.
- an overgrown railway track disappearing into bushes.
- an anime-style fairy with translucent wings glowing softly.
- a glass bottle lying on a sidewalk.
- a jar filled with fireflies, anime style glowing in the dark.
- an anime-style samurai with a tattered cloak standing under moonlight.
- a bottle of olive oil beside a loaf of bread.
- a beach umbrella folded on the sand.

**Fig. 26**:

- Digital art of Prince of Roses.
- A landscape featuring a Kyoto Animation-style building.
- A path winding through a forest depicted in digital art.
- A close-up portrait of a beautiful girl with an autumn leaves headdress and melting wax.
- A neon-soaked cyberpunk alleyway with rain-drenched streets and futuristic holograms, gritty yet vibrant, hyper-realistic, ultra-detailed, cinematic scene.
- A serene mountain landscape at sunrise, mist rolling over rugged peaks, ultra-detailed, photorealistic, soft lighting, high-resolution, digital art.
- A hyper-detailed closeup of a dew-covered insect on a vibrant leaf, extreme macro photography style, ultra-realistic, high-resolution, intricate textures.
- An anime-style magical girl in a dynamic pose, vibrant colors, ultra-detailed costume and background, energetic, high-resolution, cinematic lighting.
- An enchanted autumn forest with falling leaves and warm, glowing light, ultra-detailed, photorealistic, rich textures, digital art, serene mood.
- An elegant Renaissance portrait of a noble figure, detailed textures, soft natural lighting, ultra-detailed, classical, high-resolution, oil painting style.
- A cybernetic humanoid robot portrait with metallic textures and neon accents, ultra-detailed, photorealistic, cinematic, futuristic digital art.
- A vibrant street art mural on an urban wall, ultra-detailed, energetic, bold colors, high-resolution, digital painting, modern art style.
- A dark fantasy warrior in intricately detailed armor standing in a stormy battlefield, ultra-detailed, hyper-realistic, cinematic, dynamic action scene.
- An intense lightning storm over a vast desert landscape, ultra-detailed, dramatic, high-resolution, cinematic, digital art, atmospheric.
- A detailed nature macro shot of a vibrant flower with dewdrops, ultra-detailed, photorealistic, high-resolution, digital painting, delicate textures.

- An ultra-realistic snowy mountain village under a starry sky, ultra-detailed, atmospheric, cinematic, high-resolution, digital winter wonderland.
- A humorous portrait of a cat dressed as a Victorian aristocrat, in vintage photorealism.
- A photorealistic shot of a bouquet of wildflowers in a clear glass vase on a sunlit windowsill.
- A mecha jet fighter engages in an air battle with an explosion as a backdrop, set against a dark, starry sky in a highly-detailed art piece by Stephan Martiniere.
- A young black woman stands in front of a ringed planet in space.
- Digital art of a cherry tree overlooking a valley with a waterfall at sunset.
- An astronaut in white futuristic cybernetic armor running on the surface of the moon, featured in an artwork illustration on Artstation.
- The image is a headshot of a happy girl with white hair in a school uniform, illustrated by Ilya Kuvshinov.
- A minimalistic heart drawing created using Adobe Illustrator.
- The image is a digital art headshot of an owlfolk character with high detail and dramatic lighting.
- Close up of an eye with the Earth inside the pupil, inspired by Wes Anderson's art.
- Landrover drives through a rain-soaked forest in a highly-detailed digital artwork by Greg Rutkowski and Artgerm.
- A snowy lake in Sweden captured in a vibrant, cinematic style with intense detail and raytracing technology showcased on Artstation.
- A heron silhouetted against a beautiful sunrise, created by Greg Rutkowski.
- A surreal portrait of a woman with a giant carnation face in a flower field at sunset with colorful clouds and a large sky, created by artist Simon Stålenhag.

