# OpenReview forum: "Soft-Di[M]O: Improving One-Step Discrete Image Generation with Soft Embeddings"
_ICLR.cc/2026/Conference — ICLR 2026 Poster_

### Official Review · Reviewer_S4vL · 2025-10-25

**Soundness:** 3
**Presentation:** 4
**Contribution:** 3
**Rating:** 6
**Confidence:** 4

**Summary:**

This paper extends the Di[M]O framework by introducing a soft embedding relaxation for discrete diffusion distillation. Instead of hard token sampling, the model uses non-parametric expected embedding under the predicted token distribution, enabling differentiability throughout the training process. This modification makes the framework compatible with gradient-based extensions such as adversarial fine-tuning and preference optimization. Experiments evaluate both class-conditional image generation and text-to-image tasks.

**Strengths:**

1. Clear and intuitive motivation: The paper articulates very well why enabling differentiability is valuable — it broadens the range of learning signals applicable to discrete diffusion models.

2. Reasonable and meaningful novelty: While the relaxation itself (Gumbel–Softmax-style) is known, its application in the context of distilled discrete diffusion models is new and practically relevant. The contribution sits at a “methodological adaptation” level — modest but non-trivial.

3. Clarity of presentation: The derivation of soft embeddings, training objective, and overall algorithm are explained cleanly and are easy to follow. The paper is well written and pedagogical.

4. Empirical coverage across modalities: The inclusion of both class-conditional generation and text-to-image synthesis demonstrates generality of the approach beyond a single domain.

**Weaknesses:**

1. Overemphasis on auxiliary fine-tuning tasks: The paper devotes significant space to adversarial fine-tuning and differentiable reward fine-tuning, but these are not core contributions. They are applications or privileges enabled by the soft embedding, not new methods. This focus dilutes attention from the main relaxation technique, which deserves deeper theoretical and empirical exploration.

2. Limited ablation on the method itself: The paper could better analyze the behavior of the soft embedding mechanism—e.g., temperature schedules, entropy evolution, convergence stability, and how it compares to direct Gumbel-Softmax or straight-through variants.

3. Experimental scope modest: Experiments are conducted on moderate-resolution datasets and lack comparisons with alternative differentiable relaxations. Broader benchmarks or higher-resolution evaluation would strengthen the claims.

**Questions:**

1. Does the relaxation introduce distributional bias or oversmoothing compared to discrete sampling?

2. How does this compare against simpler continuous relaxations like Gumbel-Softmax or parametric estimators in the same setting?

---

> ### Author Response · Authors · 2025-11-22
> **Response to Reviewer S4vL (1/3)**
>
> > ## [**W1**] Overemphasis on auxiliary fine-tuning tasks: The paper devotes significant space to adversarial fine-tuning and differentiable reward fine-tuning, but these are not core contributions. They are applications or privileges enabled by the soft embedding, not new methods. This focus dilutes attention from the main relaxation technique, which deserves deeper theoretical and empirical exploration.
>
> Thank you for the comment. We agree that soft embedding is a key contribution of our work, but we believe that the descriptions of adversarial fine-tuning, reward tuning, and TTEO do NOT ''dilute'' attention. These components are essential for demonstrating the practical value of the soft embedding and ensuring reproducibility.
>
> ### **1. These Fine-Tuning Tasks are Necessary**
>
> Soft embedding is not only a theoretical relaxation, it is designed to *enable* refinement techniques for one-step MDMs and ultimately overcome teacher's limitation.
> To validate this capability, we must show:
> - how soft embedding interacts with different refinement objectives;
> - it produces stable gradients in practice;
> - the resulting one-step generator improves meaningfully across tasks.
>
> The goal of the paper is to **improve the distilled one-step generator beyond the teacher**.
> Thus these empirical demonstrations are therefore necessary to show the value of soft embedding, not auxiliary.
>
> ### **2. Each Refinement Method Involves Nontrivial, MDM-Specific Design**
>
> These subsections are not trivial applications; each method requires careful adaptation to the one-step MDM setting:
>
> - For **adversarial fine-tuning**, we reuse the pretrained teacher as feature extractor for discriminator heads, a design tailored specifically for MDMs; we also introduce techniques including masked-embedding augmentation which is unique to the MDM architecture.
> - For **reward fine-tuning**, it requires modifying the tokenizer decoder and leveraging its embedding matrix to construct differentiable objectives.
> - For **TTEO**, we propose to optimize the embedding corresponding to the initial token sequence, which is specific to one-step MDM rather than a generic test-time scaling technique.
>
> ### **3. Strengthened Theoretical Justification**
>
> In response to the reviewer's feedback, we expanded our theoretical discussion in **Section 3.1.3** and **Appendix G**, where we explain why soft embedding provides a low-bias, low-variance relaxation, which is suitable for adversarial and reward-based post-training.
>
> ### **4. Revised Structure to Highlight Soft Embedding**
>
> To better emphasize the centrality of soft embedding, we restructured Section 3.1 to focus on its definition, empirical benefits, and theoretical justification; we also combine the original Sections 3.2–3.4 be subsubsections under new Section 3.2 "Refinement Techniques Unlocked with Soft Embeddings" to highlight that these refinement techniques are *unlocked* by soft embedding.
>
> > ## [**W2**] Limited ablation on the method itself: The paper could better analysze the behavior of the soft embedding mechanism—e.g., temperature schedules, entropy evolution, convergence stability, and how it compares to direct Gumbel-Softmax or straight-through variants.
>
> Thank you for the suggestion. We agree that analyzing the behavior of the soft-embedding mechanism is important, and we already include several ablations that directly address these concerns. Below we summarize the relevant analyses and clarify them more explicitly.
>
> ### **1. Comparison with Gumbel-Softmax and Straight-Through Variants**
>
> **Figure 4(a)** (class-to-image) and **Figure 11** (text-to-image) provide direct comparisons against Gumbel-Softmax straight-through estimators.
> Our method consistently achieves substantially better FID and GenEval scores.
>
> ### **2. Training Stability and Convergence**
>
> **Figure 12** reports the GAN generator loss and discriminator logits over training.
> Gumbel-Softmax shows **high variance and unstable convergence**, while soft embedding exhibits **smooth and stable** optimization dynamics.
>
> ### **3. Temperature and Logit Sharpness**
>
> **Figure 8** studies the impact of gradient estimator temperature during training.
> Lower temperatures yield better performance, aligning with our theoretical claim that soft embedding is most accurate when logits are sharp (i.e., low-entropy distributions).
>
> ### **4. Entropy and Top-1 Probability Evolution**
>
> In addition, we provide extra experiments results in **Figures 17 and 18** to show the entropy and top-1 evolution of the token distributions during training, demonstrating that the distributions become progressively sharper over time, which further improves the accuracy of the soft-embedding approximation and supports the design choice.
>
> Together, these ablations provide detailed empirical evidence on temperature effects, entropy dynamics, convergence stability, and comparisons to Gumbel-Softmax, directly addressing the reviewer's concerns.

---

> ### Author Response · Authors · 2025-11-22
> **Response to Reviewer S4vL (2/3)**
>
> > ## [**W3**] Experimental scope modest: Experiments are conducted on moderate-resolution datasets and lack comparisons with alternative differentiable relaxations. Broader benchmarks or higher-resolution evaluation would strengthen the claims.
>
> Thank you for the comment. We agree that broader or higher-resolution studies would be valuable, but current experimentation is fundamentally constrained by the availability of pretrained one-step or MDM-style teacher models.
>
> ### **1. Experimental Scope Matches the Current State of the Field**
>
> Our experiments on **ImageNet-256** (using two different teachers) and **text-to-image generation** (also two teachers) follow the standard and widely adopted benchmarks for evaluating one-step generators. These settings are consistent with prior work in this emerging area.
>
> ### **2. High-Resolution Results Included**
>
> To the best of our knowledge, **Meissonic** is the only open-sourced 1024-resolution MDM model that can be considered as teacher model in our framework.
> We include results on Meissonic specifically to demonstrate that soft embedding generalizes to high-resolution generation.
> We will additionally apply soft embedding to any future high-resolution one-step MDM teachers as they become available.
>
> ### **3. Broader Studies Limited by Teacher Availability**
>
> The MDM ecosystem is still in its early stages, and large-scale high-resolution MDM teachers are not yet broadly available. As a results, the experimental space is currently constrained by what pretrained teachers exist publicly.
>
> Despite these limitations, our results show that soft embedding is effective and general across:
> - multiple tasks (ImageNet, T2I);
> - multiple architectures (MaskGit, MaskBit, MaskGen);
> - multiple resolutions (including 1024 via Meissonic).

---

> ### Author Response · Authors · 2025-11-22
> **Response to Reviewer S4vL (3/3)**
>
> > ## [**Q1**] Does the relaxation introduce distributional bias or oversmoothing compared to discrete sampling?
>
> ### 1. Theoretical Analysis of Bias Introduced by Soft Embedding
> In Section 3.1.3 and Appendix G, we provide both theoretical and analyses of the bias introduced by the soft-embedding relaxation.
>
> To be specific, the bias introduced by soft embedding is shown to be of *second-order* in terms of the embedding covariance. Specifically, Lemma G.3 states:
> $$
> |f(\tilde e) - \mathbb{E}[f(e_j)]| \le \tfrac12 L \|\Sigma\|,
> $$
>
> where $\Sigma = \mathbb{E}_{j } [(e_j - \tilde{e})(e_j - \tilde{e})]$ is the covariance of the token embeddings under the predicted distribution. In practice, for one-step MDMs distilled from strong teachers, the logits are highly concentrated, leading to small $\|\Sigma\|$ and thus minimal bias.
>
> ### 2. Empirical Validation of Minimal Bias
>
> In Figure 7 in Section I.1, we show that the output images decoded with sampled tokens from the predicted distribution and those decoded from soft embeddings are visually similar, indicating that the soft embedding does not introduce significant bias or over-smoothing.
>
> On the other hand, we show that sampling with Argmax from logits gives the almost identical performance to sample from the logits. This verifies that the logits are highly concentrated, hence the bias introduced by soft embedding is minimal:
>
> | Method   | FID-50K | IS  | Precision | Recall |
> | ----------- | ----------- | ----------- | ----------- | ----------- |
> | Sample _from logits  Categorical(logits).sample_ | 1.56  | 273.2 | 0.81  | 0.60  |
> | Argmax _torch.argmax(logits, dim=-1)_ |  1.57      |  273.3 |  0.81 |  0.60 |
>
> > ## [**Q2**] How does this compare against simpler continuous relaxations like Gumbel-Softmax or parametric estimators in the same setting?
>
> Thank you for the question. Below we clarify how our method compares with Gumbel-Softmax and other continuous relaxations.
>
> ### **1. Soft Embedding is the Simplest Possible Relaxation**
>
> Our soft-embedding formulation the *simplest* continuous relaxation available to the best of our knowledge, because:
> - it directly computes the expected embedding under the model’s categorical distribution.
> - introduces **no additional parameters**.
> - requires **no sampling**, temperature schedules, or straight-through heuristics.
>
> This simplicity is one of its key advantages for post-training one-step MDMs.
>
> ### **2. Comparison with Gumbel-Softmax (Theoretical and Empirical)**
>
> We provide both theoretical and empirical comparisons against Gumbel-Softmax straight-through estimators.
>
> #### **Theoretical (Sec. 3.1.3 and Appendix G)**
> - Gumbel-Softmax ST suffers from **first-order bias** due to the forward/backward mismatch.
> - This bias does **not vanish** as temperature $\to 0$ unless logits are exactly one-hot.
> - Gumbel noise also introduces **high gradient variance**, destabilizing GAN and reward optimization.
>
> #### **Empirical (Fig. 4(a), Fig. 11 and Fig. 12)**
> - Across both class-to-image and text-to-image tasks, soft embedding **substantially outperforms** Gumbel-ST in FID and GenEval score.
> - Training curves (Fig. 12) show Gumbel-ST is **unstable** with large gradient variance, whereas soft embedding is smooth.
>
>
> ### **3. Comparison with Parametric Estimators**
>
> For parametric estimators, while they may offer some benefits in certain contexts, they typically introduce additional complexity and hyperparameters that can complicate training.
> In contrast, our soft-embedding approach is parameter-free and straightforward to implement, making it more practical for post-training one-step MDMs.
>
> We would like to compare with specific parametric estimators if the reviewer can point to methods applicable to discrete MDM.

---

### Official Review · Reviewer_UkJL · 2025-10-31

**Soundness:** 3
**Presentation:** 3
**Contribution:** 3
**Rating:** 4
**Confidence:** 4

**Summary:**

This paper addresses a key limitation of one-step generative models distilled from Masked Diffusion Models: their discrete token blocks gradient flow. The authors propose soft embeddings, a simple and effective relaxation that replaces discrete tokens with the expected embedding under the generator's output probability distribution, which makes the one-step generator differentiable. By integrating this into the Di[M]O framework, the method unlocks the ability to apply refinement techniques, including GAN-based training, differentiable reward fine-tuning, and Test-Time Embedding Optimization. Empirically, it achieves state-of-the-art one-step results, notably a 1.56 FID on ImageNet-256 and improved alignment scores on text-to-image tasks.

**Strengths:**

- The paper identifies a significant and practical problem (blocked gradients in one-step discrete generators) and proposes an simple solution.
- The differentiable feature enables many technique applications like GAN, reward fine-tuning.
- The method achieves strong results on ImageNet ($1.56$ FID). And the MSE between samples decoded by soft embedding and the corresponding discrete embedding is very small.
- The authors conduct comprehensive experiments validating the effectiveness and design choices.

**Weaknesses:**

- The idea of using the expectation of discrete embeddings is not a new technique in machine learning. So its novelty lies in the application to one-step MDMs.
- The generation performance relies on the introduction of GAN loss, which potentially introduces training instability and conplexity of hyperparameter tuning.

**Questions:**

- What's the performance of only applying soft embeddings without GAN loss?

---

> ### Author Response · Authors · 2025-11-22
> **Response to Reviewer UkJL (1/2)**
>
> > ## [**W1**] The idea of using the expectation of discrete embeddings is not a new technique in machine learning. So its novelty lies in the application to one-step MDMs.
>
> Thank you for the comment. We fully agree that taking expectations over discrete embeddings is a known operation. The novelty of our work does not lie in the mathematical operator itself, but in **showing that this relaxation is exactly what makes post-training feasible for one-step MDMs, ultimately enabling the one-step student to overcome its teacher's limitation.**
>
> Our contribution is application-level and enables new capabilities:
>
> - We are the **first to demonstrate** that one-step MDMs, previously incompatible with adversarial or reward-based refinement due to the discrete bottleneck, can be effectively post-trained and overcomes the teacher's limitation.
> - The proposed soft-embedding relaxation provides the theoretically justified **stable, low-variance gradient path** that allows adversarial training, reward fine-tuning, and TTEO to all work reliably in this setting.
> - Other candidate methods (e.g., Gumbel-Softmax ST), while not been explored in any previous work for one-step MDMs, suffer from first-order bias and high variance, making them unsuitable for these post-training pipelines.
>
> We have made this distinction clearer in the revised manuscript and add a short discussion of related relaxation techniques in the appendix.
>
>
> > ## [**W2**] The generation performance relies on the introduction of GAN loss, which potentially introduces training instability and conplexity of hyperparameter tuning.
>
> Thank you for raising this concern. We clarify below why adversarial training does not introduce instability or hyperparameter complexity in our setting.
>
> ### **1. Stable Adversarial Training Similar to Diffusion GAN**
>
> In the class-to-image setup, we observe that our **teacher-reused discriminator architecture** integrates smoothly with the **diffusion-GAN framework** [1]. This framework is specifically designed for better stability in adversarial training and has been validated at scale in prior work [2–4].
>
> ### **2. Hyperparameter Robustness Demonstrated by Ablations**
> Our ablations confirm that the adversarial component is **not sensitive** to hyperparameter choices such as (1) embedding mask ratio (Fig. 4(c)) and (2) GAN loss weight (Fig. 9).
>
> ### **3. GAN Loss is Used Only for Class-to-Image (Not for T2I)**
>
> As stated in Sec. 4, adversarial training is applied **only** in the class-conditioned ImageNet experiments (Sec. 4.2).
> Our text-to-image experiments (Sec. 4.3) **do not use any GAN loss**, and rely solely on reward-based refinement through soft embedding and DiMO loss.
> Thus, the stability concerns apply only to a subset of our experiments.
>
>
> [1] *Wang, Zhendong, Huangjie Zheng, Pengcheng He, Weizhu Chen, and Mingyuan Zhou. "Diffusion-gan: Training gans with diffusion." arXiv preprint arXiv:2206.02262 (2022).*
>
> [2] *Yin, Tianwei, Michaël Gharbi, Taesung Park, Richard Zhang, Eli Shechtman, Fredo Durand, and Bill Freeman. "Improved distribution matching distillation for fast image synthesis." Advances in neural information processing systems 37 (2024): 47455-47487.*
>
> [3] *Zhou, Mingyuan, Yi Gu, and Zhendong Wang. "Few-step diffusion via score identity distillation." arXiv preprint arXiv:2505.12674 (2025).*
>
> [4] *Lin, Shanchuan, Xin Xia, Yuxi Ren, Ceyuan Yang, Xuefeng Xiao, and Lu Jiang. "Diffusion adversarial post-training for one-step video generation." arXiv preprint arXiv:2501.08316 (2025).*

---

> ### Author Response · Authors · 2025-11-22
> **Response to Reviewer UkJL (2/2)**
>
> > ## [**Q1**] What's the performance of only applying soft embeddings without GAN loss?
>
> Thank you for the question. We clarify below the relationship between soft embedding and the GAN loss.
>
> ### **1. Soft Embedding vs. GAN Refinement**
> Soft embedding and GAN refinement address different aspects of the post-training pipeline:
>
> - **Soft embedding** provides the *stable, low-variance* continuous relaxation needed for end-to-end optimization through discrete tokens.
> - **GAN refinement** corrects teacher–data mismatch and improves fidelity using real-image from datasets.
>
> Importantly, GAN refinement is only effective *because* soft embedding enables reliable gradient flow. The two components are complementary rather than interchangeable.
> For reference, below is the pipeline showing how soft embedding enables different refinement strategies:
>
>                 ┌─────────────────────────────────┐
>                 │      Soft-Embedding             │
>                 │  (Continuous Representations)   │
>                 └──────────────┬──────────────────┘
>                                │
>              ┌─────────────────┼──────────────────────┐
>              │                 │                      │
>              ▼                 ▼                      ▼
>     ┌────────────────┐  ┌──────────────┐  ┌─────────────────────────┐
>     │  Adversarial   │  │   Reward-    │  │   Test-Time Embedding   │
>     │   Training     │  │    Based     │  │   Optimization (TTEO)   │
>     │                │  │ Fine-Tuning  │  │ (Inference-Time scaling)│
>     │(GAN Refinement)│  │              │  │                         │
>     └───────┬────────┘  └──────┬───────┘  └────────────┬────────────┘
>             │                  │                       │
>             └──────────────────┼───────────────────────┘
>                                │
>                                ▼
>                 ┌────────────────────────────────┐
>                 │  Improved One-Step generation  │
>                 └────────────────────────────────┘
>
> We also updated Section 3.2–3.4 titles to highlight these interactions: now the three subsections are changed to three parallel subsubsections under the main Section 3.2 "Refinement Techniques Unlocked with Soft Embeddings".
>
>
> ### **2. Performance of Soft Embedding *Without* GAN**
>
> This setting already appears in our experiments:
> **all text-to-image (T2I) results in Section 4.2 use only soft embedding + reward tuning, with *no* GAN loss.**
>
> As discussed in the first paragraph of Section 4, adversarial refinement is used only for class-conditional ImageNet (Sec. 4.1), while T2I uses reward fine-tuning alone due to known limitations of GANs in high-diversity text-conditioned generation.
> Thus, the T2I results directly demonstrate the effectiveness of soft embedding *independently* of the GAN objective.

---

> > ### Comment · Reviewer_UkJL · 2025-11-26
> >
> > Thank the authors for providing a detailed rebuttal, which help show the paper's contribution clearly. I am now holding a positive view of this submission. However, under the setting of Table 1 and Table 2, I would like to see the ImageNet performance without GAN loss, since I wonder the effect of sole soft embedding to the FID points (does it help improve performance or hurt performance?). Or is it already in the paper but I missed?

---

> > > ### Author Response · Authors · 2025-11-27
> > >
> > > Thank you for the follow-up question, and we appreciate the reviewer's positive view of the rebuttal. We apologize for any confusion regarding the role of soft embedding versus the GAN objective.
> > >
> > > ## **1. Clarifying the purpose of soft embedding**
> > >
> > > Soft embedding is not designed as a standalone module; rather, it is proposed to  make downstream refinement (GAN loss, reward tuning, TTEO) feasible for one-step MDMs.
> > > The original DiMO objective is **purely data-free**, meaning the student can only learn from the teacher. This is precisely why we introduce additional supervision such as GAN or reward, to go beyond the teacher model.
> > >
> > > Soft embedding provides the stable, low-variance relaxation required to incorporate these additional objectives.
> > > Thus, evaluating ''soft embedding alone'' is not the primary goal of the method, since **its purpose is to unlock these refinement techniques**, as demonstrated in the diagram of our original rebuttal.
> > >
> > > ## **2. Nevertheless, we include this variant in the updated manuscript**
> > >
> > > To fully address the reviewer's curiosity, we have added an extra experiment by applying soft embedding **within the DiMO loss alone**, without any GAN refinement.
> > > This appears in Appendix Sec. I.15 of the revised manuscript.
> > >
> > > ### **2.1 How this variant is constructed**
> > >
> > > As illustrated in Fig. 19 (updated in Appendix I.15):
> > >
> > > - **Standard DiMO (top):**
> > >   sample discrete tokens $x_\theta$ → apply forward mask-diffusion → obtain intermediate state $\tilde{x}_t$ → feed to teacher $\phi$ and auxiliary $\psi$ to compute the DiMO loss gradient to update the one-step generator.
> > >
> > > - **Soft-embedding DiMO (bottom):**
> > >   compute logits $z_\theta$ → convert directly to soft embeddings via the *teacher's embedding layer* → apply the same mask schedule (replace some positions with embedding of mask-token) → feed to $\phi$ and $\psi$ to compute the DiMO loss gradient.
> > >
> > > No teacher or auxiliary parameters are changed; only that we use soft embedding to replace the original sampled tokens in the forward calculation, and *the DiMO loss gradient will not backpropagate through the soft embedding calculation* as in standard DiMO.
> > > This marks the difference with the refinement technique that requires backpropagate through the soft embedding calculation.
> > >
> > > ### **2.2 Results**
> > >
> > > By conducting experiment on MaskBit teacher (corresponding to Table 1 in the main paper), we get the following result:
> > >
> > > | Method   | FID-50K | IS  | Precision | Recall |
> > > | ----------- | ----------- | ----------- | ----------- | ----------- |
> > > | Standard DiMO | 2.89  |  310.1  | 0.87  | 0.49  |
> > > | Soft embedding variant DiMO |  3.33      |  315.9 |  0.88 |  0.47 |
> > >
> > > This table shows that soft embedding variant of DiMO has a slight degradation in FID and recall compared to standard DiMO, likely due to the smoothing effect of soft embedding. However, the IS score and precision improves slightly, potentially indicating that soft embedding helps generate higher-quality samples at the cost of diversity.
> > > This result also suggests that soft embedding serves as a good representation for the sampled tokens of one-step generator.
> > > Additional visualization of the FID curve in the distillation process is provided in Figure 20 in Appendix Sec I.15.
> > >
> > > In summary, these results help isolate the behavior of soft embedding inside DiMO, but we emphasize that this setting does **not** reflect the intended use of the soft embedding: the main contribution of soft embedding is to **enable** adversarial, reward-based, and test-time refinement of one-step MDMs, capabilities that were previously infeasible.
> > > Showing soft embedding alone is therefore mainly intended for illustration, rather than as a central empirical claim of the paper.
> > >
> > > We hope this both clarifies the purpose of soft embedding and addresses the reviewer's follow up question.

---

### Official Review · Reviewer_jPZF · 2025-11-02

**Soundness:** 3
**Presentation:** 3
**Contribution:** 3
**Rating:** 6
**Confidence:** 4

**Summary:**

The paper proposes Soft-Di[M]O, a one-step discrete image generator distilled from masked diffusion teachers (e.g., MaskBit, MaskGen). The key idea is to replace sampled discrete tokens with soft embeddings—the expectation of token embeddings under the model’s output distribution—so gradients can flow through the generator and decoder. This makes one-step students end-to-end trainable, enabling (i) GAN-based refinement, (ii) differentiable reward fine-tuning (e.g., CLIP/ImageReward).

**Strengths:**

1. Soft embeddings (expected embedding; Eq. (3)) are a minimal change that preserves compatibility with teacher backbones/tokenizers while restoring differentiability.

2. The framework cleanly integrates GAN loss, reward losses, broadening what one-step discrete students can optimize.

3. The paper studies discriminator input choices and mask schedules.

**Weaknesses:**

1. It’s unclear how much improvement comes from the GAN refinement versus the soft-embedding relaxation itself. While Fig.-style ablations compare Di[M]O vs GAN only vs combined, it only reports 5k FID.

2. TTEO (test-time scaling) increases inference compute; the paper could quantify cost-vs-quality more explicitly. For TTEO, what selection metric do you use to pick the final image—e.g., CLIP score, ImageReward/HPS, discriminator logit for GenEval and HPS v2.1?

**Questions:**

For TTEO, what selection metric do you use to pick the final image—e.g., CLIP score, ImageReward/HPS, discriminator logit for GenEval and HPS v2.1?

---

> ### Author Response · Authors · 2025-11-22
> **Response to Reviewer jPZF (1/2)**
>
> > ## [**W1**] It's unclear how much improvement comes from the GAN refinement versus the soft-embedding relaxation itself. While Fig.-style ablations compare Di[M]O vs GAN only vs combined, it only reports 5k FID.
>
> We thank the reviewer for the question. The GAN refinement and the soft-embedding relaxation serve *distinct and complementary* roles in our post-training pipeline, and their effects are not interchangeable.
>
> ### **1. Complementary Roles of Soft Embedding and GAN Refinement**
>
> - **Soft embedding** provides a *stable, low-variance* continuous relaxation that enables gradients to flow through discrete tokens and hence end-to-end optimization.
>
> - **GAN refinement** aims to overcome distributional mismatches of the teacher model and improve image fidelity by introducing adversarial feedback from real data.
> However, it is only effective because soft embedding supplies well-behaved gradients.
>
> To emphasize these dependencies, we present below a chart illustrating how each module interrelates in our pipeline:
>
>                 ┌─────────────────────────────────┐
>                 │         Soft-Embedding          │
>                 │  (Continuous Representations)   │
>                 └──────────────┬──────────────────┘
>                                │
>              ┌─────────────────┼──────────────────────┐
>              │                 │                      │
>              ▼                 ▼                      ▼
>     ┌────────────────┐  ┌──────────────┐  ┌─────────────────────────┐
>     │  Adversarial   │  │   Reward-    │  │   Test-Time Embedding   │
>     │   Training     │  │    Based     │  │   Optimization (TTEO)   │
>     │                │  │ Fine-Tuning  │  │ (Inference-Time scaling)│
>     │(GAN Refinement)│  │              │  │                         │
>     └───────┬────────┘  └──────┬───────┘  └────────────┬────────────┘
>             │                  │                       │
>             └──────────────────┼───────────────────────┘
>                                │
>                                ▼
>                 ┌────────────────────────────────┐
>                 │  Improved One-Step generation  │
>                 └────────────────────────────────┘
>
> We also updated Section 3.2–3.4 titles to highlight these interactions: now the three subsections are changed to three parallel subsubsections under the main Section 3.2 "Refinement Techniques Unlocked with Soft Embeddings".
>
> ### **2. Why We Use FID-5k for Fig.-style Ablations**
>
> Using FID-5k or FID-10k for ablation studies is standard practice [1,2] because it allows us to efficiently assess:
> - training dynamics;
> - convergence behavior;
> - the isolated contribution of each module.
>
> without incurring the prohibitive computational cost of repeatedly computing full-sample FID during training.
>
> For all *final* quantitative comparisons, we follow standard evaluation protocols:
>
> - **FID-50k** for ImageNet experiments.
> - **FID-30k** for COCO and other text-to-image setups.
>
> [1] *Peebles, William, and Saining Xie. "Scalable diffusion models with transformers." In Proceedings of the IEEE/CVF international conference on computer vision, pp. 4195-4205. 2023.*
>
> [2] *Ma, Nanye, Mark Goldstein, Michael S. Albergo, Nicholas M. Boffi, Eric Vanden-Eijnden, and Saining Xie. "Sit: Exploring flow and diffusion-based generative models with scalable interpolant transformers." In European Conference on Computer Vision, pp. 23-40. Cham: Springer Nature Switzerland, 2024.*

---

> ### Author Response · Authors · 2025-11-22
> **Response to Reviewer jPZF (2/2)**
>
> > ## [**W2 & Q1**] [*W2*] TTEO (test-time scaling) increases inference compute; the paper could quantify cost-vs-quality more explicitly. For TTEO, what selection metric do you use to pick the final image—e.g., CLIP score, ImageReward/HPS, discriminator logit for GenEval and HPS v2.1? [*Q1*] For TTEO, what selection metric do you use to pick the final image—e.g., CLIP score, ImageReward/HPS, discriminator logit for GenEval and HPS v2.1?
>
> We thank the reviewers for the questions regarding the cost–quality tradeoff of TTEO and the selection metric used during test-time scaling.
>
> ### **1. Cost–Quality Tradeoff of TTEO**
>
> We explicitly quantify the computation–quality relationship in Sec. I.8 and Fig. 14, where we report quality–saturation curves under different N of BoN and test-time update steps. We have made this reference more visible in the main text.
>
> To further clarify the runtime overhead, we include the following table (now Table 6 in appendix) showing per-prompt inference time (batch size = 4) on a single NVIDIA H100 GPU for different BoN and TTEO iteration configurations:
>
>
> | **Configuration** | **BoN=8** | **BoN=16** | **BoN=32** | **BoN=64** | **BoN=4 & TTEO iter=2** | **BoN=8 & TTEO iter=2** | **BoN=4 & TTEO iter=4** | **BoN=8 & TTEO iter=4** |
> |---|---|---|---|---|---|---|---|---|
> | **Runtime (s/prompt)** | 1.46 | 2.64 | 4.70 | 8.57 | 2.52 | 4.32 | 5.66 | 9.88 |
>
>
> These results illustrate that TTEO provides a controllable cost–quality tradeoff: moderate TTEO iterations often match or exceed large-BoN performance at lower computational cost.
>
> ### **2. Selection Metric for TTEO**
>
> For selecting the final image during test-time optimization, we follow **Reno** and use the *same weighted combination* of four widely adopted evaluation metrics:
> - **CLIP score**;
> - **ImageReward**;
> - **HPS v2.0**;
> - **PickScore**.
>
> Using the same selection criterion ensures consistency with prior test-time scaling literature and avoids introducing task-specific bias. We have clarified this choice in the revision.

---

### Official Review · Reviewer_rzsv · 2025-11-04

**Soundness:** 4
**Presentation:** 3
**Contribution:** 3
**Rating:** 4
**Confidence:** 4

**Summary:**

The paper introduces Soft-DiMO, an extension to DiMO with soft-embeddings that allow differentiating through the discrete token sampling step for post-training use cases such as supervised finetuning and human-alignment / prompt-adherence. The paper shows the superiority of their method compared to other methods, like the Gumbel Softmax trick, which allows differentiating through discrete sampling steps. The paper shows a SoTA FID score on discrete image generation trained on ImageNet 256x256.

**Strengths:**

- The authors propose a simple way to backrprop through the discrete sampling step that unlocks a wide variety of potential post-training applications.
- The authors demonstrate various post-training techniques using their method and show better performance
- The authors achieve a SoTA FID score on ImageNet 256x256
- The paper also compares the presented method to a multiple existing works

**Weaknesses:**

Minor:
- It is helpful for the reader to have the abstract be independent and include inline full citations of terms used like DiMO, MaskBit, MaskGen, HPS, etc. E.g.: (Author et al. 2022), in line with the abstract.

Major:
- I've had issues with the readability of the paper especially in Section 2 and Section 3 where the problem statement is not well set up to be complete. For instance, what is the space of vocabulary V for this problem? What are the typical patch sizes for each token? What dimensions are the output probability logits for each token?
- The central idea of the paper is to weight the output token classification probability with the corresponding token embedding to obtain a so-called "soft-embedding" that can be used to backprop to empirically make post-training work better than other techniques.
-- However, the novelty is limited since the proposed soft-embedding is a weighted average w.r.t the word probability.
-- Empirically, it is clear that it works better (at least in the hyperparam region used for experiments). But it is not clear *why* this works better, either intuitively and/or theoretically. Is this formulation really approximating backproping through the discrete sampling step better than the Gumbel-Softmax trick? If so, why? Is it purely due to optimization landscapes that are hard to theoretically prove or can some theoretical arguments be made to understand why this works better empirically?

**Questions:**

1. On Line 44, the authors write "First, they inherit modeling errors from the teacher" as one of the challenges of current one-step discrete generators. Can the authors please clarify why this is a problem? Are the authors suggesting that the errors get distilled to the student in preference to the image statistics in a more prominent way than they occur in the teacher model? Given the teacher is a powerful model with minimal region for mistakes, wouldn't the student model also be minimal in the param region for mistakes (since zero errors is unavoidable)?
2. My central question is around the arguments for why the proposed method works better than the others like Gumbel-Softmax other than a qualitative hand-wavy argument about high variance and high bias training signals. It is unclear why the proposed method alleviates such problems. A theoretical argument (despite any assumptions made) would help the reader to appreciate the idea proposed more than an empirical hack to circumvent post-training challenges of discrete sampling models.

---

> ### Author Response · Authors · 2025-11-22
> **Response to Reviewer rzsv (1/3)**
>
> > ## [**W1**] It is helpful for the reader to have the abstract be independent and include inline full citations of terms used like DiMO, MaskBit, MaskGen, HPS, etc. E.g.: (Author et al. 2022), in line with the abstract.
>
> We agree that the abstract should be self-contained and accessible to readers unfamiliar with specific terminology. In the updated version, we have revised the abstract to briefly clarify all specialized terms (e.g., DiMO, MaskBit, MaskGen, HPS etc.) and include corresponding inline citations. This ensures that the abstract is fully independent and easier to understand without referring to the main text.
>
>
> > ## [**W2**] I've had issues with the readability of the paper especially in Section 2 and Section 3 where the problem statement is not well set up to be complete. For instance, what is the space of vocabulary V for this problem? What are the typical patch sizes for each token? What dimensions are the output probability logits for each token?
>
>
> We thank the reviewer for pointing out the clarity issues in Sections 2 and 3. We agree that the problem setup benefits from more explicit definitions, and we have revised the text accordingly.
>
> The vocabulary $\mathcal{V}$ is inherited directly from the pretrained teacher model, and its size varies depending on the specific teacher used in our experiments.
> Likewise, the token patch sizes and the dimensionality of the output logits follow the design of each teacher model. We now state these details explicitly in Section 2.
>
> To further improve readability, we updated Table 5 in the appendix to list the exact vocabulary sizes, patch sizes, and other relevant specifications for each teacher model used in our experiments, as shown in the following:
>
> | Teacher Models     | MaskGit | MaskBit  | Meissonic | MaskGen (1D) |
> | ----------- | ----------- | ----------- | ----------- | ----------- |
> | Generation image resolution     | 256x256      |  256x256      | 1024x1024      | 256x256      |
> | Sequence length (L) |  256       |  256  |  4096  |  128  |
> | Downsampling factor (Patch size) | 16x16 | 16x16 | 16x16 | 512(1D) |
> | Vocabulary size (V) |  1024   |   16384  | 8192 |   8192 |
> | Output logits dimension | [B,L,V+1] | [B,L,2,128]  |  [B,L,V]  |  [B,L,V]  |
>
> Where $B$ is the batch size, $V+1$ in MaskGit suggests the included mask token, the logits shape $[B,L,2,128]$ in MaskBit corresponds to the split group strategy where vocabulary size $V=16384=2^{14}$ is split into two 7-bit ($128=2^7$) groups .

---

> ### Author Response · Authors · 2025-11-22
> **Response to Reviewer rzsv (2/3)**
>
> > ## [**W3**] The central idea of the paper is to weight the output token classification probability with the corresponding token embedding to obtain a so-called "soft-embedding" that can be used to backprop to empirically make post-training work better than other techniques. -- However, the novelty is limited since the proposed soft-embedding is a weighted average w.r.t the word probability. -- Empirically, it is clear that it works better (at least in the hyperparam region used for experiments). But it is not clear why this works better, either intuitively and/or theoretically. Is this formulation really approximating backproping through the discrete sampling step better than the Gumbel-Softmax trick? If so, why? Is it purely due to optimization landscapes that are hard to theoretically prove or can some theoretical arguments be made to understand why this works better empirically?
>
> We thank the reviewer for the detailed and constructive comments. Below we address (1) the concern about novelty and (2) the request for clearer intuitive and theoretical justification of why soft embedding works better than Gumbel-Softmax.
> We have also revised the manuscript to clarify these points.
>
> ### **1. On the Novelty of the Paper**
>
> We would like to clarify that the contribution of our work extends beyond the introduction of soft embedding.
>
> The central novelty lies in **enabling post-training for one-step discrete generators**, a capability that did not previously exist. One-step discrete generators were traditionally **not** compatible with adversarial or reward-based refinement because discrete sampling blocks gradient flow.
> Soft embedding is introduced as our **key contribution**, providing a practical and theoretically grounded mechanism to *unlock such post-training and overcome the teacher limitation*. To make this point clear, we updated the contribution part of the introduction in our revised manuscript.
>
> ### **2. Why Soft Embedding Works Better than Gumbel-Softmax**
>
> We have re-structured Section 3.1 to have three parts: (i) introduction of soft embedding; (ii) empirical benefits of soft embedding and (iii) theoretical explanations of soft embedding.
> Our revised Section 3.1.3 now provides both intuitive and theoretical reasons, with more detailed explanations and formal results (**Lemma G.3** and **Lemma G.4**) in the Appendix Sec. G.
> Below we summarize the key arguments:
> #### **(A) Theoretical Justification**
> 1. **Soft embedding introduces only _second-order bias_.**
>    Lemma G.3 shows:
>    $$
>    |f(\tilde e) - \mathbb{E}[f(e_j)]| \le \tfrac12 L \|\Sigma\|.
>    $$
>    The bias depends on the embedding covariance $\Sigma = \mathbb{E}_{j } [(e_j - \tilde{e})(e_j - \tilde{e})]$.
>    For one-step MDM generators, the output logits are typically concentrated on a dominant token $j^\star$ with $p (j^\star) = 1 - \epsilon$ (as validated in prior work and our experiments in Appendix I.1 I.13 and I.14). In this regime, the embedding covariance satisfies
>    $$
>    \|\Sigma\| = O(\epsilon),
>    $$
>    and therefore the soft-embedding introduces only a negligible bias.
>
> 2. **Gumbel-Softmax ST has _first-order_ bias that does not vanish.**
>    Lemma G.4 shows the gradient mismatch term
>    $$
>    \mathbb{E}\big[\|\hat e_{\mathrm{hard}} - \hat e_{\mathrm{soft}}\|\big]
>    $$
>    yields **first-order error** due to the forward/backward inconsistency of the straight-through estimator. This mismatch persists even as $\tau \to 0$ unless the logits are exactly one-hot, which is rare in practice.
>
> 3. **Gumbel noise injects high gradient variance.**
>    The stochastic Gumbel perturbation also increases gradient variance, which destabilizes adversarial or reward fine-tunning.
>    In contrast, soft embedding provides deterministic, low-variance gradients.
>
> We summarize the bias-variance comparison in the table below:
> | **Method** | **Bias** | **Variance** |
> |---|---|---|
> | Gumbel-Softmax ST | High | High |
> | Soft Embedding (ours) | Small (2nd-order) | Very low |
>
> #### **(B) Intuitive Explanation**
> - **One-step generators naturally produce sharp logits** because they are distilled from strong masked models.
>   In this regime, the soft embedding $\tilde e$ closely matches the embedding of the sampled token, while Gumbel-ST would suffer from mismatch and noise.
> - **Refinement methods (GAN training, reward tine-tuning) are highly sensitive to gradient noise.**
>   The deterministic gradients of soft embedding lead to more stable optimization than the noisy gradients produced by Gumbel-ST.
>
> We show that sampling with Argmax from logits gives the almost identical performance to sample from the logits, verifying that the logits are highly concentrated:
> | Method   | FID-50K | IS  | Precision | Recall |
> | -------- | -------- | -------- | -------- | -------- |
> | Sample from logits  _Categorical(logits).sample_ | 1.56  | 273.2 | 0.81  | 0.60  |
> | Argmax _torch.argmax(logits, dim=-1)_ |  1.57      |  273.3 |  0.81 |  0.60 |

---

> ### Author Response · Authors · 2025-11-22
> **Response to Reviewer rzsv (3/3)**
>
> > ## [**Q1**] On Line 44, the authors write "First, they inherit modeling errors from the teacher" as one of the challenges of current one-step discrete generators. Can the authors please clarify why this is a problem? Are the authors suggesting that the errors get distilled to the student in preference to the image statistics in a more prominent way than they occur in the teacher model? Given the teacher is a powerful model with minimal region for mistakes, wouldn't the student model also be minimal in the param region for mistakes (since zero errors is unavoidable)?
>
> Thank you for the question. We clarify that our point is *not* that the teacher is weak or inaccurate, but that **data-free distillation inherently limits the student to reproducing exactly what the teacher models**, without any ability to correct or adapt. This becomes problematic in two practical scenarios:
>
> ### **1. Dataset Under-Modeling (Our GAN Fine-tuning Experiments)**
>
> Even very strong pretrained teachers exhibit some degree of distributional mismatch or under-modeling of the training data.
> A student trained purely via data-free distillation (such as DiMO) receives **no signal other than the teacher's predictions**, and therefore:
> - it can *only* imitate the teacher's distribution.
> - it cannot compensate for the teacher's imperfections.
> - in some cases, it may even **amplify** the mismatch due to the compression into a one-step generator.
>
> By introducing real data through adversarial training, our method allows the student to correct these discrepancies.
>
>
> ### **2. Task or Preference Misalignment (Our Reward Fine-tuning Experiments)**
>
> In many applications, the desired behavior differs from the teacher's default generation distribution, e.g., aesthetic preferences, safety or reward-based alignment, domain specialization.
>
> A data-free distillation pipeline offers *zero supervision* about these downstream objectives, so the student cannot deviate from the teacher when:
>
> - the teacher is not trained for the target preference.
> - the downstream reward explicitly favors a different distribution.
>
> Our reward-based fine-tuning addresses this issue by providing **new objectives that the teacher itself does not encode**.
>
> ### **3. Summary**
> Our argument is therefore not that the teacher is ''bad'', but that **data-free distillation is inflexible**, which motivates the soft embedding we propose to enable post-training refinement.
>
> We have clarified this motivation in the revised manuscript.
>
> > ## [**Q2**] My central question is around the arguments for why the proposed method works better than the others like Gumbel-Softmax other than a qualitative hand-wavy argument about high variance and high bias training signals. It is unclear why the proposed method alleviates such problems. A theoretical argument (despite any assumptions made) would help the reader to appreciate the idea proposed more than an empirical hack to circumvent post-training challenges of discrete sampling models.
>
> Thank you for the insightful question. We refer to our detailed response to **[W3]**, where we provide theoretical explanations for why soft embedding offers a better and lower-bias relaxation than Gumbel-Softmax in one-step discrete generators, under reasonable assumptions.
>
> In the revised manuscript, we have dedicated Section G in the appendix to present these theoretical results formally.

---

### Author Response · Authors · 2025-11-22
**General Response**

We sincerely thank all the reviewers for the constructive feedback and questions. In the revised version, we clarify the core contributions of our work and reorganize the Method section 3 to make these contributions more explicit. All changes are highlighted in the updated manuscript.

## **Highlighting Our Contributions**

Our main contribution is to make **post-training feasible for one-step MDMs**, a capability that was previously unavailable due to the discrete, non-differentiable nature of masked-token generators.
To achieve this, we introduce **soft embedding**, a minimal, simple and decoder-compatible continuous relaxation that enables stable end-to-end gradient flow through discrete outputs.

Building on this relaxation, we further demonstrate that one-step MDMs can be effectively improved using **three refinement techniques**, adversarial fine-tuning, reward-based fine-tuning, and test-time embedding optimization (TTEO), which were not previously applicable in this setting.
Across class-to-image and text-to-image benchmarks, our method consistently outperforms the teacher MDMs and existing distilled variants.

## **Clarifying the New Method Section Structure**

To improve clarity, we restructure the Method section around the role of soft embeddings:

1. **Section 3.1 — Soft Embedding (Core Contribution)**
   - Introduces the formulation;
   - Explains intuitively why it enables gradient-based optimization for one-step MDMs;
   - Provides theoretical justification (bias/variance).

2. **Section 3.2 — Refinement Techniques Unlocked by Soft Embeddings (Core Applications to Verify the Effectiveness of Soft Embedding)**
   We reorganize adversarial fine-tuning, reward tuning, and TTEO into *parallel subsubsections*, all under a unified section that emphasizes:
   - these are **applications enabled by soft embedding**;
   - each method requires MDM-specific design choices (e.g., teacher-as-discriminator, embedding-optimized TTEO).

This new structure makes the contribution hierarchy clear:
**soft embedding is the foundation**, and the refinement techniques are **capabilities unlocked by this foundation**, demonstrating its practical value.

We believe the revised presentation better reflects the novelty, technical insights, and scope of the work.
Thank you again for your time and efforts. We look forward to your feedback.

---

### Meta-Review · Area_Chair_uH6e · 2026-01-06

**Summary:**

This paper focuses on the one-step distillation for Masked Diffusion Models. Specifically, it aims at addressing two key limitations of previous works: modeling bias from the teacher and discrete token outputs blocking gradient flow. This paper introduces soft embeddings to DiMO distilllation framework, improving the performance with GAN-based refinement, reward fine-tuning and Test-Time Embedding Optimization. In the initial review, this paper is rated by 4,6,4, and 6, with main concerns lying in comparison with Gumbel-Softmax, ablation on soft embeddings and limited novelty. During rebuttal, authors provide explanation and empirical results, which mostly address the concerns. Therefore, AC recommends this paper as **Accept (Poster)**.

**Reviewer Concerns:**

In the initial review, the main concerns are comparison with Gumbel-Softmax, ablation on soft embeddings and limited novelty.

Comparison with Gumbel-Softmax: authors provide theoretical and empirical comparison with Gumbel-Softmax, demonstrating that soft embedding has smaller bias and lower variance.

Ablation on soft embeddings: authors provide ablation studies on GAN loss, temperature schedules, entropy evolution, and convergence stability.

Limited novelty: although soft embeddings are existing application in discrete representations, applying it in one-step MDM with better performance is still significant.

**Reviewer Scores:**

**Reviewer rzsv** raises to 6. Reviewer rzsv did not have the chance to reply, but the main concern, comparison with gumbel-softmax, is addressed.

**Reviewer jPZF** keeps 6. Reviewer jPZF did not have the chance to reply, but the concerns are mostly addressed.

**Reviewer UkJL** raises to 6. Reviewer UkJL replied to the rebuttal and mentioned to hold a positive view.

**Reviewer S4vL** keeps 6. Reviewer S4vL did not have the chance to reply, but the concerns are mostly addressed.

---

### Decision · Program_Chairs · 2026-01-26

Accept (Poster)